# TSPulse: Tiny Pre-Trained Models with Disentangled Representations for Rapid Time-Series Analysis

**Vijay Ekambaram, Subodh Kumar[*], Arindam Jati[*], Sumanta Mukherjee[†],
Tomoya Sakai[†], Pankaj Dayama, Wesley M. Gifford, Jayant Kalagnanam**

**IBM Research**
vijaye12@in.ibm.com

## Abstract

Time-series tasks often benefit from signals expressed across multiple representation spaces (e.g., time vs. frequency) and at varying abstraction levels (e.g., local patterns vs. global semantics). However, existing pre-trained time-series models entangle these heterogeneous signals into a single large embedding, limiting transferability and direct zero-shot usability. To address this, we propose TSPulse, family of ultra-light pre-trained models (1M parameters) with disentanglement properties, specialized for various time-series diagnostic tasks. TSPulse introduces a novel pre-training framework that augments masked reconstruction with explicit disentanglement across spaces and abstractions, learning three complementary embedding views (temporal, spectral, and semantic) to effectively enable zero-shot transfer. In-addition, we introduce various lightweight post-hoc fusers that selectively attend and fuse these disentangled views based on task type, enabling simple but effective task specializations. To further improve robustness and mitigate mask-induced bias prevalent in existing approaches, we propose a simple yet effective hybrid masking strategy that enhances missing diversity during pre-training. Despite its compact size, TSPulse achieves strong and consistent gains across four TS diagnostic tasks: +20% on the TSB-AD anomaly detection leaderboard, +25% on similarity search, +50% on imputation, and +5–16% on multivariate classification, outperforming models that are 10–100× larger on over 75 datasets. TSPulse delivers state-of-the-art zero-shot performance, efficient fine-tuning, and supports GPU-free deployment. Models and source code are publicly available at https://huggingface.co/ibm-granite/granite-timeseries-tspulse-r1.

## 1 Introduction

Time-series (TS) analysis encompasses a broad class of problems that aim to extract meaningful insights and semantics from observed sequences. Among these, **time-series diagnostic tasks**—such as anomaly detection, imputation, classification, and similarity search—operate on observed data and focus on retrospective understanding, i.e., analyzing existing sequences to characterize behavior, identify irregularities, recover missing information, or compare patterns across time. These tasks are central to many real-world applications in observability, manufacturing, and industrial monitoring.

Inspired by the success of large language models (LLMs), time-series pre-trained models aim to learn reusable representations from large-scale public data for effective transfer learning. While time-series pre-trained models have seen rapid progress in forecasting (Ansari et al., 2024; Das et al., 2023; Ekambaram et al., 2024), their development for time-series diagnostic tasks remains relatively limited. A few pre-trained models—such as Moment (Goswami et al., 2024), UniTS (Gao et al., 2024), VQShape (Wen et al., 2024), and GPT4TS (Zhou et al., 2023)—support subsets of diagnostic tasks. However, their performance on time-series diagnostic tasks still leaves substantial

---

[*]Equal second authorship.    [†]Equal third authorship.

room for improvement, while their large model sizes further hinder real-time, low-latency deployment—especially in lightweight and CPU-only settings. Specifically, a key limitation underlying these approaches lies in how representations are learned during pre-training. Most existing diagnostic models rely on self-supervised objectives, with masked reconstruction emerging as one of the most widely adopted strategies (Goswami et al., 2024; Gao et al., 2024). While effective in capturing local & global structure, masked reconstruction alone is insufficient to model the full complexity of time-series data.

Meaningful insights in time-series often arises, when signals are examined across different representation spaces(e.g. time vs spectral) and abstraction levels (e.g. local patterns vs global semantics). For example, abrupt spikes and local irregularities are most apparent in the time domain, whereas periodic patterns emerge more clearly in the frequency domain. Likewise, some structures are visible only at fine temporal resolutions, while others manifest at higher semantic levels. To be broadly useful, pre-trained embeddings must therefore capture these complementary cues across both spaces and abstraction levels. However, simply learning them jointly within a single embedding often leads to entanglement, making it difficult for downstream tasks to selectively access the information they require. For broad utility—particularly in zero-shot transfer—representations must explicitly expose these insights in a disentangled form, enabling temporal, spectral, and semantic signals to be accessed as needed.

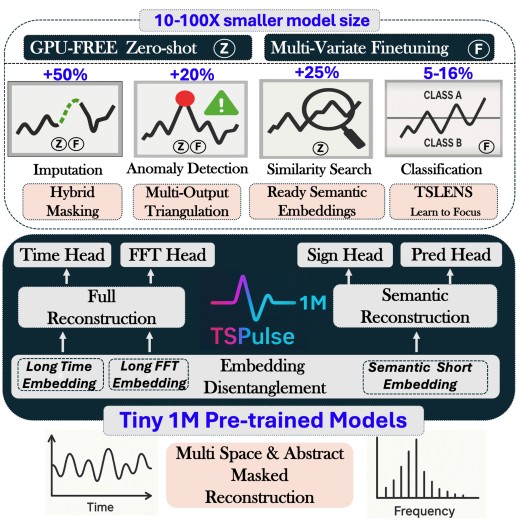

Figure 1: **TSPulse Overview.** **%** and **X** represent the accuracy and size improvements of TSPulse over SOTA pre-trained models across different benchmarks.

To address these challenges, TSPulse proposes a novel pretraining framework to enable **disentangled masked reconstruction across multiple spaces and abstraction levels**, explicitly producing three distinct types of embeddings during pre-training: (i) detailed temporal embeddings for fine-grained time analysis, (ii) detailed spectral embeddings for frequency-aware fidelity, and (iii) semantic embeddings for high-level task understanding. The model formulates semantic and full reconstruction objectives across multiple spaces, employing multi-output heads that operate on distinct segments of the embedding to yield disentangled representations across spaces and abstraction levels (Figure 1). While prior works in traditional time-series modelling have explored time–frequency fusion (Zhang et al., 2022; 2023) or investigated disentanglement in isolation (Chang et al., 2024), TSPulse advances beyond these approaches by jointly learning disentangled representations across spaces and abstraction levels within a unified pre-training framework. Optimized together, this design yields substantial gains in both performance and transferability across diverse downstream tasks.

Through extensive sensitivity analyses, we demonstrate that different segments of the learned embeddings indeed capture distinct and complementary properties that substantially enhance transfer learning. In particular, the semantic embedding exhibits strong robustness to various distortions—such as time shifts, magnitude variations, and noise—which is especially important for reliable semantic analysis.

Moreover, since different tasks benefit from different combinations of these disentangled views, we introduce a set of lightweight post-hoc fusers that selectively combine these views based on the task type, providing a simple yet effective mechanism to exploit their complementary strengths for effective task specialization. Specifically, we propose two post-hoc fusers: (i) Multi-Head Triangulation (MHT) for anomaly detection and (ii) TSLens for classification, each demonstrating strong effectiveness for its respective task.

In addition, TSPulse improves pre-training robustness through a simple but impactful refinement of masking strategies. Unlike existing approaches (Goswami et al., 2024) that rely on fixed masking types and span lengths, TSPulse adopts a hybrid masking scheme that randomizes both, better

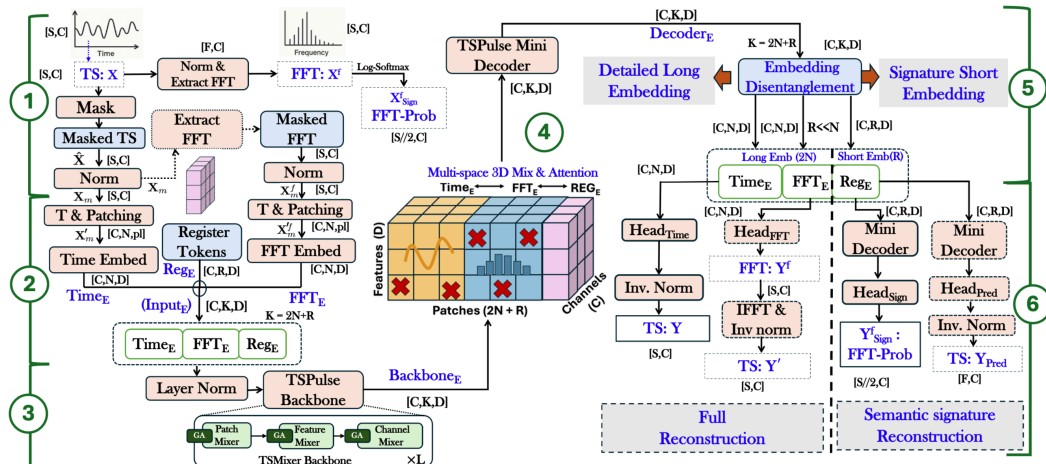

Figure 2: TSPulse Architecture. (X: Inputs, Y: Outputs, E: Embeddings). Annotations for easy reference: ①-⑥.

reflecting real-world missing patterns. Despite its simplicity, this increased corruption diversity reduces overfitting and consistently boosts downstream performance.

Finally, to ensure efficiency, TSPulse replaces conventional Transformer backbones (Vaswani et al., 2017) with light-weight TSMixers (Ekambaram et al., 2023), further enhanced with improved initialization strategies. This substitution, when combined with disentangled hybrid reconstruction, yields small and fast pre-trained models, while still delivering state-of-the-art representational power.

**Contributions: (1) Compact & Versatile.** We introduce a family of ultra-light time-series pre-trained models (1M parameters) with rapid zero-shot and fast multivariate fine-tuning support, specialized for 4 diagnostic tasks: classification, imputation, anomaly detection (AD), and semantic search. **(2) Architectural Novelties.** TSPulse introduces (i) disentangled masked reconstruction across multiple representation spaces and abstraction levels, yielding temporal, spectral, and semantic embeddings; (ii) simple yet effective post-hoc fusers (MHT, TSLens) built on these disentangled views for task-specialization; and (iii) a hybrid masking scheme to mitigate pre-training bias prevalent in existing approaches. **(3) Emergent Representation Properties.** Through extensive sensitivity analyses, we show that the learned disentangled embeddings capture complementary properties and exhibit varied level of robustness to common perturbations, including time shifts, magnitude variations, missingness, spectral perturbations, and noise. **(4) Benchmark Performance.** TSPulse achieves (i) robust anomaly detection with +20% gains on the TSB-AD leaderboard, ranking first in both uni- and multivariate settings; (ii) semantic search improvements of +25% via robust semantic embeddings; (iii) +50% gains in zero-shot imputation under diverse missing patterns; and (iv) +5–16% improvements in multivariate classification on UEA benchmarks. **(5) GPU-Free Deployment.** Despite its compact size, TSPulse consistently matches or outperforms models that are 10–100× larger across a broad range of benchmark datasets, while providing near-instant, CPU-only inference suitable for real-time applications.

## 2 TSPULSE ARCHITECTURE

Let $\mathbf{X} \in \mathbb{R}^{S \times C}$ be a multivariate time series with length $S$ and $C$ channels. We first project and mask $\mathbf{X}$ in both the time and frequency domains. The backbone and decoder then process these representations, mixing information across dimensions in both spaces. To guide learning, we use multi-output heads for semantic and full reconstruction on different parts of the embeddings, which encourages disentangled representation learning. These embeddings can then be directly used across downstream tasks. Figure 2 gives an overview of the framework.

**Masking & RevIN [Fig 2-①]:** TSPulse begins with a *masking block* that hides portions of the input sequence to enable self-supervised reconstruction. Given an input $\mathbf{X} \in \mathbb{R}^{S \times C}$, we divide it into $N$ non-overlapping patches of length $pl$ and apply masking to obtain $\hat{\mathbf{X}} \in \mathbb{R}^{S \times C}$. TSPulse

supports two masking strategies: **block masking** and **hybrid masking**. In block masking, entire patches are randomly replaced with a learnable mask token $\mathbf{M} \in \mathbb{R}^{1 \times pl}$, as commonly done in prior work (Goswami et al., 2024; Gao et al., 2024). While effective for robust feature learning, this approach is inadequate for real-world imputation tasks, where missing values occur irregularly at both patch and point levels. To address this, we introduce a more realistic **hybrid masking** pre-train strategy that masks both full and partial patches within each sample using variable masking ratios, preventing overfitting to fixed patterns. A key design choice in TSPulse is to define the mask token $\mathbf{M} \in \mathbb{R}^{1 \times pl}$ at the *raw patch level*, unlike prior approaches that insert mask tokens in the embedding space (Goswami et al., 2024; Vaswani et al., 2017). This enables individual time-points to be masked by selecting the appropriate value from $\mathbf{M}$ based on their relative index within a patch (Figure 3(a)), allowing a single token to flexibly support both full and partial masking. After masking, a learnable RevIN block (Kim et al., 2022) is applied to normalize the input, yielding $\mathbf{X}_m \in \mathbb{R}^{S \times C}$.

**FFT Extraction [Fig 2-①]:** As TSPulse reconstructs in both time and frequency domains, this block extracts masked FFT features for backbone processing and also prepares the corresponding ground-truth for loss computation. Instead of explicitly masking the frequency space, we feed the scaled and masked time-series $\mathbf{X}_m$ directly into the Fast Fourier Transform (rfft), propagating the mask and ensuring the same data is consistently hidden in both spaces, preventing leaks. The real and imaginary FFT outputs from $\mathbf{X}_m$ are then packaged, scaled, and processed into tensor $\mathbf{X}_m^f \in \mathbb{R}^{S \times C}$ for further backbone processing. Refer to Appendix. A.10 for more details. Simultaneously, two ground-truths are computed from the frequency representation. First, the unmasked scaled time-series is transformed using the same approach described above to get $\mathbf{X}^f \in \mathbb{R}^{S \times C}$, a clean, unaltered frequency representation of the time-series, which is used to guide the model's reconstruction. In addition, we also compute the log-magnitude spectrum of the unmasked time-series and apply softmax to obtain $\mathbf{X}_{\text{sign}}^f \in \mathbb{R}^{S/2 \times C}$, a normalized global frequency signature (Appendix. A.10 for more details). This global signature serves as an auxiliary reconstruction target, helping the model capture high-level semantic patterns and improving generalization to downstream tasks. The log transformation reduces the dynamic range and stabilizes training, while softmax emphasizes dominant spectral components, mapping the output to a probability-like distribution.

**Encoding [Fig 2-②]:** This block projects the input to an embedding space. The masked time-domain input $\mathbf{X}_m \in \mathbb{R}^{S \times C}$ is transposed and then divided into $N$ non-overlapping patches of length $pl$, resulting in a tensor of shape $\mathbb{R}^{C \times N \times pl}$. Now, each patch is projected via a linear layer from $\mathbb{R}^{pl} \to \mathbb{R}^D$ to obtain time-encoded features $\mathbf{Time}_{\text{E}} \in \mathbb{R}^{C \times N \times D}$. Similarly, the masked frequency-domain input $\mathbf{X}_m^f \in \mathbb{R}^{S \times C}$ is transposed, patched and projected to produce frequency-encoded features $\mathbf{FFT}_{\text{E}} \in \mathbb{R}^{C \times N \times D}$. Motivated by recent advances in vision transformers (Darcet et al., 2024), where adding learnable register tokens stabilizes training and improves transfer learning, we introduce $R$ such tokens shared across channels: $\mathbf{Reg}_{\text{E}} \in \mathbb{R}^{C \times R \times D}$. The full input to the backbone is constructed by concatenating time, frequency, and register tokens along the patch axis and layer normalized: $\mathbf{Input}_{\text{E}} = [\mathbf{Time}_{\text{E}}; \mathbf{FFT}_{\text{E}}; \mathbf{Reg}_{\text{E}}] \in \mathbb{R}^{C \times (2N+R) \times D} = \mathbb{R}^{C \times K \times D}$.

**TSPulse Backbone [Fig 2 - ③,④]:** The TSPulse backbone receives $\mathbf{Input}_{\text{E}} \in \mathbb{R}^{C \times K \times D}$, a unified sequence of masked patches from both time and frequency domains, along with learnable register tokens. Its goal is to transform this input into semantically rich, task-robust representations. To maintain efficiency, we use the *TSMixer* backbone (Ekambaram et al., 2023), an MLP-Mixer based alternative to Transformers that performs strongly with reduced compute. TSMixer has stacked Mixer blocks interleaved with lightweight gated attention, enabling flexible feature mixing across three dimensions: within-patch, across-patch, and across channels. Since $\mathbf{Input}_{\text{E}}$ already integrates both time and frequency information, TSMixer effectively fuses these views, learning *dual-space representations* that capture temporal and spectral correlations. Gated attentions in TSMixer further prioritizes informative regions, enhancing the model's ability to generalize across downstream tasks.

**TSPulse Mini-Decoder [Fig 2-④]:** The backbone output ($\mathbf{Backbone}_{\text{E}} \in \mathbb{R}^{C \times K \times D}$) is passed through a lightweight *mini-decoder*, which mirrors the backbone but is only 10–20% of its size to output $\mathbf{Decoder}_{\text{E}} \in \mathbb{R}^{C \times K \times D}$, where $K = 2N + R$. This compact decoder adapts representations during fine-tuning, enabling fast & efficient data-specific adaptation

**Multi-Objective Heads [Fig 2-⑤]:** The decoder output $\mathbf{Decoder}_{\text{E}}$, which consists of 3 segments $[\mathbf{Time}_{\text{E}}; \mathbf{FFT}_{\text{E}}; \mathbf{Reg}_{\text{E}}]$ is disentangled by optimizing each segment with a distinct head objective:

- **Full Reconstruction Heads [Fig 2-⑥]:** The first $N$ patch embeddings ($\mathbf{Time}_\mathrm{E}$) from the decoder pass through a linear layer (Time Head) and inverse RevIN to obtain the full reconstruction $\mathbf{Y}$ of the input time-series. The next $N$ embeddings ($\mathbf{FFT}_\mathrm{E}$) from the decoder are projected (via the FFT Head) to reconstruct the input frequency spectrum $\mathbf{Y}^f$, which is further reshaped, passed through `torch.fft.irfft`, and inverse RevIN to yield $\mathbf{Y}'$, an alternate reconstruction of the input time-series from FFT-space. Losses are computed as Mean Squared Errors (MSE): $\mathcal{L}_{\mathrm{time1}} = \mathrm{MSE}(\mathbf{X}, \mathbf{Y})$, $\mathcal{L}_{\mathrm{time2}} = \mathrm{MSE}(\mathbf{X}, \mathbf{Y}')$, and $\mathcal{L}_{\mathrm{fft}} = \mathrm{MSE}(\mathbf{X}^f, \mathbf{Y}^f)$, where the first two losses are computed only on the masked time-points. This disentangled losses enables $\mathbf{Time}_\mathrm{E}$ and $\mathbf{FFT}_\mathrm{E}$ to capture fine-grained temporal and spectral insights.

- **Semantic Heads [Fig 2-⑥]:** The final $R$ register embeddings (a.k.a semantic embeddings) from $\mathbf{Decoder}_\mathrm{E}$ are primarily trained through a signature head that predicts $\mathbf{Y}^f_{\mathrm{sign}}$, a softmax distribution over the log-magnitude frequency spectrum (i.e. semantic signature). This objective, optimized with cross-entropy loss $\mathcal{L}_{\mathrm{sign}} = \mathrm{CE}(\mathbf{X}^f_{\mathrm{sign}}, \mathbf{Y}^f_{\mathrm{sign}})$, captures global spectral semantics and forms the core of semantic reconstruction. Optionally, a lightweight next-point prediction head $\mathbf{Y}_{\mathrm{pred}}$ can also be added, trained with $\mathcal{L}_{\mathrm{pred}} = \mathrm{MSE}(\mathbf{X}_{\mathrm{pred}}, \mathbf{Y}_{\mathrm{pred}})$, to inject temporal cues into the semantic embeddings. $\mathbf{X}_{\mathrm{pred}}$ denotes the ground-truth future points. Importantly, this auxiliary head is limited to only a few points and is not designed for full-fledged forecasting; its sole purpose is to enrich the semantic representation. Together, these objectives ensure that the register tokens are converted into semantic embeddings for high-level understanding.

Finally, a weighted sum of all the above losses across heads is jointly minimized during pre-training.

## 3  TSPULSE WORKFLOWS

### 3.1  PRE-TRAINING:

TSPulse is pre-trained on diverse $\sim$1B TS samples as detailed in Appendix A.8. Inspired by the success of small, task-specialized pre-trained models in the language/vision domain (Schick & Schütze, 2020; Nguyen et al., 2024; Fu et al., 2023; Ling et al., 2024)—which achieve strong performance through minimal task-specific adaptations—we extend this strategy to time-series. Specifically, we specialize the pre-training for every task through reweighting loss objectives to prioritize heads most relevant to the target task. This enables TSPulse to refine task-specific representations while maintaining its lightweight design, facilitating efficient transfer learning across any datasets for the specified downstream task. Refer Appendix A.9 for more details. Pre-training on 1B samples takes just one day with 8×A100 GPUs, thus there are no practical challenges in pre-training task-specific models. In addition, given the heterogeneous channel counts in pre-training datasets, TSPulse is pre-trained in a univariate mode ($c = 1$), treating each channel independently. Cross-channel modeling is deferred to fine-tuning, where channel-mixing is selectively activated based on the target dataset (Ekambaram et al., 2024).

### 3.2  TARGET DATA FINE-TUNING:

During fine-tuning, the pre-trained model—already strong in zero-shot settings—is further adapted to the target data by updating the decoder and task-specific heads. For multivariate inputs, we enable *channel mixing* in the decoder to capture inter-channel correlations, which are absent in the univariate pre-training setup. Our design draws inspiration from TSMixer and TTM (Ekambaram et al., 2023; 2024), where channel mixer blocks are interleaved between patch and feature mixers within each TSMixer layer. A key limitation in the original design is the *random initialization* of these mixers, which introduces untrained parameters between already pre-trained layers. This can disrupt information flow and create sharp activation shifts, leading to unstable gradient propagation, especially during the early stages of fine-tuning. To address this, we initialize channel mixers with *identity weights*, which enable smooth gradient flow between pre-trained weights. These layers gradually learn inter-channel dependencies without interfering with earlier knowledge, leading to a significantly more stable fine-tuning process, as confirmed by our experiments.

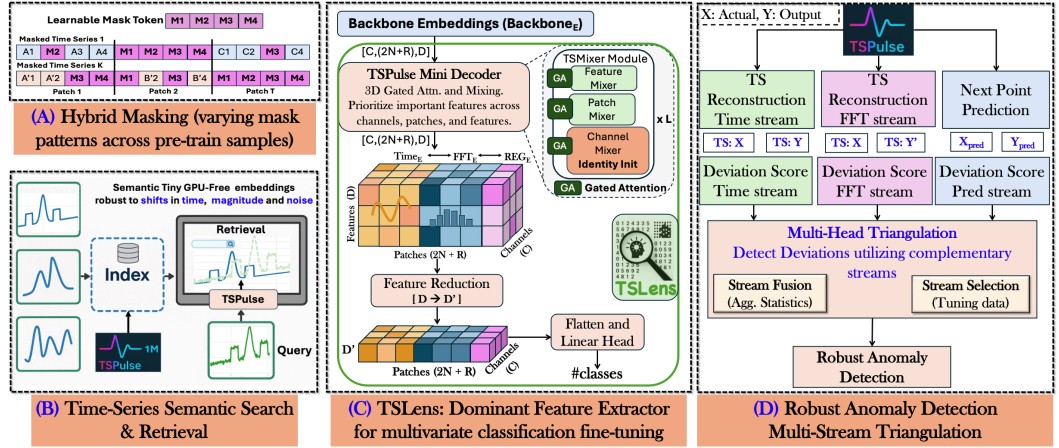

Figure 3: TSPulse Downstream Differentiators.

## 3.3 Downstream Tasks & Post-Hoc Fusers:

TSPulse delivers strong zero-shot imputation benefitting from hybrid masked pre-training (Fig. 3-A), which exposes the model to diverse corruption patterns. For semantic similarity search, the register token embeddings provide invariant representations that remain resilient to time shifts, magnitude changes, and noise (Fig. 3-B). For classification and anomaly detection, TSPulse further incorporates **task-specific post-hoc fusers**, detailed below.

**Classification via TSLens:** TSPulse supports multivariate classification through a lightweight fine-tuning module, **TSLens** (Fig. 3-C), which replaces standard pooling across channels with a learned mechanism that adaptively extracts relevant features from disentangled embeddings. Unlike conventional methods, which average or max-pool all patch-level embeddings across channels followed by a linear head, TSLens selectively attends to and weights features across fine-grained patch embeddings and high-level register tokens.

TSLens takes the backbone output $\mathbf{Backbone}_E \in \mathbb{R}^{C \times (2N+R) \times D}$, passes it through the mini decoder (initialized with pre-trained weights and channel-mixing enabled), and learns cross-channel dependencies via identity-initialized channel mixers as explained in Section 3.2. The resulting representation $\mathbf{H} \in \mathbb{R}^{C \times (2N+R) \times D}$ is projected to a lower-dimensional space $\mathbf{H}' \in \mathbb{R}^{C \times (2N+R) \times D'}$, flattened into $\mathbf{H}_{\text{flat}} \in \mathbb{R}^{C \cdot (2N+R) \cdot D'}$, and passed through a linear layer to produce class logits $\mathbf{y}_{\text{pred}} \in \mathbb{R}^{\texttt{num\_classes}}$. The model is optimized with cross-entropy loss. This design allows TSPulse to dynamically focus on the most informative features across local and global representations, improving classification accuracy across diverse datasets.

**Robust Anomaly Detection via Multi-Head Triangulation:**

In anomaly detection, certain anomalies manifest in the time domain (sudden spikes), others in the frequency domain (periodicity breaks), and others in predictive space (missing trends). TSPulse leverages multi-output heads—$\texttt{Head}_{\texttt{time}}$, $\texttt{Head}_{\texttt{fft}}$, and $\texttt{Head}_{\texttt{pred}}$—to reconstruct or predict from complementary views, capturing signal continuity, spectral consistency, and temporal dynamics. This unified design enables detection of diverse anomaly types.

During inference, anomaly scores are computed from each head based on the deviations between the original and predicted signals(Fig. 3-D). Once the deviations are obtained from all heads, two approaches are possible. **Approach 1** ($\texttt{Head}_{\texttt{ensemble}}$) is to fuse the normalized scores using statistics, such as the maximum, to generate a unified score. In **Approach 2** ($\texttt{Head}_{\texttt{triang.}}$), when a small labeled validation set is available, it can be used to select the most effective head in zero-shot from the above four heads (including $\texttt{Head}_{\texttt{ensemble}}$). This allows the model to adapt to the anomaly type and structure specific to each application. Notably, TSPulse is the first pre-trained model to unify and triangulate multi-space outputs in a single lightweight framework, enabling robust anomaly detection in both zero-shot and fine-tuned settings.

# 4 EXPERIMENTS

We evaluate TSPulse across 4 TS diagnostic tasks: classification, anomaly detection, imputation, and similarity search. Details of the pre-trained model configurations are in Appendix A.9. Pre-training datasets are listed in Table 10, and they do not overlap with any of the evaluation datasets.

## 4.1 ANOMALY DETECTION (AD)

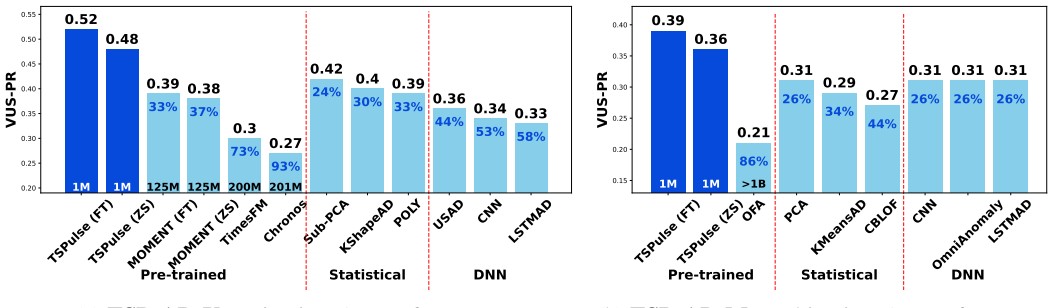

(a) **TSB-AD-U**: Univariate AD Performance          (b) **TSB-AD-M**: Multivariate AD Performance

Figure 4: VUS-PR (official metric, higher is better) scores of different AD methods on the TSB-AD leaderboard; IMP(%)—the percentage improvement of TSPulse over baselines; A maximum of top three SOTA models have been reported in each category. Full results across all 40 models are in Appendix A.11.

**Setup:** We evaluate TSPulse on the TSB-AD benchmark (Liu & Paparrizos, 2024) (recent comprehensive leaderboard for AD), which comprises 40 eval datasets, covering both univariate (TSB-AD-U) and multivariate (TSB-AD-M) anomaly detection. The benchmark includes results from 40 SOTA methods and establishes VUS-PR (Paparrizos et al., 2022) as the primary and robust evaluation metric. A small labeled official *tuning-set* is provided for hyperparameter selection, consistently used across all leaderboard methods. We adopt this tuning set for multi-head triangulation to select the best-performing head and report scores on the test set for both zero-shot (TSPulse-ZS) and fine-tuned (TSPulse-FT) variants. In the zero-shot case, TSPulse is evaluated directly without training on the target data; in the fine-tuned case, it is self-supervised using the official training split without access to any anomaly labels. Note that all non-pretrained neural network models are also trained using the same training split. Figure 4 summarizes VUS-PR results. Full details in Appendix A.11.

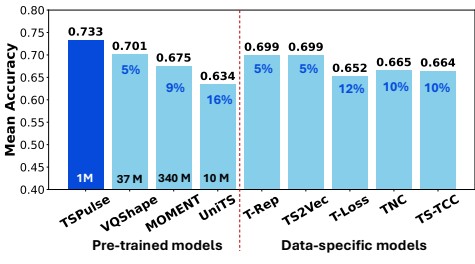

Figure 5: Classification Mean Accuracy results (higher is better); IMP(%)—the percentage improvement of TSPulse over baselines.

**Results:** As illustrated in Figure 4, TSPulse (ZS) outperforms all existing SOTA methods on both the uni and multi-variate AD benchmarks. Specifically, TSPulse (ZS) achieves 14% and 16% higher VUS-PR scores compared to the best-performing baselines—SubPCA for the univariate setting and CNN for the multivariate setting, respectively. Notably, TSPulse, without any training on the target data, outperforms all models trained on it, underscoring its strong transfer learning. TSPulse also outperforms all the pre-trained models by +30% by using just a fraction of their model size. The fine-tuned variant, TSPulse (FT), further improves results, achieving 24% and 26% gains over SOTA on uni and multivariate benchmarks. These results underscore the effectiveness of TSPulse for diverse AD tasks.

## 4.2 CLASSIFICATION

**Setup:** We evaluate TSPulse results on 29 datasets from the UEA Multivariate Time Series Classification Archive (Bagnall et al., 2018). Dataset and hyperparameter details are provided in Appendix A.12. We compare against recent pre-trained models—VQShape (Wen et al., 2024), Moment (Goswami et al., 2024), and UniTS (Gao et al., 2024)—as well as strong data-specific baselines including T-

Rep (Fraikin et al., 2024), TS2Vec (Yue et al., 2022), T-Loss (Franceschi et al., 2019), TS-TCC (Eldele et al., 2021) and TNC (Tonekaboni et al., 2021).

**Results** Classification results fine-tuned/trained on the labeled data are reported in Figure 5. TSPulse achieves state-of-the-art accuracy, surpassing VQShape, UniTS, and Moment by 5–16%, while being drastically smaller (1M vs. 10–340M parameters). It also outperforms contrastive and supervised baselines by 5–12%, highlighting the effectiveness of TSPulse fine-tuning with TSLens.

## 4.3 IMPUTATION

**Setup:** We evaluate TSPulse on 6 LTSF benchmark datasets (Wu et al., 2021): ETTh1, ETTh2, ETTm1, ETTm2, Weather, and Electricity, under 4 mask ratios (12.5%, 25%, 37.5%, 50%) using *irregular hybrid masking* (a mix of block and point masks) to simulate real-world missingness.

**Results:** We first evaluate TSPulse in a fully Zero-Shot (ZS) setup, requiring no data-specific tuning. Among pre-trained baselines, MOMENT supports native zero-shot imputation, for UniTS (Gao et al., 2024) the pretrained model is prompt-tuned (PMT) with 10% data in multi-task setup. We also compare against statistical baselines in ZS setup. As shown in Figure 6, TSPulse (ZS) outperforms MOMENT by over 70% and UniTS by 50% despite its prompt-tuning. Compared to statistical interpolation methods, TSPulse shows 50%+ gains, highlighting the effectiveness of TSPulse in hybrid masking setup and robust zero-shot generalization. More details in Appendix A.13

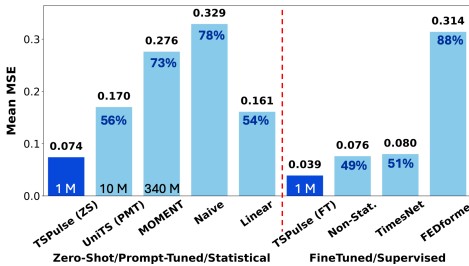

Figure 6: Hybrid Masking imputation MSE results (Lower is better). IMP(%)—the percentage improvement of TSPulse over baselines.

this setting as well.

When Fine-Tuning (FT) is desired, TSPulse can be extended with a channel-mixing decoder to capture inter-variable dependencies, outperforming strong supervised models like TimesNet (Wu et al., 2022), FedFormer (Zhou et al., 2022), and Non-Stationary Transformers (Non-Stat.) (Liu et al., 2022) by over 40%. Remarkably, TSPulse's zero-shot performance already exceeds many of the fine-tuned benchmarks, demonstrating its strong generalization and transferability. We further evaluated TSPulse under the full block masking strategy in both ZS and FT settings, as illustrated in Appendix Figure 13. TSPulse continues to outperform all baselines by a significant margin in

## 4.4 TIME-SERIES SIMILARITY SEARCH

**Setup:** We evaluate TSPulse's similarity search using its zero-shot semantic embeddings to retrieve time-series segments with similar patterns, even under real-world distortions like time shifts, magnitude changes, and noise. Since time-series are typically indexed via high-stride sliding

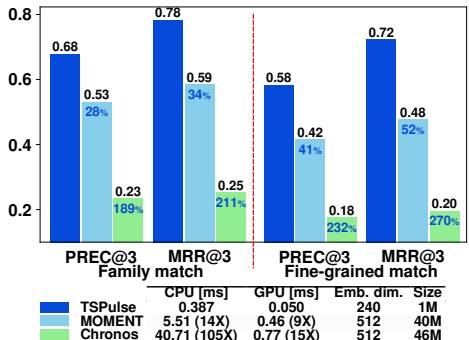

Figure 7: Similarity Search results using zero-shot embeddings (Higher is better). IMP(%)—the percentage improvement of TSPulse over baselines.

windows for efficient storage, the same pattern can appear in different positions, making distortion-invariant embeddings essential for similarity search. We use real and synthetic data for indexing, and generate query samples by applying complex augmentations (time shifts, magnitude changes, and noise distortions) from indexed samples.

This setup tests the embeddings' robustness in retrieving distorted similar patterns and simplifies evaluation, as the correct matches for each query are already known. Two tasks are defined: *Family Match* for high-level pattern retrieval and *Fine-Grained Match* for precise pattern matching and evaluated using PREC@k and MRR@k (Valcarce et al., 2020). We construct a synthetic dataset and a real dataset based on the UCR dataset (Chen et al., 2015), and report the average score across both for each task. See Appendix A.14 for full details.

| Variate | TSPulse Head | VUS-PR | |
|---|---|---|---|
| Uni. | $\text{Head}_{\text{triang.}}$ | **0.48** | |
| | $\text{Head}_{\text{ensemble}}$ | 0.44 | 9% ↓ |
| | $\text{Head}_{\text{time}}$ | 0.42 | 14% ↓ |
| | $\text{Head}_{\text{fft}}$ | 0.42 | 14% ↓ |
| | $\text{Head}_{\text{pred}}$ | 0.30 | 60% ↓ |
| Multi. | $\text{Head}_{\text{triang.}}$ | **0.36** | |
| | $\text{Head}_{\text{ensemble}}$ | 0.31 | 16% ↓ |
| | $\text{Head}_{\text{time}}$ | 0.31 | 16% ↓ |
| | $\text{Head}_{\text{fft}}$ | 0.31 | 16% ↓ |
| | $\text{Head}_{\text{pred}}$ | 0.24 | 50% ↓ |

(a) Zero-shot Anomaly Detection

| Model Variant | Accuracy | |
|---|---|---|
| **TSPulse** | **0.747** | |
| w/o Short Embedding | 0.689 | 8% ↓ |
| w/o Long Embedding | 0.681 | 10% ↓ |
| w/o Mask | 0.691 | 8% ↓ |
| w/o CM Identity Init | 0.685 | 9% ↓ |
| w/o Channel Expansion | 0.734 | 2% ↓ |
| w/o TSLens (Avg-Pool) | 0.675 | 11% ↓ |
| w/o TSLens (Max-Pool) | 0.645 | 16% ↓ |
| w/o Dual-space Learning | 0.696 | 7% ↓ |

(b) Classification Accuracy

| Model Variant | MSE | |
|---|---|---|
| **TSPulse** | **0.074** | |
| w/o Dual-Space | 0.081 | 8% ↓ |
| w/o Hybrid PT | 0.354 | 79% ↓ |

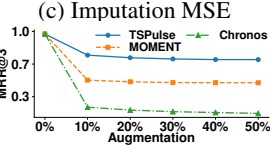

(c) Imputation MSE

(d) Similarity Search

Table 1: Ablation results across tasks. **[VUS-PR, Accuracy, MRR]**: Higher is better; **[MSE]**: Lower is better. IMP(%) indicates the percentage improvement of the bold variant over the compared variant.

**Results:** Figure 7 compares TSPulse's similarity search performance against zero-shot embeddings from MOMENT and Chronos. We use their smallest variants to closely match TSPulse's embedding size and enable faster indexing for a fair comparison. As shown, TSPulse outperforms MOMENT by over 25% in family-level and 40% in fine-grained match accuracy, and surpasses Chronos by 100%. Notably, TSPulse's zero-shot embeddings are 2X smaller and enable 10–100X faster CPU inference, 9–15X faster GPU inference, and come from a model that is 40× smaller than the baselines. Further discussion in Appendix A.14.

## 5 ABLATION STUDIES

- **Anomaly Detection (AD)** We evaluate the performance of individual TSPulse heads for anomaly detection. Table 1(a) reports average VUS-PR scores for each head used independently. The proposed multi-head triangulation outperforms all single-head variants on both TSB-AD-U and TSB-AD-M, demonstrating the strength of multi-head TSPulse. See Appendix A.11 for details.

- **Classification** We evaluate the impact of key design components in Table 1(b), using a representative subset of 17 UEA datasets for faster analysis. Removing either the short or long embedding from the *disentanglement* design reduces mean accuracy by 8–10%, confirming the importance of capturing both semantic and fine-grained features. Disabling *masking* during fine-tuning leads to an 8% drop—especially on smaller datasets—highlighting its role as a regularizer. Replacing *TSLens* with simple pooling causes an 11–16% drop, emphasizing the value of feature-attention. Randomly initializing the *channel-mixing (CM)* blocks instead of using identity weights leads to a 9% drop, reflecting the need for stable gradient flow. Removing *dual-space learning* (i.e., reconstructing only in time domain) lowers accuracy by 7%, and omitting *virtual channel expansion*, critical for low-channel datasets—causes a further 2% drop. More details in Appendix 18.

- **Imputation** Table 1(c) shows that removing dual-space learning leads to an 8% drop in zero-shot accuracy. When pre-training (PT) is done with only block masking (i.e., w/o Hybrid PT), performance drops by 79% under hybrid-mask eval settings, underscoring the importance of hybrid masking in pre-training for robust, generalizable imputation, where missingness is irregular and more reflective of real-world scenarios.

- **Similarity Search** Table 1(d) shows that TSPulse and baselines perform similarly without distortion. As augmentation distortion increases, all models degrade, but TSPulse remains notably more robust to time shifts, magnitude changes, and noise—highlighting the resilience of its embeddings in retrieving distorted yet similar patterns. Also, use of hybrid masking & semantic embedding boosts search performance by over 20% (Appendix A.14)

- **Efficiency** Appendix A.2 shows TSPulse is significantly faster, smaller and CPU-friendly.

## 6 SENSITIVITY ANALYSIS OF EMBEDDING DISENTANGLEMENT

To validate that TSPulse learns genuinely disentangled temporal, spectral, and semantic representations, we conduct controlled experiments on synthetic signals under three perturbation settings: missing data, additive noise, and phase/time shifts. These perturbations allow us to isolate how each

embedding type responds to missing data in the input (masking), stochastic changes (noise), and temporal misalignment (phase shift).

We quantify embedding stability using a distortion metric that measures how much each embedding changes under controlled perturbations (formal definitions in Appendix A.3). Representative results are summarised in Table 2. Lower values indicate greater robustness.

Table 2: Representative distortion results (in %). Lower indicates less distortion. Time and FFT embeddings have dimension $d = 1536$, while the semantic embedding has dimension $d = 256$.

| Experiment | Time ($d = 1536$) | FFT ($d = 1536$) | Semantic ($d = 256$) |
|---|---|---|---|
| 30% Missing Data | 8.3% | 27.4% | **4.6%** |
| Noise Level $\eta = 0.5$ | 2.7% | 6.8% | **2.5%** |
| Phase / Time Shift | 130% | 21% | **12%** |

The results exhibit clear and expected disentanglement patterns.

**Temporal embeddings** are highly sensitive to phase/time shifts (130% distortion), confirming preservation of fine-grained temporal alignment. This property is critical for tasks that depend on precise timing cues.

**FFT embeddings** demonstrate substantially lower phase sensitivity, reflecting invariance to temporal alignment while retaining spectral characteristics.

**Semantic embeddings (a.k.a Register embeddings)** are the most robust to missing data and noise and the least sensitive to phase shifts, consistent with their role as high-level structural abstractions rather than fine-grained signal encoders.

These complementary behaviours directly translate to downstream utility. Tasks such as anomaly detection and imputation benefit from time and FFT embeddings for high-fidelity reconstruction, while retrieval tasks primarily leverage semantic embeddings for compact summarisation. Moreover, disentangled reconstruction across spaces enables triangulation-based mechanisms that detect anomaly types missed by single-view models.

For more details, refer to Appendix A.3. We have also conducted a sensitivity deep-dive analysis focused on register embeddings as explained in Appendix A.4. Together, these results confirm that TSPulse achieves effective disentanglement across both representational spaces (time vs. FFT) and abstraction levels (fine-grained vs. semantic), forming a core foundation for its strong zero-shot transfer performance.

## 7 CONCLUSION

TSPulse sets a new benchmark for ultra-compact time-series pre-trained models, achieving state-of-the-art performance in classification, imputation, anomaly detection, and similarity search—all with under 1M parameters. Powered by innovations like disentangled masked reconstruction across spaces and abstractions, TSLens, hybrid masking, and multi-head triangulation, TSPulse enables robust zero-shot and fine-tuned performance. Despite its small size, it outperforms models 10–100X larger and runs efficiently on CPUs, making it both powerful and deployment-ready. Appendix A.17 outlines limitations and future directions, including opportunities to expand to additional downstream tasks, enable incremental learning and reduce supervision requirements. We believe this work will inspire more advanced research and innovation in the field of lightweight time-series modeling.

## REPRODUCIBILITY STATEMENT

Models and source code are publicly available at `https://huggingface.co/ibm-granite/granite-timeseries-tspulse-r1`. Detailed model parameters are included in Appendix A.9. All pretraining datasets are publicly available and referenced in Appendix A.8, and all evaluation datasets are likewise publicly accessible.

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

# A   APPENDIX

## A.1   USE OF LARGE LANGUAGE MODELS (LLMS)

LLMs were used only as a writing aid to polish phrasing and correct grammatical errors in the manuscript.

## A.2   RUNTIME COMPUTATIONAL ANALYSIS OF TSPULSE AND OTHER PRE-TRAINED MODELS

In this section we do the computational analysis of TSPulse with all the other pre-trained models that are used for benchmarking across different tasks in this work. These models include MOMENT (Goswami et al., 2024), UniTS (Gao et al., 2024), VQShape (Wen et al., 2024), Chronos (Ansari et al., 2024), TimesFM (Das et al., 2023) and Lag-llama (Rasul et al., 2023).

In this analysis, we demonstrate the suitability of TSPulse for rapid time-series analysis by highlighting its computational advantages over larger pre-trained models. With only 1M parameters, TSPulse offers significant efficiency benefits in terms of per-batch CPU inference time, GPU inference time, and peak GPU memory usage. These comparisons are made against all pre-trained baselines discussed above. Refer to Table 3 for detailed results.

All computational analysis experiments were conducted on a system equipped with an NVIDIA A100 GPU (80GB), 16 CPU cores, and 256GB of RAM. For inference time and peak memory usage measurements, a zero tensor input of shape (batch size = 32, sequence length = 512, channels = 5) was used.

## A.3   SENSITIVITY ANALYSIS ACROSS DIFFERENT TSPULSE EMBEDDINGS

We performed detailed control experiments on a synthetic dataset to investigate disentanglement properties in the **TSPulse** embedding space.

| Model | Params (**M**) | GPU Inference Time (**ms**) | CPU Inference Time (**s**) | Max. Memory (**GB**) |
|---|---|---|---|---|
| **TSPulse** | **1.06** | **7.16** | **0.06** | **0.39** |
| GPT4TS/OFA | 82 (77X) | 89 (12.43X) | 2.23 (37.16X) | 5.42 (13.89X) |
| VQShape | 37.09 (35.0X) | 65.37 (9.1X) | 1.15 (19.2X) | 6.88 (17.6X) |
| Lag-llama | 2.45 (2.3X) | 236.00 (33.0X) | 0.24 (4.0X) | 1.01 (2.6X) |
| TimesFM | 203.57 (192.0X) | 101.80 (14.2X) | 1.14 (19.0X) | 0.87 (2.2X) |
| MOMENT (small) | 35.34 (33.3X) | 32.57 (4.5X) | 2.74 (45.7X) | 0.56 (1.4X) |
| MOMENT (base) | 109.64 (103.4X) | 120.27 (16.8X) | 7.60 (126.7X) | 1.15 (2.9X) |
| MOMENT (large) | 341.24 (321.9X) | 405.42 (56.6X) | 21.98 (366.3X) | 2.30 (5.9X) |
| UniTS | 3.5 (3.24X) | 7.94 (1.11X) | 1.50 (25X) | 0.57 (1.4X) |
| CHRONOS (tiny) | 8.39 (7.9X) | 39.81 (5.6X) | 66.15 (1102.5X) | 2.91 (7.5X) |
| CHRONOS (small) | 46.15 (43.5X) | 112.62 (15.7X) | 201.62 (3360.3X) | 5.86 (15.0X) |
| CHRONOS (base) | 201.37 (190.0X) | 328.37 (45.9X) | 642.38 (10706.3X) | 8.97 (23.0X) |
| CHRONOS (large) | 708.96 (668.8X) | 897.86 (125.4X) | 1754.87 (29247.8X) | 12.80 (32.8X) |

Table 3: **Computational Analysis** of TSPulse and other pre-trained models. Per batch GPU inference time in milli-seconds(ms), CPU inference time in seconds(s), Maximum memory usage in GB and Number of parameters in millions(M) are reported. Lower is better.

### A.3.1   DATASET

The dataset consists of three distinct signals, each constructed by scaling and exponentiating a pure sine wave to obtain different periodic shapes (see Appendix Figure 10). White noise was added at a specified signal-to-noise ratio (SNR) to generate the final dataset.

### A.3.2   EXPERIMENTAL SETUP

We designed three controlled experiments to analyse the effect of missing data, noise, and phase shift on the learned embeddings.

**Missing Data.**   TSPulse supports user-defined masks during inference. We leverage this capability to emulate missing observations. From the synthetic dataset described above, we randomly sampled 2,000 instances and extracted embeddings under different user-defined masks. The proportion of zeros in each mask corresponds to the fraction of missing data.

**Noise.**   The synthetic generator allows precise control over noise levels. We created multiple groups of samples from the same base signal while varying only the noise level. Within each group, indices are aligned such that the same index corresponds to the same underlying base signal across all noise conditions.

**Phase / Time Shift.**   We utilise a non-linearly scaled sine wave as the base signal to explicitly control frequency and phase. A base dataset is generated by sampling the signal at fixed frequency

and waveform type. Across subsequent groups, only the phase is varied in discrete steps from $0$ to $\pi$, while keeping all other parameters constant.

### A.3.3 EXPECTED BEHAVIOUR OF A GOOD SEMANTIC EMBEDDING

**Missing Data Robustness.** An embedding should exhibit minimal sensitivity to missing data. For samples masked with a fixed ratio, their neighbourhood relationships should remain preserved.

**Noise Robustness.** Embeddings should be robust to noise, such that similarity in embedding space reflects similarity of the underlying signal rather than noise perturbations.

**Phase Sensitivity.** Temporal embeddings should be sensitive to phase/time shifts. In contrast, spectral (FFT) and semantic embeddings should exhibit minimal sensitivity to phase variations.

### A.3.4 EVALUATION METRICS

**Distortion.** We measure distortion as the $\ell_2$ norm ratio of embedding changes induced by missing data, noise, or phase shifts relative to the base embedding. Lower distortion indicates greater robustness.

We slightly refine the distortion definition to align with the expected behaviour described above.

### A.3.5 NOTATION

- $\mathcal{E}(x, \mathbb{M})$: Embedding of dataset $x$ under user-defined mask $\mathbb{M}$.
- $\mathcal{E}(x)$: Embedding without mask.
- $x(\eta)$: Data instance with noise level $\eta$, where $x(0)$ denotes the pure signal.
- $\Phi(x)$: Phase associated with signal $x$.

### A.3.6 DISTORTION MEASURES

**Mask-Induced Distortion.** For a given mask $\mathbb{M}$:

$$\delta_m(\mathbb{M}) = \mathop{\mathbb{E}}_{(x,y)\in\{x\neq y\}} \left[ \left| 1 - \frac{\|\mathcal{E}(x;\mathbb{M}) - \mathcal{E}(y;\mathbb{M})\|}{\|\mathcal{E}(x) - \mathcal{E}(y)\|} \right| \right]. \tag{1}$$

**Noise-Induced Distortion.** For a given noise level $\eta$:

$$\delta_e(\eta) = \mathop{\mathbb{E}}_{x} \left[ \left| \frac{\|\mathcal{E}(x(\eta))\|}{\|\mathcal{E}(x(0))\|} - 1 \right| \right]. \tag{2}$$

**Phase-Induced Distortion.** For a phase difference $\phi$:

$$\delta_p(\phi) = \mathop{\mathbb{E}}_{(x,y)\in\{|\Phi(x)-\Phi(y)|=\phi\}} \left[ \frac{\|\mathcal{E}(x) - \mathcal{E}(y)\|}{\min\left(\|\mathcal{E}(x)\|, \|\mathcal{E}(y)\|\right)} \right]. \tag{3}$$

### A.3.7 REPRESENTATIVE RESULTS (LOWER IS BETTER)

Table 4: Representative distortion results across missing data, noise, and phase shift experiments (in %). Lower is better.

| Experiment | Time (1536) | FFT (1536) | Register (256) |
|---|---|---|---|
| 30% Missing Data | 8.3% | 27.4% | 4.6% |
| Noise Level $\eta = 0.5$ | 2.7% | 6.8% | 2.5% |
| Phase / Time Shift | 130% | 21% | 12% |

Table 4 summarises the key findings across the three controlled perturbation settings.

Time embeddings (dimension 1536) exhibit very high distortion under phase/time shifts (130%), indicating strong temporal sensitivity. This confirms that temporal information is preserved — a critical property for tasks that rely on timing cues, which are typically lost in purely spectral representations.

FFT embeddings (dimension 1536) show substantially lower distortion under phase shifts (21%), reflecting their ability to capture frequency characteristics while being relatively invariant to temporal alignment.

Compact semantic (register) embeddings (dimension 256) demonstrate the lowest distortion under phase shifts (12%), as well as the strongest robustness to missing data and noise. This behaviour aligns with their intended role of encoding high-level structural abstractions. Additional sensitivity analyses and PCA visualisations are provided in Appendix A.4.

Each embedding type therefore provides complementary strengths. Some datasets rely primarily on raw temporal dynamics, others on frequency characteristics, and some on high-level semantic summaries. **TSLens** attends jointly to all three embeddings, allowing the model to automatically emphasise the most informative view for a given dataset.

Anomaly detection and imputation benefit primarily from time and FFT embeddings, which preserve fine-grained reconstruction details. Disentangled reconstruction across spaces enables triangulation, improving detection of anomaly types that may be missed by single-view models. In contrast, retrieval tasks benefit mainly from semantic embeddings, which provide compact summary-level representations.

By explicitly exposing disentangled embeddings, **TSPulse** enables flexible and effective transfer across diverse zero-shot tasks. The framework introduces disentanglement across both representational spaces (time vs. FFT) and abstraction levels (fine-grained vs. semantic), leading to double-digit performance gains across multiple benchmarks.

### A.3.8 DETAILED RESULTS FOR EACH EXPERIMENT

All distortion values are reported in %. Lower is better.

Table 5: Distortion under varying levels of missing data (in %).

| Embedding | 11% | 19% | 30% | 36% | 70% |
|---|---|---|---|---|---|
| Time | 3.8% | 5.6% | 8.3% | 9.4% | 23.1% |
| FFT | 11.8% | 18.9% | 27.4% | 32.6% | 61.8% |
| Register | 1.6% | 2.8% | 4.6% | 5.7% | 11.8% |

As shown in Table 5, semantic embeddings consistently exhibit the lowest distortion across all masking levels, demonstrating strong robustness to missing observations. FFT embeddings are most affected, indicating higher sensitivity to partial signal removal.

Table 6: Distortion under varying noise levels $\eta$ (in %).

| Embedding | $\eta = 0.1$ | $\eta = 0.25$ | $\eta = 0.5$ | $\eta = 0.75$ |
|---|---|---|---|---|
| Time | 1% | 1.4% | 2.7% | 4.3% |
| FFT | 2.4% | 4% | 6.8% | 11.7% |
| Register | 0.8% | 1.3% | 2.5% | 4.3% |

Table 6 further confirms that semantic embeddings maintain the strongest robustness to additive noise, while FFT embeddings are more sensitive to spectral perturbations.

Finally, Table 7 highlights the clear separation between representational roles: time embeddings are highly phase-sensitive, while FFT and semantic embeddings remain comparatively invariant. This confirms that disentanglement across representational spaces has been successfully achieved.

Table 7: Distortion under varying phase shifts $\Phi$ (in %).

| Embedding | $\Phi = \frac{\pi}{3}$ | $\Phi = \frac{2\pi}{3}$ | $\Phi = \pi$ |
|---|---|---|---|
| Time | 169% | 163% | 76% |
| FFT | 24% | 23% | 19.6% |
| Register | 13.6% | 13.7% | 10.5% |

## A.4 SENSITIVITY ANALYSIS OF TSPULSE SEMANTIC EMBEDDINGS

In this section, we study the behavior of the short register embeddings—i.e., zero-shot semantic embeddings—learned by *TSPulse*. These embeddings are designed to capture compact, meaningful representations of time-series patterns and are primarily used for retrieval and semantic similarity tasks. We perform a detailed sensitivity analysis to understand how these embeddings respond when various properties of the input data are altered, including magnitude changes, added noise, missing values, time-shifts (i.e., phase variations), and changes in frequency and patterns.

Ideally, we want the embeddings to remain invariant to factors such as noise, magnitude scaling, missing values, and temporal misalignments, while being highly responsive to core semantic characteristics like frequency patterns (e.g., periodicity, seasonality) and distinctive shape structures in the data. Such behavior ensures robust semantic retrieval—retrieving similar patterns despite distortions—while preserving the ability to distinguish fundamentally different signal classes.

**Sensitivity to magnitude scaling** To assess whether the register embeddings capture only semantically meaningful variations, we analyze their sensitivity to magnitude/amplitude scaling and compare it against their responses to frequency shifts. Frequency shift directly influences the structural semantics of time-series signals—such as periodicity—whereas magnitude scale changes are often irrelevant. This contrast allows us to quantify and appreciate the embedding's robustness to irrelevant perturbations. We generate a synthetic dataset using sinusoidal signals with three distinct frequencies, and we apply a linear amplitude scaling factor $c$, sampled uniformly from the range $[1, 25]$. We then extract register embeddings for each scaled variant and analyze their variation in the embedding space.

Our results in Figure 8a show that while register embeddings vary noticeably across different frequencies, they remain very stable under magnitude scaling. For this analysis, we extract the *TSPulse* register embedding for each data point, along with its associated scaling factor and frequency label. PCA projections reveal that the intra-cluster variation caused by amplitude scaling is small compared to the inter-cluster separation induced by frequency differences. This indicates that the embeddings are largely invariant to amplitude, an essential property for semantic retrieval tasks, where magnitude often varies due to normalization, sensor artifacts, or contextual shifts. By maintaining sensitivity to core signal properties such as frequency while suppressing irrelevant variations like scale, the model yields robust and meaningful representations suitable for retrieval.

**Sensitivity to noise** We evaluate the impact of additive noise on the register embeddings by analyzing how noise-induced variations compare to those caused by changes in frequency. Synthetic sine wave signals are generated with distinct frequency groups, and Gaussian noise is added to simulate varying signal-to-noise ratios. Embeddings are extracted for each noisy signal, and their spatial variation is analyzed using intra-group and inter-group distances. Intra-group variation reflects sensitivity to noise, while inter-group separation captures the embedding's response to frequency changes. As shown in Figures 8b, the impact of noise on the embeddings is minimal compared to the clear separation observed across different frequencies. This highlights the robustness of *TSPulse* embeddings to noise—an essential property for maintaining semantic consistency in real-world applications where signal degradation to noise is common.

**Sensitivity to time shifts (phase variations)** To assess the sensitivity of register embeddings to phase (i.e. time-shift) variations, we conduct a control experiment using synthetic signals of the form $\sin(f\pi x + c) + \epsilon$, where $f \in \{3, 3.2, 3.5\}$ represents the frequency, $c$ denotes the phase shift (i.e., time-shift), and $\epsilon$ is Gaussian noise. We change the phase $c$ continuously from $-\pi$ to $\pi$ to simulate temporal misalignments commonly seen in real-world indexed data.

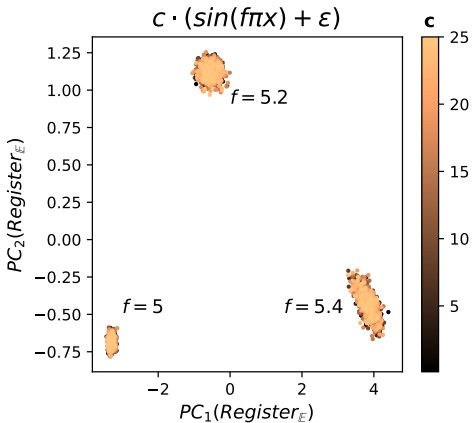 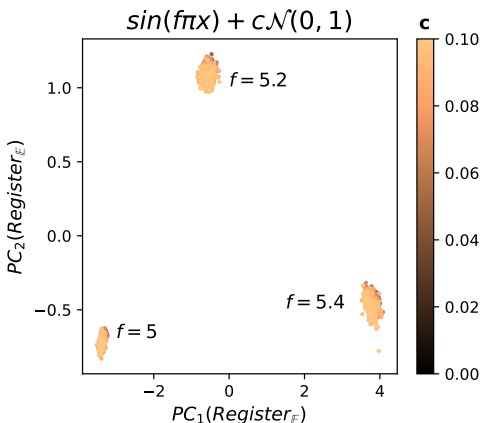

(a) **Sensitivity to Magnitude Scaling:** PCA of register embeddings with three distinct frequency groups, colored by amplitude scale factor ($c$). The embeddings remain compact across scale variations.

(b) **Sensitivity to Noise:** PCA of register embeddings for noisy sine waves with three distinct frequency groups, colored by noise factor ($c$). Embedding variation due to noise is small compared to frequency-induced separation.

Figure 8: Scatter plots of the first two PCA components of *TSPulse* register embeddings under (a) amplitude scaling and (b) additive Gaussian noise. Each setup contains three distinct frequency-based groups ($f = \{5, 5.2, 5.4\}$). The colour in the scatter plot indicates the value of the variable $c$ as indicated by the corresponding equation.

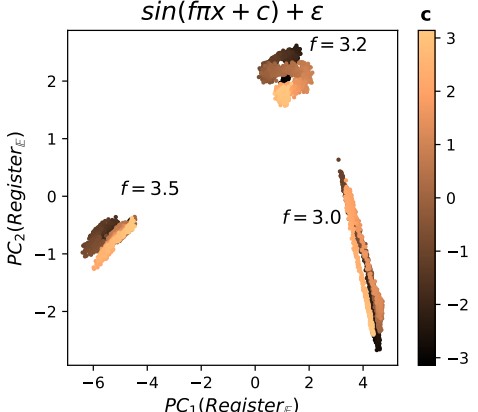 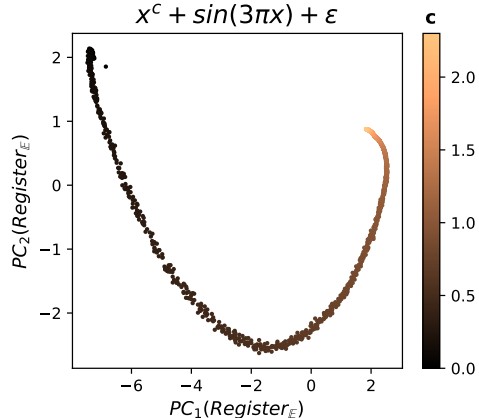

(a) **Sensitivity to Phase Shifts:** PCA projections of the register embeddings of phase shifted instances with three distinct frequencies.

(b) **Sensitivity to Trends:** PCA projections of the register embeddings of the data with varying trend.

Figure 9: Data patterns in register embedding space. The colour in the scatter plot indicates the value of the variable $c$ as indicated by the corresponding equation. The control experiment reveals, register embeddings are strongly sensitive to data trend and frequency and relatively less sensitive to phase (time shift).

This setup emulates practical scenarios in semantic time-series search, where the same underlying pattern may be indexed at different time offsets in a database. In such cases, time shift invariant embeddings enable robust retrieval, ensuring semantically similar signals are matched even with minor time shifts. Conversely, sensitivity to frequency is desirable, as frequency often encodes core data characteristics such as seasonality or periodic structure. Thus, being able to distinguish small frequency differences allows the model to separate fundamentally different patterns more effectively.

PCA on register embeddings shows variations in frequency lead to significantly larger shifts in the embedding space compared to phase changes. The first two PCA components reveal tight, well-separated clusters corresponding to each frequency, while phase shifts produce only minor within-cluster variation (Figure 9a). This confirms that the embeddings are relatively less sensitive to phase shifts while remaining distinctly responsive to frequency variations – an ideal characteristic for robust semantic search.

**Sensitivity to data trend**    Here, we investigate how the register embedding models data trends. To carry out the control experiment with synthetic data with functional form $x^c + \sin(3\pi x) + \epsilon$. By varying the variable $c$, we generate different monotonic trend patterns in the data. We continuously varied the data trend parameter ($c$) from 0 to 2.5. Where 0 means no data trend, while 2.0 represents a quadratic pattern in the input data. The PCA analysis of the register embeddings shows that it distinctly models the monotonic data trend.

**Sensitivity to shape space**    In these experiments, we analyse the sensitivity of the embedding space to changes in the function shape, compared to frequency and phase change. For these experiment we use three distinct data generating functions, of the form $\mathbb{F}_1(x) = \sin(x)$, $\mathbb{F}_2 = 2*\left(\sin(x)+1\right)^4 - 1$, and $\mathbb{F}_3 = 2 * \sqrt[4]{\sin(x)+1} - 1$ (Figure 10). Each of these functions is periodic and trigonometric. We generate a dataset using these three generating functions. For each generating function, we sample data with three distinct frequency groups $f = \{5, 5.2, 5.4\}$, for each frequency we generate synthetic data by randomly sampling phase between $(-\pi, \pi)$. We carry out PCA analysis on the register embedding of this dataset (Figure 11). The analysis reveals the frequency groups form distinct clusters. Frequency clusters from same generating function are closer compared to clusters from different generating function.

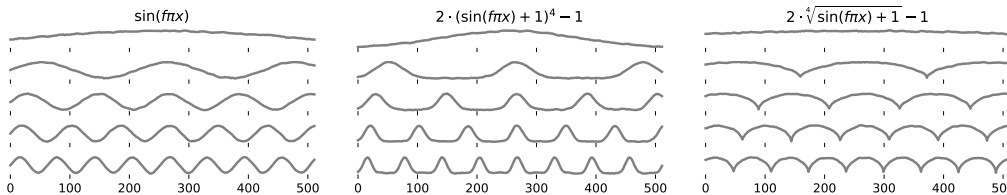

Figure 10: Data generated from the three synthetic data generator functions with varying frequencies.

**Sensitivity to missing values**    To assess the robustness of register embeddings to missing data, we generate synthetic signals using the function $\sin(c\pi x)$, where the frequency parameter $c$ is uniformly sampled from the range $[1, 15]$. Missing values are simulated by applying block masks of fixed patch length ($pl = 8$) to the input sequence, and we vary the mask ratio to control the extent of missingness.

We extract register embeddings under different masking levels and project them using PCA. As shown in Figure 12, the embedding space remains highly stable and preserves the underlying frequency-based structure even when up to 35% of the input data missingness. Distortion in the embedding manifold is observed only at very high mask ratios, where a majority of the signal is occluded. These results demonstrate that *TSPulse* embeddings are very robust to moderate levels of missing data—a valuable property in real-world time-series applications where sensor dropouts and irregular sampling are common.

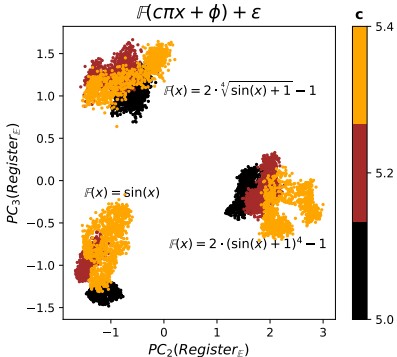

Figure 11: **Sensitivity to shape:** Cluster analysis of the register embedding of the data with three distinct function class. The cluster corresponding to the respective generator function is annotated in the figure. The colour represents the frequency associated with the data point ($c$). The strong separation between the embeddings of the data instances from different generator functions indicates that register embedding distinguishes the data instances by their shapes.

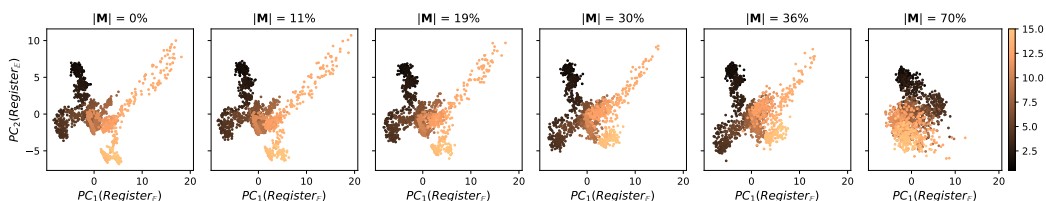

Figure 12: **Sensitivity to missing values:** First two principal components computed of the register embeddings, with varying mask ratio applied to the input data. The colour indicates the frequency of the synthetic data point. The plot indicates register embedding is fairly robust to missing data.

## A.5 NOTATION DETAILS

In this section, we detail the notations used in the architecture for clarity. Table 8 has the notation summary.

| Symbol | Description |
|---|---|
| $S$ | Sequence length |
| $C$ | Number of input channels |
| $pl$ | Patch length |
| $D$ | Feature dimension |
| $\mathbf{M} \in \mathbb{R}^{1 \times pl}$ | Mask token |
| $\mathbf{X} \in \mathbb{R}^{S \times C}$ | Multivariate time series input |
| $\hat{\mathbf{X}} \in \mathbb{R}^{S \times C}$ | Masked time series |
| $\mathbf{X}_m \in \mathbb{R}^{S \times C}$ | Masked and scaled input time series |
| $\mathbf{X}'_m \in \mathbb{R}^{C \times N \times pl}$ | Transposed and patched $\mathbf{X}_m$ |
| $\mathbf{X}^f \in \mathbb{R}^{S \times C}$ | Normalized frequency domain ground truth signal |
| $\mathbf{X}^f_{\text{sign}} \in \mathbb{R}^{S/2 \times C}$ | Normalized frequency probability ground truth signal |
| $\mathbf{X}^f_m \in \mathbb{R}^{S \times C}$ | Normalized frequency domain masked signal |
| $\mathbf{X}'^f_m \in \mathbb{R}^{C \times N \times pl}$ | Transposed and patched $\mathbf{X}^f_m$ |
| $\mathbf{X}_{\text{pred}} \in \mathbb{R}^{F \times C}$ | Next-Point Ground Truth of length $F$ |
| $\mathbf{Time}_E \in \mathbb{R}^{C \times N \times D}$ | Time-encoded features |
| $\mathbf{FFT}_E \in \mathbb{R}^{C \times N \times D}$ | Frequency-encoded features |
| $\mathbf{Reg}_E \in \mathbb{R}^{C \times R \times D}$ | Learnable register tokens |
| $\mathbf{Input}_E = [\mathbf{Time}_E; \mathbf{FFT}_E; \mathbf{Reg}_E] \in \mathbb{R}^{C \times (2N+R) \times D}$ | Concatenated input to the backbone |
| $\mathbf{Backbone}_E \in \mathbb{R}^{C \times (2N+R) \times D}$ | Output embedding from the backbone |
| $\mathbf{Decoder}_E \in \mathbb{R}^{C \times (2N+R) \times D}$ | Output embeddings from the decoder |
| $\mathbf{Y} \in \mathbb{R}^{S \times C}$ | Time domain signal reconstruction |
| $\mathbf{Y}_f \in \mathbb{R}^{S \times C}$ | Frequency domain signal reconstruction |
| $\mathbf{Y}' \in \mathbb{R}^{S \times C}$ | Time domain signal reconstructed from frequency embedding |
| $\mathbf{Y}^f_{\text{sign}} \in \mathbb{R}^{S/2 \times C}$ | Predicted frequency probability |
| $\mathbf{Y}_{\text{pred}} \in \mathbb{R}^{F \times C}$ | Predicted Next Points of length $F$ |

Table 8: Notation Summary

## A.6 RELATED ART

### A.6.1 DATASET-SPECIFIC NEURAL ARCHITECTURES

**Neural Architectures for Time-Series Forecasting Tasks**  Over the past few years, time-series forecasting has benefited significantly from advances in neural architectures, particularly with the adaptation of models from NLP and vision to temporal data. Early breakthroughs came from transformer-based designs such as Informer (Zhou et al., 2021), Autoformer (Wu et al., 2021), and FEDFormer (Zhou et al., 2022), which extended attention mechanisms to model long-range dependencies and seasonal-trend decomposition in time-series. Despite their success, these models were soon challenged by simpler alternatives. DLinear (Zeng et al., 2022) revealed that with basic principles like per-channel modeling and time-series decomposition, a linear model could match or surpass many transformer-based approaches. This raised important questions about the complexity-performance tradeoff in time-series modeling. A key turning point was the introduction of PatchTST (Nie et al., 2023), which demonstrated that transformers could be highly effective when embeddings are retrieved at the patch level instead of the time-point level. This patching strategy not only preserved local semantics and improved scalability, but also re-established the superiority of transformers across multiple benchmarks.

To overcome the computational burden of transformers, new lightweight models began to emerge. TSMixer (Ekambaram et al., 2023), built upon the MLP-Mixer (Tolstikhin et al., 2021) paradigm,

replaced attention with feedforward mixing operations, achieving competitive performance with significantly reduced memory and latency. Similarly, iTransformer (Liu et al., 2024) introduced attention mechanisms over the variate (feature) dimension rather than the temporal axis, enabling better generalization across multivariate inputs. TimeMixer (Wang et al., 2024) leveraged multi-scale mixer blocks to simultaneously capture short- and long-term dynamics, while TimesNet (Wu et al., 2022) incorporated an inception-style architecture to disentangle intra- and inter-period patterns, enhancing its ability to model complex seasonal behaviors.

These developments highlight a growing diversity in time-series modeling strategies—from heavy transformers to fast feedforward mixers—each offering a different balance of accuracy, interpretability, and efficiency. However, these innovations have predominantly focused on time series forecasting tasks.

**Unsupervised Representation Learning for Time-Series Diagnostic Tasks**   Learning meaningful representations from unlabeled time-series is a foundational step for many downstream diagnostic tasks, including classification, retrieval, imputation, and anomaly detection. Contrastive learning has emerged as a dominant strategy in this space. Temporal Neighborhood Coding (TNC) (Tonekaboni et al., 2021) introduced a local context-aware contrastive learning framework that defines neighborhoods over smoothed signals, pushing representations of nearby time segments closer while distinguishing distant ones. TS2Vec (Yue et al., 2022) extended this idea by contrasting hierarchical segments both temporally and across instances, achieving robust scale-invariant embeddings. T-Rep (Fraikin et al., 2024) focused on interpretability by encoding temporal attributes like trends and periodicity explicitly in the representation. BTSF (Yang & Hong, 2022) proposed a joint time-frequency contrastive approach, fusing insights from both domains to learn rich, discriminative embeddings. TimeDRL (Chang et al., 2024) proposed a disentangled representation learning framework that separately captures timestamp-level and instance-level embeddings, yielding improved performance across five classification and six forecasting datasets. TF-C (Zhang et al., 2022) introduced a method that encourages entangled representations by aligning temporal and spectral embeddings within the representation space. TF-C was evaluated under both one-to-one and one-to-many transfer learning settings. While these methods demonstrate strong performance, they typically rely on compact architectures trained individually per dataset. Consequently, their scalability—particularly in scenarios involving pretraining on large corpora and evaluation across extensive test sets—remains unestablished. Moreover, contrastive approaches, though effective, face inherent scalability limitations due to their reliance on contrastive sampling. The quadratic growth in pairwise comparisons not only increases computational overhead but also hinders generalization across diverse domains.

### A.6.2   TIME-SERIES PRE-TRAINED MODELS

Inspired by advances in pre-trained models from language and vision domains, the time-series community has begun exploring large-scale pre-trained models capable of generalizing across datasets and tasks. These time-series pre-trained models are typically trained using self-supervised objectives such as masked reconstruction (particularly suited for TS diagnostic tasks) or forecasting on millions of sequences (more suited for TS forecasting tasks).

**Transformer-Based TS Pre-trained Models**   Several high-capacity models have been introduced with a focus on either task specialization or general-purpose transfer. In task-specialized models for forecasting, Chronos (Ansari et al., 2024), TimesFM (Das et al., 2023), and Moirai (Woo et al., 2024) represent key milestones. Chronos adopts a T5-style encoder-decoder trained on discretized sequences for forecasting, while TimesFM uses an autoregressive decoder with fixed-length patch tokens. Moirai leverages an encoder-only design with multi-scale patches and outputs probabilistic distributions, supporting multivariate forecasting with high fidelity. In task-specialized models for classification, VQShape (Wen et al., 2024) offers state-of-the-art results with interpretabilty as they convert time-series in to sequence of meaningful and interpretable tokens. Generalist models such as Moment (Goswami et al., 2024), GPT4TS (Zhou et al., 2023), and UniTS (Gao et al., 2024) expand support to a wider set of tasks like classification, anomaly detection, and imputation. Moment employs masked reconstruction using a T5 encoder, while GPT4TS reuses a frozen GPT2 backbone with task-specific heads. UniTS introduces prompt and task tokens and replaces attention with DyLinear layers (dynamic operators) for efficient inference.

**Non-Transformer TS Pre-trained Models**   In parallel, compact TS pre-trained models have emerged to challenge the notion that larger is always better. TinyTimeMixer (TTM) (Ekambaram et al., 2024) exemplifies this category by using TSMixer architecture trained on forecasting objectives. Despite having a minimal 1-5 M parameters, it delivers competitive zero/few-shot forecasting, showcasing the potential of compact FMs. However, TTM is primarily tailored for forecasting, and there is growing interest in extending this compact modeling paradigm to a broader range of downstream diagnostic tasks—including anomaly detection, classification, and imputation—where efficiency remains an open challenge.

In summary, while contrastive and supervised representation learning methods are typically compact and effective, they are often trained separately on each dataset and lack transferability across domains. In contrast, TS pre-trained models offer strong generalization by pre-training on large, diverse corpora—but at the cost of significantly higher model complexity and computational demands. The ongoing challenge is to combine the generalization of pre-trained models with the compactness and efficiency of specialized architectures—an area where models like TSPulse seek to make impactful strides.

## A.7   BASELINE METHODOLOGIES

Refer to Table 9 for an overview of the methodologies used to report the baseline results. We either report the results directly from the associated papers or run their official/popular implementation to get the results.

| Task | Model | Results Derivation | Reference (Code/Paper) |
|---|---|---|---|
| Anomaly Detection | All SOTA Models | Reported in (Liu & Paparrizos, 2024) | TSB-AD-Leaderbaord |
| Classification | MOMENT | Reported in (Goswami et al., 2024) | MOMENT |
| | VQShape | Official Implementation | VQShape |
| | UniTS | Official Implementation | UniTS |
| | T-Rep | Reported in (Fraikin et al., 2024) | T-Rep |
| | TS2Vec | Reported in (Yue et al., 2022) | TS2Vec |
| | TNC | Reported in (Yue et al., 2022) | TS2Vec |
| | TS-TCC | Reported in (Yue et al., 2022) | TS2Vec |
| | T-Loss | Reported in (Franceschi et al., 2019) | T-Loss |
| Imputation | MOMENT | Official Implementation | MOMENT |
| | UniTS | Official Implementation | UniTS |
| | Statistical | Implementation | MOMENT |
| | TimesNet | Official Implementation | Time-Series-Library |
| | FEDformer | Implementation | Time-Series-Library |
| | Non-Stat. Transformer | Implementation | Time-Series-Library |
| Search | MOMENT | Official Implementation | MOMENT |
| | Chronos | Official Implementation | Chronos |

Table 9: Summary of Baselines used in this work across different tasks

## A.8   PRE-TRAINING DATASETS

Pre-training utilizes a subset of $\sim$ 1B time points drawn from the Monash (Godahewa et al., 2021) and LibCity (Wang et al., 2021; Woo et al., 2024) data collections. We leverage the same pre-training data as used in (Ekambaram et al., 2024) for training our models. For Monash, we select a subset of datasets from the Monash Time Series Forecasting Repository (Godahewa et al., 2021). For LibCity, we incorporated the data from the source locations. The complete list of datasets employed for pre-training is provided in Table 10. All datasets used are selected under permissive licenses that allow both open-source and commercial use.

## A.9   MODEL DETAILS

The default pre-training configuration for TSPulse is as follows: context length ($S$) = 512, patch length ($pl$) = 8, stride = $pl$, hidden feature size $D = 3 \times pl$, dropout = 0.2, head dropout = 0.2, number of backbone layers ($L$) = 8, number of decoder layers = 2, masking strategy = hybrid masking, training epochs = 20, and gated attention activation function = softmax. During pre-training, the backbone and the decoder are both operated in channel-independent approach, i.e., channel-mixing is not activated in the TSMixer modules.

| Origin | Data Source | Temporal Granularity |
|---|---|---|
| **Monash** | wind 4 sec dataset + **DRS** | 4-sec, 10-min, 15-min, 30-min, Hourly |
| | wind farms minutely dataset (no missing) + **DRS** | Min, 10-min, 15-min, 30-min, Hourly |
| | us births dataset | Daily |
| | solar 10 minutes dataset + **DRS** | 10-min, 30-min, Hourly |
| | australian electricity demand dataset + **DRS** | 30-min, Hourly, Daily |
| | kaggle web traffic dataset (no missing) | Daily |
| | nn5 daily dataset (no missing) | Daily |
| | solar 4 sec dataset + **DRS** | 4-sec, 10-min, 15-min, 30-min, Hourly |
| | saugeenday dataset | Daily |
| | sunspot dataset (no missing) | Daily |
| | kdd cup 2018 dataset (no missing) | Hourly |
| | bitcoin dataset (no missing) | Daily |
| | london smart meters dataset (no missing) + **DRS** | 30-min, Hourly, Daily |
| | australian weather dataset | Daily |
| **LibCity** | PEMS [03,04,07,08] + **DRS** | 5-min, 10-min, 15-min, 30-min, Hourly |
| | PEMS BAY + **DRS** | 5-min, 10-min, 15-min, 30-min, Hourly |
| | LOS LOOP + **DRS** | 5-min, 10-min, 15-min, 30-min, Hourly |

Table 10: **Pre-training Datasets for TSPulse.** We leverage the same pre-training data as used in Ekambaram et al. (2024) for training our models. Entries annotated with "+ **DRS**" reflect the application of Diversity Resolution Sampling strategy to derive versions at different lower temporal resolutions, as proposed in Ekambaram et al. (2024). All pre-training datasets are strictly disjoint from evaluation sets. For example, `australian electricity demand dataset` and `australian weather dataset` are unrelated to the standard ECL and Weather datasets used in the evaluation tasks.

Motivated by the success of lightweight, task-specialized pre-trained models in the language domain (Schick & Schütze, 2020; Nguyen et al., 2024; Fu et al., 2023; Ling et al., 2024)—which achieve strong performance through minimal task-specific adaptations—we extend this strategy to TSPulse via head reweighting during pretraining. We offer flexibility in pretraining: either by assigning equal weight to all heads or by specializing the pretraining through reweighting loss objectives to prioritize heads most relevant to the target task. Specifically, for task-specialized pre-training, we adjust the head usage and weighting as follows: (i) For anomaly detection, all heads are retained and assigned equal loss weights to facilitate triangulation across time, frequency, and prediction perspectives. (ii) For tasks such as imputation, classification, and retrieval, we emphasize $Head_{time}$ and $Head_{sign}$ by assigning them higher weights, while down-weighting or pruning the remaining heads to reduce task-irrelevant complexity. This enables TSPulse to refine task-specific representations while maintaining its lightweight design, facilitating efficient transfer learning across any dataset for the specified task.

Additionally, based on our validation analysis, we observe that for classification tasks - overall performance improves when using a longer patch size ($pl = 16$) with full patch/block masking during pre-training. This reflects an important insight: while block masking tends to hurt performance in realistic imputation settings (where masking follows irregular patterns - patch or point impute), it is beneficial for classification, where the objective is to learn strong semantic embeddings at the patch level. Block masking introduces more challenging pre-training conditions, helping the model extract more meaningful patch representations.

Thus, TSPulse adopts hybrid masking for all reconstruction-oriented tasks (imputation, anomaly detection, retrieval) and block masking specifically for classification-oriented pre-training. Pre-training one model per task does not pose any practical issues with TSPulse, as its pretraining is extremely fast and efficient. We can pre-train a model on the full ∼1B samples in just one day using 8×A100 GPUs, enabling rapid training of task-specialized models with ease. We use 8-32 A100 GPUs for rapidly pre-training these task-specialized models. For more details on the used evaluation datasets, hyperparameter optimization and train/valid/test split ratios, refer to the associated downstream task sections in the Appendix.

## A.10 FFT EXTRACTION DETAILS

Since TSPulse performs masked reconstruction in both time and frequency domains, we first extract masked FFT features for processing by the backbone and prepare the corresponding ground-truth targets for loss computation.

The input time-series $\mathbf{X} \in \mathbb{R}^{S \times C}$ is scaled and masking is applied in the time domain, resulting in the scaled, masked version $\mathbf{X}_m$. Importantly, we avoid explicitly masking the frequency space. Instead, we directly feed the scaled, masked time-series $\mathbf{X}_m$ into the Fast Fourier Transform (FFT) via the function `torch.fft.rfft` (PyTorch Contributors, 2025). This operation transforms the time-domain signal into the frequency domain. By using the masked input, the transformation naturally propagates the masking into the frequency space, meaning that the same data is masked both in the time and frequency domains, making the reconstruction task more challenging. This step is critical for ensuring that information leakage does not occur between the two domains.

The FFT operation produces a complex-valued tensor of shape $\mathbb{R}^{(S/2)+1 \times C}$, where each element contains both a real and an imaginary component. This represents the signal in the frequency domain, capturing both amplitude and phase information for each frequency bin. To maintain consistency in the output dimensions and to simplify further processing, we discard the last frequency bin, reducing the tensor shape to $\mathbb{R}^{S/2 \times C}$.

Next, we normalize the real and imaginary components separately for each channel by dividing them by their respective maximum absolute values. This step ensures that both components are within a comparable scale, preventing issues related to uneven amplitude distributions across channels. After normalization, the real and imaginary components are concatenated together to form the masked frequency-domain features $\mathbf{X}_m^f \in \mathbb{R}^{S \times C}$. This tensor is then ready for further processing by the backbone, ensuring that it is compatible with the expected input shape and aligns with the time-domain features.

We now compute two separate ground-truths from the frequency representation of the input. These ground-truths serve as targets for the reconstruction tasks during training.

First, we apply the same FFT transformation to the unmasked input, yielding the frequency-domain ground-truth representation $\mathbf{X}^f \in \mathbb{R}^{S \times C}$. This serves as the clean, unaltered frequency representation of the time-series, which will be used to guide the model's reconstruction.

To create the second ground-truth, we compute the log-magnitude spectrum of the frequency representation $\mathbf{X}^f$, excluding the DC component (the first frequency bin, which represents the mean signal). The log-magnitude spectrum is computed as:

$$\text{log-magnitude}(\mathbf{X}^f) = \log\left(\sqrt{\text{Re}(\mathbf{X}^f)^2 + \text{Im}(\mathbf{X}^f)^2}\right)$$

where $\text{Re}(\mathbf{X}^f)$ and $\text{Im}(\mathbf{X}^f)$ represent the real and imaginary parts of the FFT output, respectively. The log transformation reduces the dynamic range of the frequency values, making the learning process more stable and preventing potential issues caused by extreme values in the frequency domain.

After computing the log-magnitude spectrum, we apply softmax across the frequency bins for each channel. This produces $\mathbf{X}_{\text{sign}}^f \in \mathbb{R}^{S/2 \times C}$, a normalized global frequency signature. The softmax operation converts the frequency magnitudes into a probability-like distribution, emphasizing the dominant spectral components. This normalized frequency signature serves as an auxiliary reconstruction target, encouraging the model to learn the most significant global patterns in the frequency domain.

The two ground-truth targets—$\mathbf{X}^f$ (the raw frequency-domain representation) and $\mathbf{X}_{\text{sign}}^f$ (the normalized global frequency signature)—are used during training for the reconstruction task. The first target guides the model in recovering the original frequency representation, while the second target helps the model learn high-level semantic patterns that enhance its generalization to downstream tasks, such as classification or anomaly detection.

## A.11 TIME SERIES ANOMALY DETECTION

### A.11.1 BASICS

Model-based time-series anomaly detection (AD) leverages predictive models—such as *TSPulse*—to identify anomalous regions within a sequence. Any significant deviation between the model's

prediction ($\mathbf{Y} \in \mathbb{R}^{S \times C}$) and the actual observed sequence ($\mathbf{X} \in \mathbb{R}^{S \times C}$) is interpreted as a process deviation and quantified as an anomaly score, denoted by $\alpha$. These scores reflect the degree of divergence and serve as indicators of potential anomalies, particularly when they represent out-of-distribution behavior.

The anomaly score is computed by aggregating the prediction error over a defined temporal window, referred to as the aggregation window, $w$, using Equation 4 as shown below.

$$\alpha_{\tau:\tau+w-1}(\mathbf{X}, \mathbf{Y}) = \frac{1}{wC} \sum_{i=\tau}^{\tau+w-1} \sum_{j=1}^{C} \left( \mathbf{X}_{i,j} - \mathbf{Y}_{i,j} \right)^2 \tag{4}$$

Therefore, a smaller window size ($w$) is more effective for detecting point anomalies, whereas a larger window size is better suited for identifying range anomalies.

### A.11.2 IMPLEMENTATION DETAILS OF MULTI-HEAD TRIANGULATION

TSPulse utilizes both time and frequency representations of time series, and produces multiple prediction outputs, each output either utilizes a specific data representation for data reconstruction or next point prediction. While the reconstruction task relies on estimating data distribution, the prediction task focuses on data evolution. Hence, errors in the reconstruction or prediction tasks can potentially indicate different types of anomalies. We leverage multiple output streams from TSPulse ($\text{Head}_{\text{time}}$, $\text{Head}_{\text{fft}}$, and $\text{Head}_{\text{pred}}$) to derive different anomaly scoring mechanisms:

1. $\text{Head}_{\text{time}}$: The first AD score is generated using the time reconstruction head of TSPulse. Following the notation of Equation 4, the score can be expressed as, $\alpha_{S-w:S-1}^{\text{time}}$. Here, $w < S$, i.e., the aggregation window size is smaller than the context length of the TSPulse, and the last $w$ points from the TSPulse's reconstruction are utilized to calculate the aggregated error. The resulting error is assigned as the anomaly score to the time point at the center of the aggregation window, and this process is repeated with a stride of one.

2. $\text{Head}_{\text{fft}}$: Similarly, this AD score utilizes the FFT reconstruction head of TSPulse, and can be expressed as $\alpha_{S-w:S-1}^{\text{fft}}$.

3. $\text{Head}_{\text{pred}}$: Following the existing pre-trained forecasting models in TSB-AD leaderboard (Liu & Paparrizos, 2024) like Chronos and TimesFM, we utilize the first time point in the forecast horizon coming out of the prediction head of TSPulse to calculate the forecast error, and assign it to that particular time point as the anomaly score: $\alpha_S^{\text{pred}}(\mathbf{X}_{\text{pred}}, \mathbf{Y}_{\text{pred}}) = \frac{1}{C} \sum_{j=1}^{C} \left( [\mathbf{X}_{\text{pred}}]_{S,j} - [\mathbf{Y}_{\text{pred}}]_{S,j} \right)^2$.

4. $\text{Head}_{\text{ensemble}}$: In this scoring mechanism, we temporally align the above three scores, and generate an ensemble score for every time point by computing the maximum of the three scores.

### A.11.3 TSB-AD LEADERBOARD

**Datasets**   The TSB-AD leaderboard (Liu & Paparrizos, 2024) features a comprehensive collection of 1,070 time series drawn from 40 diverse datasets, all curated and annotated by human experts. It stands as one of the most recent and extensive benchmarks for anomaly detection (AD) in time series data.

The leaderboard is divided into two tracks:

- **TSB-AD-U:** Univariate time series anomaly detection,
- **TSB-AD-M:** Multivariate time series anomaly detection.

Each dataset contains one or more time series. To ensure fair evaluation, the data is split into distinct subsets. For detailed information on the data splits (evaluation and tuning), refer to Figure 3 in Liu & Paparrizos (2024). Here, we provide a brief summary:

- **Evaluation Data:** The TSB-AD-U and TSB-AD-M benchmarks contain 350 and 180 time series for evaluation, respectively. Their average anomaly ratios are 4.5% and 5%.

- **Small Training Data:** A short, anomaly-free (hence, unlabeled) historical segment from each evaluation time-series is provided for *optional* model training or fine-tuning. For TSPulse (FT), we utilize a fraction (20%) of this training data as validation data to perform early stopping.

- **Tuning Data:** A set of time-series different from the evaluation series are kept aside for tuning or hyperparameter optimization (HPO). All the models can leverage this tuning data and labels for finding optimal hyperparameters. A total of 48 and 20 time-series are kept for tuning for univariate and multivariate benchmarks, respectively.

**Models and Workflows**   The TSB-AD leaderboard supports two primary workflows for anomaly detection:

- **Unsupervised AD:** Models are applied directly in zero-shot setting to the evaluation data to generate anomaly scores. `TSPulse(ZS)` falls into this category.

- **Semi-Supervised/Self-Supervised AD:** Models are trained or fine-tuned using the small, anomaly-free training segment before being evaluated. Note that no labels are available for the training. `TSPulse(FT)` falls into this category.

In both workflows, models are allowed to use the tuning data (with labels) for hyperparameter optimization.

The leaderboard currently reports results for 40 models across both the univariate and multivariate benchmarks, categorized into three broad sets: statistical (mostly unsupervised), neural networks (mostly self-supervised), and pre-trained/foundation models (both unsupervised and self-supervised).

**HPO for TSPulse**   As mentioned briefly above, the TSB-AD leaderboard (Liu & Paparrizos, 2024) provides separate *tuning datasets* for univariate (TSB-AD-U) and multivariate (TSB-AD-M) benchmarks. All the algorithms (including zero-shot and fine-tuned/trained models) leverage the tuning dataset for hyperparameter tuning. First, we employ the tuning data to find an optimal value of the aggregation window size, $w$, for the two reconstruction-based scoring mechanisms, $\alpha_w^{\text{time}}$ and $\alpha_w^{\text{fft}}$. From a set of three candidate values $(64, 96, 128)$, we find the optimal value, $w^* = 96$, following zero-shot VUS-PR scores obtained in the tuning datasets. Next, we utilize the tuning data for selecting the optimal scoring mechanism for each dataset. For each setting (univariate or multivariate), we perform a zero-shot inference of the TSPulse on the tuning data and generate the VUS-PR metrics for all four scoring mechanisms as stated above. Now, for each dataset, we choose the best scoring mechanism for that dataset (see Table 11 and Table 12). This is performed only once with the zero-shot workflow of TSPulse, and the selection is utilized in both zero-shot and fine-tuned workflows of TSPulse.

### A.11.4   Full Results Tables

Table 13 shows the full version of Figure 4 containing all algorithms. VUS-PR scores for all methods except TSPulse are directly taken from Liu & Paparrizos (2024). Table 14 and Table 15 show the dataset-wise average VUS-PR scores of TSPulse along with SOTA algorithms.

### A.12   Classification

We evaluate TSPulse on the UEA Multivariate Time Series Classification Archive (Bagnall et al., 2018), a widely used benchmark for multivariate time-series classification tasks. Table 16 summarizes the key statistics of the datasets. TSPulse is designed with a base input context length of 512. To accommodate sequences of arbitrary lengths—both shorter and longer—we apply interpolation using `torch.nn.functional.interpolate` (PyTorch Contributors, 2025) to bring them to the base context length 512. Our framework supports up to 20× interpolation, enabling the model to process sequences as long as 10,000 time steps, which is sufficient for most real-world applications. Based on this context scaling capability, we benchmark TSPulse on 29 diverse UEA datasets. Importantly, none of the UEA datasets are included in the TSPulse pre-training corpus.

Since the UEA archive does not provide predefined validation splits, we randomly sample 10% of the training data to get a tuning validation set. We use the cross-entropy loss as the primary

| Dataset | Best TSPulse Output |
|---|---|
| Exathlon | $Head_{fft}$ |
| IOPS | $Head_{time}$ |
| LTDB | $Head_{fft}$ |
| MGAB | $Head_{pred}$ |
| MITDB | $Head_{ensemble}$ |
| MSL | $Head_{time}$ |
| NAB | $Head_{ensemble}$ |
| NEK | $Head_{ensemble}$ |
| OPPORTUNITY | $Head_{ensemble}$ |
| SED | $Head_{pred}$ |
| SMAP | $Head_{time}$ |
| SMD | $Head_{fft}$ |
| SVDB | $Head_{fft}$ |
| Stock | $Head_{pred}$ |
| TAO | $Head_{pred}$ |
| TODS | $Head_{time}$ |
| UCR | $Head_{time}$ |
| WSD | $Head_{ensemble}$ |
| YAHOO | $Head_{pred}$ |

Table 11: Best TSPulse output head as detected on the Tuning dataset for univariate AD benchmark.

| Dataset | Best TSPulse Output |
|---|---|
| CATSv2 | $Head_{pred}$ |
| Exathlon | $Head_{fft}$ |
| GHL | $Head_{fft}$ |
| LTDB | $Head_{time}$ |
| MITDB | $Head_{ensemble}$ |
| MSL | $Head_{time}$ |
| OPPORTUNITY | $Head_{fft}$ |
| SMAP | $Head_{ensemble}$ |
| SMD | $Head_{ensemble}$ |
| SVDB | $Head_{fft}$ |
| TAO | $Head_{pred}$ |

Table 12: Best TSPulse output head as detected on the Tuning dataset for multivariate AD benchmark.

metric for performing Hyperparameter Optimization (HPO). The HPO is conducted based on the performance on the validation data, tuning the following parameters: TSLens projection size $D'$ (1 or 2), mask ratio (0.3 or None), channel expansion factor (1 or 2), and head activation function (softmax or sigmoid). When mask ratio is enabled, block masking is applied during fine-tuning but disabled during evaluation to improve generalization and reduce overfitting. The default decoder configuration uses the *mix-channel* mode, which works effectively for most realistic channel counts (up to several hundred channels). In extreme cases where the number of channels is exceptionally high (e.g., DuckDuckGeese dataset with 1000+ channels), we disable mix-channel and revert to simple avg-pooling for computational efficiency, as the size of channel-mixer blocks scales proportionally with the number of channels. Additionally, given the very large channel count but extremely limited number of training samples in DuckDuckGeese, we employ a k-fold cross-validation for this instance for improved stability. In general, the batch size and number of epochs are chosen from a predefined pool set based on the scale of the training samples to prevent out-of-memory error and enable faster fine-tuning.

We benchmark TSPulse on classification against a broad set of baselines, including recent pre-trained models (MOMENT (Goswami et al., 2024), VQShape (Wen et al., 2024), UniTS (Gao et al., 2024)) and data-specific methods (T-Rep (Fraikin et al., 2024), TS2Vec (Yue et al., 2022), TNC (Tonekaboni et al., 2021), T-Loss (Franceschi et al., 2019), TS-TCC (Eldele et al., 2021)). As detailed in the Table 9, we either report the results directly from the paper or run their official implementation. Specifically, for UniTS, we adopt the prompt-tuned (PMT) approach proposed in Gao et al. (2024), applying it to the available pretrained model across all 29 datasets in a multi-task setting and report the resulting accuracy. The overall classification results, including comparisons against baselines, are presented in Table 17, and detailed ablation studies are shown in Table 18.

| Method | VUS-PR |
|---|---|
| TSPulse (FT) | 0.52 |
| TSPulse (ZS) | 0.48 |
| Sub-PCA | 0.42 |
| KShapeAD | 0.40 |
| POLY | 0.39 |
| Series2Graph | 0.39 |
| MOMENT (FT) | 0.39 |
| MOMENT (ZS) | 0.38 |
| KMeansAD | 0.37 |
| USAD | 0.36 |
| Sub-KNN | 0.35 |
| MatrixProfile | 0.35 |
| SAND | 0.34 |
| CNN | 0.34 |
| LSTMAD | 0.33 |
| SR | 0.32 |
| TimesFM | 0.30 |
| IForest | 0.30 |
| OmniAnomaly | 0.29 |
| Lag-Llama | 0.27 |
| Chronos | 0.27 |
| TimesNet | 0.26 |
| AutoEncoder | 0.26 |
| TranAD | 0.26 |
| FITS | 0.26 |
| Sub-LOF | 0.25 |
| OFA | 0.24 |
| Sub-MCD | 0.24 |
| Sub-HBOS | 0.23 |
| Sub-OCSVM | 0.23 |
| Sub-IForest | 0.22 |
| Donut | 0.20 |
| LOF | 0.17 |
| AnomalyTransformer | 0.12 |

(a) TSB-AD-U

| Method | VUS-PR |
|---|---|
| TSPulse (FT) | 0.39 |
| TSPulse (ZS) | 0.36 |
| CNN | 0.31 |
| OmniAnomaly | 0.31 |
| PCA | 0.31 |
| LSTMAD | 0.31 |
| USAD | 0.30 |
| AutoEncoder | 0.30 |
| KMeansAD | 0.29 |
| CBLOF | 0.27 |
| MCD | 0.27 |
| OCSVM | 0.26 |
| Donut | 0.26 |
| RobustPCA | 0.24 |
| FITS | 0.21 |
| OFA | 0.21 |
| EIF | 0.21 |
| COPOD | 0.20 |
| IForest | 0.20 |
| HBOS | 0.19 |
| TimesNet | 0.19 |
| KNN | 0.18 |
| TranAD | 0.18 |
| LOF | 0.14 |
| AnomalyTransformer | 0.12 |

(b) TSB-AD-M

Table 13: **Anomaly Detection Full Table**. VUS-PR metric (Higher is better) averaged across all time series across all datasets for all available algorithms in the TSB-AD leaderabord.

## A.13   IMPUTATION

**Dataset**   We evaluate TSPulse and several baseline methods on six real-world datasets from the LTSF benchmark suite (Wu et al., 2021)—ETTh1, ETTh2, ETTm1, ETTm2, Weather, and Electricity. We use context length = 512, stride = 1 and the same train/valid/test split ratio as proposed in Wu et al. (2021).

**Masking Strategies**   We assess model performance under two masking regimes: *block masking* and *hybrid masking*, defined as follows:

- **Block Masking:** This setup simulates scenarios like sensor failures or maintenance intervals, where data is missing in contiguous segments. Specifically, blocks of 8 consecutive time-points (i.e. full patch) are masked and must be reconstructed by the model.

- **Hybrid Masking:** To reflect more realistic settings, this scheme introduces both structured and unstructured missingness. Half of the missing points occur in full-length patches (8-point blocks), while the other half are scattered randomly across the sequence. This design captures a mix of localized and irregular masking patterns.

**Baselines**   To evaluate the effectiveness of TSPulse under both masking setups, we benchmark it against a diverse set of baselines categorized as follows:

- **Zero-Shot / Prompt-Tuned / Statistical:** We include two recent pre-trained models—MOMENT (large) (Goswami et al., 2024) and UniTS (Gao et al., 2024). MOMENT is evaluated in a pure zero-shot manner, consistent with TSPulse. UniTS is prompt-tuned using

| Method | CATSv2 | Daphnet | Exathlon | IOPS | LTDB | MGAB | MITDB | MSL | NAB | NEK | OPPORTUNITY | Power | SED | SMAP | SMD | SVDB | SWaT | Stock | TAO | TODS | UCR | WSD | YAHOO |
|---|---|---|---|---|---|---|---|---|---|---|---|---|---|---|---|---|---|---|---|---|---|---|---|
| TSPulse (FT) | 0.40 | 0.54 | 0.87 | 0.42 | 0.52 | 0.01 | 0.22 | 0.66 | 0.51 | 0.58 | 0.06 | 0.08 | 0.07 | 0.74 | 0.80 | 0.56 | 0.10 | 0.98 | 1.00 | 0.68 | 0.28 | 0.43 | 0.83 |
| TSPulse (ZS) | 0.35 | 0.55 | 0.83 | 0.42 | 0.41 | 0.00 | 0.10 | 0.64 | 0.50 | 0.58 | 0.06 | 0.07 | 0.06 | 0.71 | 0.78 | 0.36 | 0.10 | 0.98 | 1.00 | 0.68 | 0.18 | 0.42 | 0.83 |
| Sub-PCA | 0.26 | 0.42 | 0.93 | 0.23 | 0.56 | 0.01 | 0.36 | 0.51 | 0.44 | 0.91 | 0.91 | 0.08 | 0.03 | 0.52 | 0.45 | 0.52 | 0.39 | 0.84 | 0.93 | 0.54 | 0.12 | 0.09 | 0.14 |
| KShapeAD | 0.25 | 0.04 | 0.33 | 0.09 | 0.83 | 0.02 | 0.69 | 0.55 | 0.37 | 0.24 | 0.33 | 0.19 | 0.89 | 0.58 | 0.13 | 0.82 | 0.43 | 0.75 | 0.91 | 0.75 | 0.38 | 0.10 | 0.55 |
| POLY | 0.23 | 0.51 | 0.74 | 0.31 | 0.51 | 0.01 | 0.34 | 0.54 | 0.48 | 0.61 | 0.10 | 0.09 | 0.04 | 0.64 | 0.61 | 0.44 | 0.10 | 0.82 | 0.92 | 0.57 | 0.13 | 0.41 | 0.25 |
| Series2Graph | 0.21 | 0.19 | 0.60 | 0.22 | 0.79 | 0.00 | 0.61 | 0.25 | 0.44 | 0.67 | 0.11 | 0.07 | 0.15 | 0.55 | 0.46 | 0.55 | 0.22 | 0.79 | 0.91 | 0.73 | 0.25 | 0.27 | 0.28 |
| MOMENT (FT) | 0.38 | 0.51 | 0.83 | 0.38 | 0.45 | 0.00 | 0.13 | 0.53 | 0.39 | 0.73 | 0.07 | 0.07 | 0.04 | 0.63 | 0.75 | 0.23 | 0.08 | 0.81 | 0.94 | 0.58 | 0.08 | 0.50 | 0.25 |
| MOMENT (ZS) | 0.30 | 0.52 | 0.81 | 0.37 | 0.44 | 0.00 | 0.14 | 0.53 | 0.39 | 0.73 | 0.07 | 0.08 | 0.04 | 0.62 | 0.74 | 0.27 | 0.07 | 0.81 | 0.94 | 0.58 | 0.07 | 0.49 | 0.23 |
| KMeansAD | 0.23 | 0.04 | 0.41 | 0.06 | 0.49 | 0.01 | 0.27 | 0.48 | 0.33 | 0.20 | 0.30 | 0.39 | 0.87 | 0.63 | 0.18 | 0.44 | 0.10 | 0.76 | 0.92 | 0.65 | 0.38 | 0.10 | 0.56 |
| USAD | 0.40 | 0.12 | 0.89 | 0.13 | 0.55 | 0.00 | 0.18 | 0.27 | 0.28 | 0.73 | 0.67 | 0.06 | 0.03 | 0.66 | 0.43 | 0.37 | 0.10 | 0.75 | 0.93 | 0.52 | 0.08 | 0.04 | 0.10 |
| Sub-KNN | 0.29 | 0.04 | 0.47 | 0.10 | 0.58 | 0.24 | 0.36 | 0.33 | 0.29 | 0.23 | 0.30 | 0.21 | 0.87 | 0.51 | 0.14 | 0.56 | 0.10 | 0.75 | 0.92 | 0.65 | 0.37 | 0.10 | 0.31 |
| MatrixProfile | 0.36 | 0.04 | 0.56 | 0.10 | 0.58 | 0.29 | 0.39 | 0.48 | 0.32 | 0.13 | 0.25 | 0.15 | 0.72 | 0.47 | 0.13 | 0.36 | 0.11 | 0.76 | 0.93 | 0.76 | 0.34 | 0.02 | 0.43 |
| SAND | 0.27 | 0.04 | 0.25 | 0.06 | 0.79 | 0.01 | 0.67 | 0.30 | 0.38 | 0.32 | 0.18 | 0.16 | 0.75 | 0.56 | 0.11 | 0.72 | 0.21 | 0.74 | 0.91 | 0.70 | 0.34 | 0.08 | 0.41 |
| CNN | 0.32 | 0.40 | 0.61 | 0.26 | 0.42 | 0.01 | 0.15 | 0.33 | 0.19 | 0.73 | 0.40 | 0.08 | 0.06 | 0.34 | 0.55 | 0.21 | 0.68 | 0.92 | 1.00 | 0.54 | 0.05 | 0.24 | 0.53 |
| LSTMAD | 0.33 | 0.13 | 0.73 | 0.20 | 0.36 | 0.03 | 0.12 | 0.32 | 0.18 | 0.73 | 0.58 | 0.07 | 0.06 | 0.26 | 0.49 | 0.13 | 0.67 | 0.85 | 1.00 | 0.47 | 0.02 | 0.13 | 0.45 |
| SR | 0.28 | 0.20 | 0.73 | 0.24 | 0.29 | 0.01 | 0.07 | 0.22 | 0.20 | 0.50 | 0.33 | 0.10 | 0.07 | 0.29 | 0.36 | 0.08 | 0.35 | 1.00 | 1.00 | 0.64 | 0.07 | 0.22 | 0.61 |
| TimesFM | 0.25 | 0.36 | 0.53 | 0.20 | 0.27 | 0.00 | 0.06 | 0.32 | 0.18 | 0.35 | 0.05 | 0.08 | 0.05 | 0.30 | 0.40 | 0.06 | 0.22 | 0.99 | 0.99 | 0.75 | 0.07 | 0.21 | 0.81 |
| IForest | 0.08 | 0.36 | 0.67 | 0.28 | 0.34 | 0.00 | 0.10 | 0.29 | 0.22 | 0.59 | 0.43 | 0.08 | 0.36 | 0.25 | 0.34 | 0.09 | 0.50 | 0.99 | 0.99 | 0.52 | 0.02 | 0.14 | 0.44 |
| OmniAnomaly | 0.12 | 0.16 | 0.83 | 0.20 | 0.32 | 0.00 | 0.10 | 0.25 | 0.19 | 0.85 | 0.60 | 0.07 | 0.06 | 0.15 | 0.36 | 0.09 | 0.44 | 0.82 | 0.98 | 0.44 | 0.03 | 0.14 | 0.19 |
| Lag-Llama | 0.21 | 0.39 | 0.53 | 0.22 | 0.29 | 0.00 | 0.08 | 0.31 | 0.18 | 0.38 | 0.05 | 0.08 | 0.07 | 0.28 | 0.36 | 0.08 | 0.09 | 0.97 | 0.99 | 0.61 | 0.02 | 0.22 | 0.68 |
| Chronos | 0.10 | 0.31 | 0.45 | 0.18 | 0.26 | 0.00 | 0.06 | 0.18 | 0.18 | 0.34 | 0.06 | 0.08 | 0.06 | 0.19 | 0.32 | 0.06 | 0.14 | 0.99 | 1.00 | 0.70 | 0.07 | 0.18 | 0.80 |
| TimesNet | 0.10 | 0.39 | 0.53 | 0.22 | 0.29 | 0.00 | 0.08 | 0.31 | 0.20 | 0.37 | 0.05 | 0.08 | 0.05 | 0.38 | 0.54 | 0.09 | 0.11 | 0.79 | 0.91 | 0.59 | 0.02 | 0.27 | 0.29 |
| AutoEncoder | 0.18 | 0.09 | 0.36 | 0.25 | 0.69 | 0.01 | 0.07 | 0.27 | 0.32 | 0.51 | 0.12 | 0.09 | 0.41 | 0.49 | 0.14 | 0.32 | 0.38 | 0.72 | 0.93 | 0.65 | 0.09 | 0.14 | 0.29 |
| TranAD | 0.08 | 0.13 | 0.72 | 0.18 | 0.31 | 0.00 | 0.09 | 0.18 | 0.18 | 0.72 | 0.58 | 0.07 | 0.05 | 0.13 | 0.16 | 0.09 | 0.46 | 0.79 | 0.94 | 0.45 | 0.02 | 0.11 | 0.28 |
| FITS | 0.17 | 0.43 | 0.55 | 0.17 | 0.34 | 0.00 | 0.09 | 0.36 | 0.24 | 0.49 | 0.07 | 0.07 | 0.05 | 0.42 | 0.54 | 0.10 | 0.10 | 0.76 | 0.91 | 0.58 | 0.02 | 0.14 | 0.18 |
| Sub-LOF | 0.31 | 0.04 | 0.25 | 0.11 | 0.34 | 0.44 | 0.26 | 0.35 | 0.32 | 0.25 | 0.12 | 0.14 | 0.22 | 0.40 | 0.04 | 0.18 | 0.11 | 0.76 | 0.92 | 0.53 | 0.29 | 0.03 | 0.27 |
| OFA | 0.16 | 0.36 | 0.55 | 0.20 | 0.30 | 0.00 | 0.07 | 0.29 | 0.21 | 0.37 | 0.05 | 0.08 | 0.06 | 0.33 | 0.45 | 0.07 | 0.11 | 0.76 | 0.91 | 0.54 | 0.02 | 0.16 | 0.24 |
| Sub-MCD | 0.37 | 0.04 | 0.23 | 0.13 | 0.24 | 0.01 | 0.11 | 0.16 | 0.19 | 0.11 | 0.32 | 0.30 | 0.12 | 0.30 | 0.08 | 0.07 | 0.09 | 0.75 | 0.90 | 0.64 | 0.26 | 0.15 | 0.28 |
| Sub-HBOS | 0.04 | 0.05 | 0.45 | 0.05 | 0.69 | 0.00 | 0.17 | 0.25 | 0.30 | 0.23 | 0.08 | 0.12 | 0.88 | 0.55 | 0.10 | 0.24 | 0.12 | 0.70 | 0.93 | 0.64 | 0.14 | 0.01 | 0.06 |
| Sub-OCSVM | 0.26 | 0.06 | 0.29 | 0.07 | 0.33 | 0.01 | 0.14 | 0.28 | 0.26 | 0.11 | 0.16 | 0.06 | 0.06 | 0.51 | 0.08 | 0.20 | 0.09 | 0.73 | 0.92 | 0.65 | 0.18 | 0.03 | 0.23 |
| Sub-IForest | 0.05 | 0.07 | 0.49 | 0.04 | 0.66 | 0.00 | 0.24 | 0.36 | 0.30 | 0.22 | 0.07 | 0.12 | 0.79 | 0.47 | 0.09 | 0.27 | 0.13 | 0.69 | 0.90 | 0.66 | 0.10 | 0.01 | 0.06 |
| Donut | 0.08 | 0.06 | 0.45 | 0.10 | 0.31 | 0.00 | 0.10 | 0.20 | 0.18 | 0.47 | 0.18 | 0.09 | 0.14 | 0.31 | 0.29 | 0.08 | 0.47 | 0.78 | 0.91 | 0.48 | 0.01 | 0.06 | 0.12 |
| LOF | 0.06 | 0.13 | 0.20 | 0.12 | 0.26 | 0.00 | 0.06 | 0.15 | 0.17 | 0.38 | 0.14 | 0.09 | 0.11 | 0.15 | 0.13 | 0.05 | 0.12 | 0.75 | 0.91 | 0.49 | 0.02 | 0.09 | 0.37 |
| AnomalyTransformer | 0.05 | 0.07 | 0.13 | 0.06 | 0.27 | 0.00 | 0.09 | 0.14 | 0.14 | 0.23 | 0.07 | 0.09 | 0.09 | 0.09 | 0.18 | 0.07 | 0.10 | 0.75 | 0.90 | 0.46 | 0.01 | 0.02 | 0.07 |

Table 14: **VUS-PR score (Higher is better) averaged over all time series for each dataset** for univariate anomaly detection. Note that every dataset has one or more time series, and the mean scores reported in Figure 4a in the main text and Table 13 above are averaged over all 350 "Eval" time series (similar to the TSB-AD-U leaderboard). This table can be compared with the dataset-wise Table-6 of Liu & Paparrizos (2024).

| Method | CATSv2 | CreditCard | Daphnet | Exathlon | GECCO | GHL | Genesis | LTDB | MITDB | MSL | OPPORTUNITY | PSM | SMAP | SMD | SVDB | SWaT | TAO |
|---|---|---|---|---|---|---|---|---|---|---|---|---|---|---|---|---|---|
| TSPulse (FT) | 0.07 | 0.00 | 0.35 | 0.91 | 0.18 | 0.01 | 0.02 | 0.57 | 0.14 | 0.21 | 0.07 | 0.14 | 0.32 | 0.36 | 0.47 | 0.14 | 0.93 |
| TSPulse (ZS) | 0.05 | 0.00 | 0.35 | 0.89 | 0.17 | 0.01 | 0.01 | 0.36 | 0.07 | 0.20 | 0.07 | 0.14 | 0.30 | 0.35 | 0.38 | 0.13 | 0.93 |
| CNN | 0.08 | 0.02 | 0.21 | 0.68 | 0.03 | 0.02 | 0.00 | 0.33 | 0.14 | 0.35 | 0.16 | 0.22 | 0.19 | 0.35 | 0.19 | 0.41 | 1.00 |
| OmniAnomaly | 0.04 | 0.02 | 0.34 | 0.84 | 0.02 | 0.07 | 0.00 | 0.44 | 0.11 | 0.22 | 0.18 | 0.16 | 0.12 | 0.17 | 0.35 | 0.15 | 0.81 |
| PCA | 0.12 | 0.10 | 0.13 | 0.95 | 0.20 | 0.01 | 0.02 | 0.24 | 0.07 | 0.15 | 0.30 | 0.16 | 0.09 | 0.36 | 0.11 | 0.45 | 1.00 |
| LSTMAD | 0.04 | 0.02 | 0.31 | 0.82 | 0.02 | 0.06 | 0.04 | 0.30 | 0.09 | 0.22 | 0.17 | 0.24 | 0.16 | 0.33 | 0.15 | 0.16 | 0.99 |
| USAD | 0.04 | 0.02 | 0.34 | 0.84 | 0.02 | 0.06 | 0.00 | 0.41 | 0.12 | 0.23 | 0.18 | 0.19 | 0.11 | 0.16 | 0.32 | 0.15 | 0.81 |
| AutoEncoder | 0.06 | 0.03 | 0.13 | 0.91 | 0.05 | 0.05 | 0.01 | 0.21 | 0.04 | 0.22 | 0.14 | 0.28 | 0.13 | 0.30 | 0.06 | 0.58 | 1.00 |
| KMeansAD | 0.12 | 0.02 | 0.30 | 0.37 | 0.06 | 0.03 | 0.89 | 0.41 | 0.06 | 0.44 | 0.06 | 0.21 | 0.38 | 0.36 | 0.20 | 0.16 | 0.86 |
| CBLOF | 0.06 | 0.03 | 0.10 | 0.86 | 0.03 | 0.02 | 0.02 | 0.20 | 0.04 | 0.21 | 0.14 | 0.19 | 0.14 | 0.22 | 0.07 | 0.29 | 1.00 |
| MCD | 0.13 | 0.06 | 0.14 | 0.80 | 0.03 | 0.01 | 0.06 | 0.21 | 0.04 | 0.23 | 0.17 | 0.26 | 0.10 | 0.26 | 0.07 | 0.54 | 1.00 |
| OCSVM | 0.08 | 0.02 | 0.06 | 0.83 | 0.04 | 0.04 | 0.08 | 0.20 | 0.04 | 0.22 | 0.12 | 0.19 | 0.12 | 0.28 | 0.06 | 0.44 | 0.81 |
| Donut | 0.07 | 0.02 | 0.17 | 0.66 | 0.03 | 0.05 | 0.18 | 0.26 | 0.12 | 0.30 | 0.15 | 0.20 | 0.18 | 0.19 | 0.11 | 0.44 | 0.75 |
| RobustPCA | 0.04 | 0.02 | 0.06 | 0.77 | 0.02 | 0.03 | 0.00 | 0.23 | 0.04 | 0.22 | 0.13 | 0.12 | 0.07 | 0.10 | 0.08 | 0.12 | 1.00 |
| FITS | 0.13 | 0.02 | 0.33 | 0.63 | 0.03 | 0.01 | 0.10 | 0.23 | 0.05 | 0.17 | 0.05 | 0.13 | 0.08 | 0.17 | 0.10 | 0.15 | 0.78 |
| OFA | 0.13 | 0.02 | 0.31 | 0.58 | 0.04 | 0.01 | 0.22 | 0.29 | 0.06 | 0.14 | 0.05 | 0.17 | 0.08 | 0.17 | 0.12 | 0.12 | 0.78 |
| EIF | 0.06 | 0.02 | 0.15 | 0.41 | 0.04 | 0.02 | 0.06 | 0.19 | 0.04 | 0.18 | 0.10 | 0.18 | 0.13 | 0.32 | 0.07 | 0.32 | 0.89 |
| COPOD | 0.05 | 0.05 | 0.11 | 0.40 | 0.04 | 0.03 | 0.08 | 0.21 | 0.04 | 0.21 | 0.17 | 0.20 | 0.10 | 0.19 | 0.07 | 0.31 | 0.99 |
| IForest | 0.05 | 0.03 | 0.13 | 0.35 | 0.04 | 0.05 | 0.04 | 0.21 | 0.04 | 0.21 | 0.18 | 0.19 | 0.09 | 0.26 | 0.07 | 0.39 | 0.93 |
| HBOS | 0.05 | 0.04 | 0.15 | 0.32 | 0.04 | 0.04 | 0.08 | 0.21 | 0.04 | 0.23 | 0.17 | 0.17 | 0.09 | 0.25 | 0.07 | 0.30 | 0.83 |
| TimesNet | 0.07 | 0.02 | 0.27 | 0.42 | 0.03 | 0.01 | 0.02 | 0.27 | 0.07 | 0.17 | 0.06 | 0.14 | 0.09 | 0.14 | 0.11 | 0.14 | 0.79 |
| KNN | 0.07 | 0.02 | 0.25 | 0.33 | 0.11 | 0.01 | 0.04 | 0.19 | 0.04 | 0.18 | 0.06 | 0.12 | 0.12 | 0.30 | 0.06 | 0.11 | 0.78 |
| TranAD | 0.04 | 0.02 | 0.31 | 0.10 | 0.02 | 0.06 | 0.04 | 0.26 | 0.07 | 0.24 | 0.16 | 0.23 | 0.09 | 0.30 | 0.12 | 0.15 | 0.81 |
| LOF | 0.05 | 0.02 | 0.11 | 0.16 | 0.13 | 0.01 | 0.08 | 0.19 | 0.04 | 0.14 | 0.10 | 0.15 | 0.09 | 0.16 | 0.06 | 0.15 | 0.79 |
| AnomalyTransformer | 0.03 | 0.02 | 0.07 | 0.10 | 0.02 | 0.03 | 0.01 | 0.21 | 0.05 | 0.12 | 0.07 | 0.21 | 0.06 | 0.07 | 0.08 | 0.18 | 0.77 |

Table 15: **VUS-PR score (Higher is better) averaged over all time series for each dataset** for multivariate anomaly detection. Note that every dataset has one or more time series, and the mean scores reported in Figure 4b in the main text and Table 13 above are averaged over all 180 "Eval" time series (similar to the TSB-AD-M leaderboard). This table can be compared with the dataset-wise Table-7 of Liu & Paparrizos (2024).

| ID | Dataset | # of {Train, Test} Samples | Classes | {Length, Channels} |
|---|---|---|---|---|
| 1 | ArticularyWordRecognition | {275, 300} | 25 | {144, 9} |
| 2 | AtrialFibrillation | {15, 15} | 3 | {640, 2} |
| 3 | BasicMotions | {40, 40} | 4 | {100, 6} |
| 4 | Cricket | {108, 72} | 12 | {1197, 6} |
| 5 | Epilepsy | {137, 138} | 4 | {206, 3} |
| 6 | EthanolConcentration | {261, 263} | 4 | {1751, 3} |
| 7 | ERing | {30, 30} | 6 | {65, 4} |
| 8 | FingerMovements | {316, 100} | 2 | {50, 28} |
| 9 | HandMovementDirection | {320, 147} | 4 | {400, 10} |
| 10 | Handwriting | {150, 850} | 26 | {152, 3} |
| 11 | JapaneseVowels | {270, 370} | 9 | {29, 12} |
| 12 | Libras | {180, 180} | 15 | {45, 2} |
| 13 | LSST | {2459, 2466} | 14 | {36, 6} |
| 14 | MotorImagery | {278, 100} | 2 | {3000, 64} |
| 15 | NATOPS | {180, 180} | 6 | {51, 24} |
| 16 | RacketSports | {151, 152} | 4 | {30, 6} |
| 17 | SelfRegulationSCP1 | {268, 293} | 2 | {896, 6} |
| 18 | SelfRegulationSCP2 | {200, 180} | 2 | {1152, 7} |
| 19 | SpokenArabicDigits | {6599, 2199} | 10 | {93, 13} |
| 20 | StandWalkJump | {12, 15} | 3 | {2500, 4} |
| 21 | UWaveGestureLibrary | {120, 320} | 8 | {315, 3} |
| 22 | PEMS-SF | {267, 173} | 7 | {144, 963} |
| 23 | Phoneme | {3315, 3353} | 39 | {217, 11} |
| 24 | PenDigits | {7494, 3498} | 10 | {8, 2} |
| 25 | Heartbeat | {204, 205} | 2 | {405, 61} |
| 26 | FaceDetection | {5890, 3524} | 2 | {62, 144} |
| 27 | CharacterTrajectories | {1422, 1436} | 20 | {182, 3} |
| 28 | DuckDuckGeese | {60, 40} | 5 | {270, 1345} |
| 29 | InsectWingbeat | {30000, 20000} | 10 | {78, 200} |

Table 16: Summary of the UEA Multivariate Time Series Classification Archive (Bagnall et al., 2018)

10% of the dataset (across all six benchmarks) in a multi-task setting, as per the original implementation.

*MOMENT Setup:* For block masking, missing patches are replaced with the learned mask token in the embedding space, following Goswami et al. (2024). In hybrid masking, we adopt a mixed approach—using the mask token for fully missing 8-point patches and zeros for the irregularly missing points. This **Heterogeneous** strategy consistently outperformed the simpler **Only-zeros** strategy (where all missing points are zeroed out) for MOMENT, as detailed in Table 25.

*UniTS Setup:* We similarly compare **Heterogeneous** and **Only-zeros** masking strategies. For UniTS, the **Only-zeros** approach yielded better results—likely due to prompt tuning on 10% of the data, allowing the model to better adapt to uniformly zero-masked inputs than to a mix of learned tokens and zeros. Full ablations are shown in Table 25.

*Note:* TSPulse uses a single learnable raw-level mask patch, enabling it to uniformly handle both block and hybrid masking without requiring masking strategy-specific adjustments.

In addition to these pre-trained models, we evaluate four classical interpolation methods: Naive, Linear, Nearest, and Cubic.

- **Fine-Tuned / Supervised:** We include fully supervised baselines—TimesNet (Wu et al., 2022), Non-Stationary Transformer (Non-Stat.) (Liu et al., 2022), and FEDformer (Zhou et al., 2022)—trained end-to-end on the complete datasets individually. These are compared against the fine-tuned variant of TSPulse (TSPulse-FT).

**Results** Figure 6 and Figure 13 summarizes TSPulse's imputation performance across all six datasets and masking ratios under hybrid and block masking setups respectively. Detailed per-dataset results averaged over four mask ratios are presented in Table 19. Complete breakdowns for each dataset and ratio are provided in Table 20 (hybrid masking) and Table 21 (block masking).

| Dataset | TSPulse | VQShape | MOMENT | UniTS | T-Rep | TS2Vec | T-Loss | TNC | TS-TCC |
|---|---|---|---|---|---|---|---|---|---|
| ArticularyWordRecognition | 0.98 | 0.987 | 0.99 | 0.947 | 0.968 | 0.987 | 0.943 | 0.973 | 0.953 |
| AtrialFibrillation | 0.467 | 0.52 | 0.2 | 0.2 | 0.354 | 0.2 | 0.133 | 0.133 | 0.267 |
| BasicMotions | 1.0 | 0.885 | 1.0 | 0.8 | 1.0 | 0.975 | 1.0 | 0.975 | 1.0 |
| CharacterTrajectories | 0.987 | 0.962 | 0.964* | 0.968 | 0.989 | 0.995 | 0.993 | 0.967 | 0.985 |
| Cricket | 0.917 | 0.975 | 0.986 | 0.958 | 0.958 | 0.972 | 0.972 | 0.958 | 0.917 |
| DuckDuckGeese | 0.72 | 0.344 | 0.6 | 0.26 | 0.457 | 0.68 | 0.65 | 0.46 | 0.38 |
| ERing | 0.937 | 0.947 | 0.959 | 0.841 | 0.943 | 0.874 | 0.133 | 0.852 | 0.904 |
| Epilepsy | 0.971 | 0.755 | 0.993 | 0.949 | 0.97 | 0.964 | 0.971 | 0.957 | 0.957 |
| EthanolConcentration | 0.247 | 0.306 | 0.357 | 0.266 | 0.333 | 0.308 | 0.205 | 0.297 | 0.285 |
| FaceDetection | 0.675 | 0.653 | 0.633 | 0.569 | 0.581 | 0.501 | 0.513 | 0.536 | 0.544 |
| FingerMovements | 0.53 | 0.616 | 0.49 | 0.53 | 0.495 | 0.48 | 0.58 | 0.47 | 0.46 |
| HandMovementDirection | 0.649 | 0.514 | 0.324 | 0.284 | 0.536 | 0.338 | 0.351 | 0.324 | 0.243 |
| Handwriting | 0.215 | 0.266 | 0.308 | 0.149 | 0.414 | 0.515 | 0.451 | 0.249 | 0.498 |
| Heartbeat | 0.702 | 0.632 | 0.722 | 0.61 | 0.725 | 0.683 | 0.741 | 0.746 | 0.75 |
| InsectWingbeat | 0.723 | NA | 0.246 | 0.484 | 0.328 | 0.466 | 0.156 | 0.469 | 0.264 |
| JapaneseVowels | 0.976 | 0.941 | 0.716 | 0.851 | 0.962 | 0.984 | 0.989 | 0.978 | 0.93 |
| LSST | 0.518 | 0.511 | 0.411 | 0.467 | 0.526 | 0.537 | 0.509 | 0.595 | 0.474 |
| Libras | 0.767 | 0.808 | 0.85 | 0.756 | 0.829 | 0.867 | 0.883 | 0.817 | 0.822 |
| MotorImagery | 0.58 | 0.638 | 0.5 | 0.49 | 0.495 | 0.51 | 0.58 | 0.5 | 0.61 |
| NATOPS | 0.878 | 0.804 | 0.828 | 0.75 | 0.804 | 0.928 | 0.917 | 0.911 | 0.822 |
| PEMS-SF | 0.855 | 0.85 | 0.896 | 0.855 | 0.8 | 0.682 | 0.676 | 0.699 | 0.734 |
| PenDigits | 0.969 | 0.973 | 0.972 | 0.956 | 0.971 | 0.989 | 0.981 | 0.979 | 0.974 |
| PhonemeSpectra | 0.156 | 0.087 | 0.233 | 0.173 | 0.232 | 0.233 | 0.222 | 0.207 | 0.252 |
| RacketSports | 0.901 | 0.838 | 0.796 | 0.697 | 0.883 | 0.855 | 0.855 | 0.776 | 0.816 |
| SelfRegulationSCP1 | 0.836 | 0.889 | 0.84 | 0.823 | 0.819 | 0.812 | 0.843 | 0.799 | 0.823 |
| SelfRegulationSCP2 | 0.511 | 0.586 | 0.478 | 0.556 | 0.591 | 0.578 | 0.539 | 0.55 | 0.533 |
| SpokenArabicDigits | 0.984 | 0.976 | 0.981 | 0.957 | 0.994 | 0.988 | 0.905 | 0.934 | 0.97 |
| StandWalkJump | 0.733 | 0.707 | 0.4 | 0.4 | 0.441 | 0.467 | 0.333 | 0.4 | 0.333 |
| UWaveGestureLibrary | 0.881 | 0.879 | 0.909 | 0.85 | 0.885 | 0.906 | 0.875 | 0.759 | 0.753 |
| **Mean** | **0.733** | **0.701** | **0.675** | **0.634** | **0.699** | **0.699** | **0.652** | **0.665** | **0.664** |
| **IMP(%)** | | **5%** | **9%** | **16%** | **5%** | **5%** | **12%** | **10%** | **10%** |

Table 17: **Classification accuracy results (Higher is better) on UEA datasets.** We report both the mean accuracy and the IMP (%)—the percentage improvement of TSPulse over each baseline in terms of mean accuracy. On the InsectWingbeat dataset, VQShape encountered an out-of-memory (OOM) error due to the large data volume and its relatively large model size. Accordingly, the VQShape result is reported as NA for this dataset. However, for averaging purposes, we substitute the best-performing baseline result in place of VQShape to maintain consistency in the overall mean accuracy comparison. *For CharacterTrajectories dataset we used the official implementation of MOMENT to get the results.

| Data | TSPulse | w/o Short Emb | w/o Long Emb | w/o Mask | w/o CM Identity Init | w/o Channel Expansion | w/o TSLens (Avg-Pool) | w/o TSLens (Max-Pool) | w/o Dual-space |
|---|---|---|---|---|---|---|---|---|---|
| ArticularyWordRecognition | 0.98 | 0.973 | 0.963 | 0.983 | 0.973 | 0.987 | 0.973 | 0.953 | 0.98 |
| AtrialFibrillation | 0.467 | 0.267 | 0.333 | 0.333 | 0.333 | 0.467 | 0.333 | 0.333 | 0.467 |
| BasicMotions | 1.0 | 1.0 | 0.975 | 1.0 | 0.975 | 0.975 | 0.975 | 0.875 | 1.0 |
| Cricket | 0.917 | 0.944 | 0.917 | 0.944 | 0.944 | 0.931 | 0.972 | 0.986 | 0.944 |
| ERing | 0.937 | 0.922 | 0.889 | 0.919 | 0.922 | 0.937 | 0.837 | 0.848 | 0.915 |
| Epilepsy | 0.971 | 0.986 | 0.942 | 0.978 | 0.964 | 0.971 | 0.957 | 0.899 | 0.949 |
| EthanolConcentration | 0.247 | 0.262 | 0.247 | 0.262 | 0.251 | 0.259 | 0.319 | 0.243 | 0.266 |
| FingerMovements | 0.53 | 0.52 | 0.54 | 0.51 | 0.57 | 0.52 | 0.46 | 0.44 | 0.48 |
| HandMovementDirection | 0.649 | 0.486 | 0.486 | 0.351 | 0.527 | 0.649 | 0.473 | 0.351 | 0.473 |
| Handwriting | 0.215 | 0.189 | 0.055 | 0.215 | 0.199 | 0.026 | 0.034 | 0.182 | 0.175 |
| JapaneseVowels | 0.976 | 0.976 | 0.97 | 0.968 | 0.959 | 0.981 | 0.968 | 0.719 | 0.97 |
| MotorImagery | 0.58 | 0.47 | 0.52 | 0.58 | 0.52 | 0.58 | 0.47 | 0.47 | 0.49 |
| NATOPS | 0.878 | 0.817 | 0.861 | 0.861 | 0.811 | 0.878 | 0.894 | 0.778 | 0.872 |
| RacketSports | 0.901 | 0.816 | 0.803 | 0.783 | 0.829 | 0.842 | 0.842 | 0.809 | 0.803 |
| SelfRegulationSCP1 | 0.836 | 0.802 | 0.829 | 0.836 | 0.853 | 0.836 | 0.867 | 0.816 | 0.853 |
| StandWalkJump | 0.733 | 0.4 | 0.4 | 0.333 | 0.133 | 0.733 | 0.267 | 0.4 | 0.333 |
| UWaveGestureLibrary | 0.881 | 0.878 | 0.847 | 0.891 | 0.875 | 0.9 | 0.825 | 0.859 | 0.866 |
| **Mean** | **0.747** | **0.689** | **0.681** | **0.691** | **0.685** | **0.734** | **0.675** | **0.645** | **0.696** |
| **IMP(%)** | | **8%** | **10%** | **8%** | **9%** | **2%** | **11%** | **16%** | **7%** |

Table 18: **Classification Ablation study (Higher is better).** We report both the mean accuracy and the IMP (%)—the percentage improvement of TSPulse over each dropped component in terms of the mean accuracy.

**Ablation study** As shown in Table 23, removing dual-space reconstruction (i.e. using only time space and ignoring FFT space) during pre-training results in a consistent 8% drop in zero-shot imputation accuracy for hybrid mask set-up. Furthermore, when pre-training is done with only block

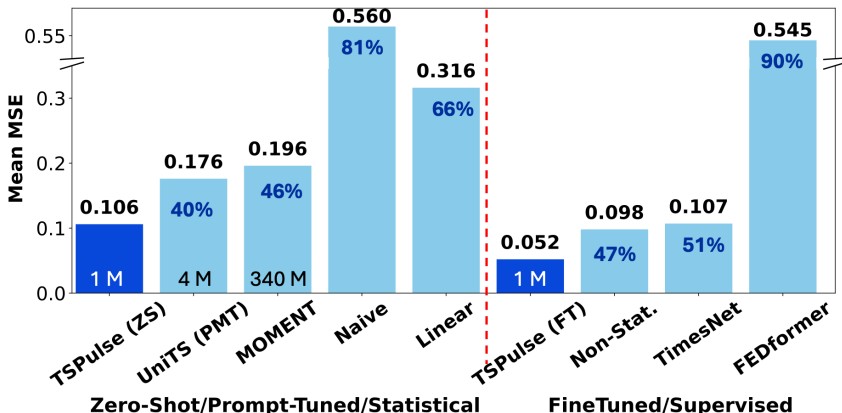

Figure 13: **Block Masking imputation MSE results (lower is better).** Averaged across 6 LTSF benchmark datasets (Wu et al., 2021): ETTh1, ETTh2, ETTm1, ETTm2, Weather, and Electricity, under 4 mask ratios (12.5%, 25%, 37.5%, 50%)

masking (i.e w/o Hybrid pre-training), model suffers a drastic 79% drop hybrid-masked settings, where missingness is irregular and more reflective of real-world scenarios. This sharp contrast underscores the critical role of hybrid-masking during pre-training for achieving robust, generalizable imputation across diverse masking patterns.

| Dataset | Block_Masking | | | | | | | | | Hybrid_Masking | | | | | | | | |
| --- | --- | --- | --- | --- | --- | --- | --- | --- | --- | --- | --- | --- | --- | --- | --- | --- | --- | --- |
| | ZS/PMT/Statistical | | | | | FT/SUP | | | | ZS/PMT/Statistical | | | | | FT/SUP | | | |
| | TSP ZS | UTS PMT | MNT | Naive | Lin. | TSP FT | Non-Stat. | TNT | FED | TSP ZS | UTS PMT | MNT | Naive | Lin. | TSP FT | Non-Stat. | TNT | FED |
| ETTh1 | **0.238** | 0.284 | 0.401 | 1.106 | 0.677 | **0.088** | 0.174 | 0.190 | 0.416 | **0.163** | 0.269 | 0.437 | 0.675 | 0.346 | **0.060** | 0.123 | 0.129 | 0.269 |
| ETTh2 | **0.095** | 0.239 | 0.134 | 0.228 | 0.140 | **0.058** | 0.115 | 0.114 | 1.124 | **0.070** | 0.254 | 0.302 | 0.155 | 0.093 | **0.046** | 0.093 | 0.085 | 0.629 |
| ETTm1 | **0.094** | 0.195 | 0.224 | 0.437 | 0.157 | **0.035** | 0.076 | 0.098 | 0.335 | **0.057** | 0.172 | 0.270 | 0.197 | 0.070 | **0.024** | 0.053 | 0.059 | 0.195 |
| ETTm2 | **0.054** | 0.132 | 0.082 | 0.113 | 0.062 | **0.030** | 0.052 | 0.060 | 0.895 | **0.037** | 0.122 | 0.247 | 0.070 | 0.039 | **0.023** | 0.041 | 0.040 | 0.422 |
| Weather | **0.056** | 0.078 | 0.081 | 0.107 | 0.062 | **0.040** | 0.060 | 0.056 | 0.229 | **0.039** | 0.072 | 0.107 | 0.071 | 0.042 | **0.034** | 0.045 | 0.044 | 0.132 |
| Electricity | **0.097** | 0.126 | 0.252 | 1.371 | 0.797 | **0.060** | 0.114 | 0.126 | 0.271 | **0.077** | 0.130 | 0.293 | 0.808 | 0.378 | **0.049** | 0.104 | 0.122 | 0.236 |
| Mean | **0.106** | 0.176 | 0.196 | 0.560 | 0.316 | **0.052** | 0.098 | 0.107 | 0.545 | **0.074** | 0.170 | 0.276 | 0.329 | 0.161 | **0.039** | 0.076 | 0.08 | 0.314 |
| % IMP. of TSP_ZS | – | 40% | 46% | 81% | 66% | – | – | – | – | – | 56% | 73% | 78% | 54% | – | – | – | – |
| % IMP. of TSP_FT | – | – | – | – | – | 47% | 51% | 90% | – | – | – | – | – | – | 49% | 51% | 88% |

Table 19: **Imputation Result (MSE, Lower is better): Hybrid** and **Block masking** setup averaged across all 4 mask ratios. Lowest MSE score across models is highlighted in **Bold** and second lowest MSE score is underlined.
**Abbreviations:** FT : Finetuned, PMT : Prompt_Tuned, SUP : Supervised, ZS : Zeroshot, TSP : TSPulse, MNT : MOMENT, UTS : UniTS, TNT : TimesNet, Lin. : Linear Interpolation, Non-Stat. : Non-Stationary Transformer, FED : FEDformer. Refer to Table 20 and Table 21 for detailed results

## A.14 SIMILARITY SEARCH

We evaluate the zero-shot similarity search capabilities of **TSPulse**, using its short register embeddings (i.e. zero-shot semantic embeddings) to identify time-series segments that share similar patterns with a given query, even under real-world distortions such as time shifts, magnitude changes, and additive noise. Such similarity search is valuable in practical time-series analysis, where retrieving similar patterns from historical data is often essential for interpretability and post-diagnostics—for example, understanding the context of a detected anomaly or performing root-cause analysis by matching query patterns to historical logs.

This task is particularly challenging because time-series data are typically long and stored as short windows using high-stride sliding windows to reduce indexing memory. This leads to the same underlying pattern appearing at different positions across windows - making exact alignment between query and index samples unlikely. Thus, models must learn robust and distortion-invariant embeddings to enable effective retrieval.

| Dataset | MR | Zeroshot/Prompt_Tuned/Statistical | | | | | | | Finetuned/Supervised | | | |
|---|---|---|---|---|---|---|---|---|---|---|---|---|
| | | TSPulse | UniTS (PMT) | MOMENT | Naive | Linear | Nearest | Cubic | TSPulse | Non-Stationary Transformer | TimesNet | FED-former |
| ETTh1 | 0.125 | **0.146** | 0.220 | 0.324 | 0.608 | 0.284 | 0.388 | 0.491 | **0.042** | 0.089 | 0.098 | 0.180 |
| | 0.250 | **0.155** | 0.250 | 0.387 | 0.648 | 0.321 | 0.425 | 0.618 | **0.051** | 0.113 | 0.120 | 0.241 |
| | 0.375 | **0.168** | 0.287 | 0.468 | 0.696 | 0.365 | 0.471 | 0.791 | **0.065** | 0.135 | 0.127 | 0.298 |
| | 0.500 | **0.183** | 0.319 | 0.568 | 0.749 | 0.414 | 0.525 | 1.011 | **0.081** | 0.156 | 0.169 | 0.358 |
| | Mean | **0.163** | 0.269 | 0.437 | 0.675 | 0.346 | 0.452 | 0.728 | **0.060** | 0.123 | 0.129 | 0.269 |
| ETTh2 | 0.125 | **0.065** | 0.151 | 0.132 | 0.143 | 0.084 | 0.114 | 0.296 | **0.041** | 0.081 | 0.071 | 0.375 |
| | 0.250 | **0.068** | 0.199 | 0.202 | 0.150 | 0.090 | 0.119 | 0.358 | **0.043** | 0.088 | 0.080 | 0.553 |
| | 0.375 | **0.072** | 0.300 | 0.332 | 0.159 | 0.095 | 0.124 | 0.430 | **0.047** | 0.097 | 0.088 | 0.716 |
| | 0.500 | **0.077** | 0.364 | 0.542 | 0.169 | 0.103 | 0.132 | 0.505 | **0.053** | 0.104 | 0.100 | 0.873 |
| | Mean | **0.070** | 0.254 | 0.302 | 0.155 | 0.093 | 0.122 | 0.397 | **0.046** | 0.093 | 0.085 | 0.629 |
| ETTm1 | 0.125 | **0.049** | 0.123 | 0.156 | 0.154 | 0.056 | 0.091 | 0.173 | **0.018** | 0.040 | 0.048 | 0.109 |
| | 0.250 | **0.053** | 0.156 | 0.218 | 0.179 | 0.064 | 0.101 | 0.210 | **0.022** | 0.048 | 0.058 | 0.169 |
| | 0.375 | **0.058** | 0.187 | 0.299 | 0.209 | 0.073 | 0.114 | 0.254 | **0.026** | 0.057 | 0.061 | 0.225 |
| | 0.500 | **0.066** | 0.222 | 0.407 | 0.248 | 0.086 | 0.131 | 0.305 | **0.031** | 0.067 | 0.070 | 0.275 |
| | Mean | **0.057** | 0.172 | 0.270 | 0.197 | 0.070 | 0.109 | 0.235 | **0.024** | 0.053 | 0.059 | 0.195 |
| ETTm2 | 0.125 | **0.034** | 0.096 | 0.085 | 0.063 | 0.036 | 0.049 | 0.115 | **0.020** | 0.035 | 0.036 | 0.214 |
| | 0.250 | **0.036** | 0.092 | 0.149 | 0.067 | 0.038 | 0.052 | 0.139 | **0.022** | 0.039 | 0.038 | 0.353 |
| | 0.375 | **0.038** | 0.126 | 0.272 | 0.072 | 0.040 | 0.055 | 0.171 | **0.024** | 0.042 | 0.041 | 0.492 |
| | 0.500 | **0.041** | 0.172 | 0.480 | 0.079 | 0.044 | 0.058 | 0.207 | **0.026** | 0.047 | 0.045 | 0.628 |
| | Mean | **0.037** | 0.122 | 0.247 | 0.070 | 0.039 | 0.054 | 0.158 | **0.023** | 0.041 | 0.040 | 0.422 |
| Weather | 0.125 | **0.036** | 0.059 | 0.069 | 0.065 | 0.038 | 0.053 | 0.131 | **0.036** | 0.040 | 0.039 | 0.076 |
| | 0.250 | **0.038** | 0.070 | 0.086 | 0.068 | 0.040 | 0.055 | 0.163 | **0.033** | 0.043 | 0.042 | 0.106 |
| | 0.375 | **0.040** | 0.074 | 0.115 | 0.072 | 0.043 | 0.057 | 0.190 | **0.033** | 0.045 | 0.045 | 0.148 |
| | 0.500 | **0.042** | 0.085 | 0.158 | 0.0780 | 0.046 | 0.060 | 0.228 | **0.035** | 0.050 | 0.048 | 0.198 |
| | Mean | **0.039** | 0.072 | 0.107 | 0.071 | 0.042 | 0.056 | 0.178 | **0.034** | 0.045 | 0.044 | 0.132 |
| Electricity | 0.125 | **0.070** | 0.109 | 0.181 | 0.712 | 0.289 | 0.417 | 0.434 | **0.043** | 0.100 | 0.120 | 0.213 |
| | 0.250 | **0.074** | 0.123 | 0.235 | 0.773 | 0.344 | 0.468 | 0.573 | **0.047** | 0.102 | 0.123 | 0.229 |
| | 0.375 | **0.078** | 0.136 | 0.317 | 0.837 | 0.405 | 0.528 | 0.755 | **0.050** | 0.105 | 0.122 | 0.244 |
| | 0.500 | **0.085** | 0.151 | 0.439 | 0.910 | 0.475 | 0.601 | 1.008 | **0.054** | 0.107 | 0.124 | 0.259 |
| | Mean | **0.077** | 0.130 | 0.293 | 0.808 | 0.378 | 0.504 | 0.693 | **0.049** | 0.104 | 0.122 | 0.236 |

Table 20: Full **Imputation Results (MSE, Lower is better):** Hybrid Masking setup. Lowest MSE score is indicated in **Bold** and second lowest MSE score is Underlined

**Datasets** We construct two benchmark datasets—one synthetic and one real-world, which are indexed for retrieval:

- **Synthetic Dataset:** We begin with 8 distinct base patterns (see Table 27) and form 56 unique combinations by pairing them through addition and multiplication operations ($_8C_2 \times 2$). Each combination is then varied by a frequency parameter $f = 1, \ldots, 10$. Light augmentations—1% Gaussian noise and 1% magnitude scaling—are applied to each sample, resulting in three variants per time-series. This process yields a total of $56 \times 10 \times 3 = 1,680$ indexed time-series segments.

- **Real Dataset:** We use the UCR Archive (Chen et al., 2015), which consists of 128 time-series classification datasets. For each class in each dataset, we randomly sample 10 time-series segments of length 512. The datasets with fewer than 10 samples in a class are excluded. This results in a total of 5,960 indexed time-series segments.

**Query Generation** To construct the query sets, we apply stronger augmentations to the indexed samples to simulate real-world distortions and test the robustness of the learned embeddings. Specifically, each query is created by applying a combination of:

- within $\pm 20\%$ random time shift,
- within $\pm 20\%$ random scaling transformation, and
- Gaussian noise with standard deviation of $10\%$ magnitude of the scaled time-series data.

| Dataset | MR | Zeroshot/Prompt_Tuned/Statistical | | | | | | | Finetuned/Supervised | | | |
|---|---|---|---|---|---|---|---|---|---|---|---|---|
| | | TSPulse | UniTS (PMT) | MOMENT | Naive | Linear | Nearest | Cubic | TSPulse | Non-Stationary Transformer | TimesNet | FED-former |
| ETTh1 | 0.125 | **0.209** | 0.239 | 0.302 | 1.035 | 0.530 | 0.645 | 1.100 | **0.055** | 0.115 | 0.138 | 0.264 |
| | 0.250 | **0.225** | 0.269 | 0.355 | 1.092 | 0.633 | 0.760 | 1.819 | **0.070** | 0.156 | 0.157 | 0.377 |
| | 0.375 | **0.246** | 0.299 | 0.430 | 1.131 | 0.729 | 0.875 | 2.978 | **0.098** | 0.192 | 0.201 | 0.469 |
| | 0.500 | **0.272** | 0.331 | 0.518 | 1.165 | 0.816 | 0.987 | 5.101 | **0.129** | 0.234 | 0.263 | 0.554 |
| | Mean | **0.238** | 0.284 | 0.401 | 1.106 | 0.677 | 0.817 | 2.749 | **0.088** | 0.174 | 0.19 | 0.416 |
| ETTh2 | 0.125 | **0.081** | 0.150 | 0.105 | 0.201 | 0.117 | 0.147 | 0.640 | **0.050** | 0.093 | 0.089 | 0.556 |
| | 0.250 | **0.088** | 0.195 | 0.121 | 0.218 | 0.132 | 0.163 | 0.997 | **0.055** | 0.109 | 0.104 | 0.948 |
| | 0.375 | **0.098** | 0.250 | 0.142 | 0.235 | 0.148 | 0.182 | 1.640 | **0.059** | 0.121 | 0.123 | 1.320 |
| | 0.500 | **0.113** | 0.360 | 0.168 | 0.255 | 0.165 | 0.202 | 2.871 | **0.068** | 0.136 | 0.140 | 1.673 |
| | Mean | **0.095** | 0.239 | 0.134 | 0.228 | 0.140 | 0.173 | 1.537 | **0.058** | 0.115 | 0.114 | 1.124 |
| ETTm1 | 0.125 | **0.068** | 0.137 | 0.133 | 0.279 | 0.088 | 0.140 | 0.373 | **0.023** | 0.047 | 0.060 | 0.181 |
| | 0.250 | **0.081** | 0.175 | 0.179 | 0.369 | 0.120 | 0.181 | 0.595 | **0.028** | 0.062 | 0.082 | 0.280 |
| | 0.375 | **0.100** | 0.206 | 0.246 | 0.480 | 0.170 | 0.238 | 0.976 | **0.038** | 0.085 | 0.106 | 0.384 |
| | 0.500 | **0.128** | 0.262 | 0.339 | 0.620 | 0.248 | 0.325 | 1.732 | **0.052** | 0.107 | 0.146 | 0.493 |
| | Mean | **0.094** | 0.195 | 0.224 | 0.437 | 0.157 | 0.221 | 0.919 | **0.035** | 0.076 | 0.098 | 0.335 |
| ETTm2 | 0.125 | **0.046** | 0.099 | 0.061 | 0.092 | 0.051 | 0.066 | 0.251 | **0.022** | 0.041 | 0.046 | 0.448 |
| | 0.250 | **0.050** | 0.113 | 0.072 | 0.104 | 0.057 | 0.074 | 0.393 | **0.027** | 0.047 | 0.057 | 0.712 |
| | 0.375 | **0.056** | 0.138 | 0.088 | 0.119 | 0.065 | 0.083 | 0.630 | **0.030** | 0.055 | 0.068 | 1.015 |
| | 0.500 | **0.065** | 0.179 | 0.108 | 0.139 | 0.076 | 0.096 | 1.094 | **0.039** | 0.064 | 0.070 | 1.404 |
| | Mean | **0.054** | 0.132 | 0.082 | 0.113 | 0.062 | 0.080 | 0.592 | **0.030** | 0.052 | 0.060 | 0.895 |
| Weather | 0.125 | **0.048** | 0.065 | 0.063 | 0.090 | 0.053 | 0.068 | 0.281 | **0.034** | 0.051 | 0.046 | 0.133 |
| | 0.250 | **0.053** | 0.075 | 0.072 | 0.099 | 0.058 | 0.074 | 0.447 | **0.038** | 0.056 | 0.052 | 0.197 |
| | 0.375 | **0.058** | 0.081 | 0.086 | 0.112 | 0.065 | 0.082 | 0.731 | **0.042** | 0.062 | 0.057 | 0.262 |
| | 0.500 | **0.066** | 0.093 | 0.103 | 0.128 | 0.074 | 0.093 | 1.219 | **0.047** | 0.073 | 0.067 | 0.324 |
| | Mean | **0.056** | 0.078 | 0.081 | 0.107 | 0.062 | 0.079 | 0.669 | **0.04** | 0.060 | 0.056 | 0.229 |
| Electricity | 0.125 | **0.084** | 0.110 | 0.159 | 1.257 | 0.571 | 0.718 | 1.018 | **0.053** | 0.110 | 0.121 | 0.238 |
| | 0.250 | **0.090** | 0.121 | 0.197 | 1.351 | 0.732 | 0.880 | 1.741 | **0.057** | 0.114 | 0.124 | 0.262 |
| | 0.375 | **0.100** | 0.131 | 0.264 | 1.416 | 0.878 | 1.050 | 2.924 | **0.062** | 0.114 | 0.128 | 0.283 |
| | 0.5 | **0.115** | 0.141 | 0.388 | 1.46 | 1.005 | 1.212 | 5.008 | **0.068** | 0.118 | 0.131 | 0.302 |
| | Mean | **0.097** | 0.126 | 0.252 | 1.371 | 0.797 | 0.965 | 2.673 | **0.060** | 0.114 | 0.126 | 0.271 |

Table 21: Full **Imputation Results (MSE, Lower is better):** Block-wise masking setup. Lowest MSE score is indicated in **Bold** and second lowest MSE score is Underlined

| Model Variant | MSE (↓) | IMP(%) |
|---|---|---|
| **Irregular Hybrid Masking Eval** | | |
| TSPulse | **0.074** | |
| w/o Dual-Space | 0.081 | 8% ↓ |
| w/o Hybrid Pretraining | 0.354 | 79% ↓ |
| **Regular Block Masking Eval** | | |
| TSPulse | 0.106 | |
| w/o Dual-Space | 0.115 | 8% ↓ |
| w/o Hybrid Pretraining | **0.098** | 7.5% ↑ |

Table 22: **Imputation–Ablation Study Summary, MSE, Lower is better**. Detailed results in Table 23

This setup serves two purposes: (i) it evaluates whether embeddings are invariant to temporal misalignment and other distortions and (ii) it allows for automatic ground-truth labeling since each query is derived from a known indexed sample. The same augmentation strategy is applied to both the synthetic and real datasets, ensuring a consistent evaluation framework. These challenging augmentations make the retrieval task more realistic and better reflect practical use cases in time-series analysis.

**Tasks**   We define two types of retrieval tasks to evaluate semantic similarity at different levels:

| Dataset | Block_Masking | | | Hybrid_Masking | | |
|---|---|---|---|---|---|---|
| | TSPulse w/o Dual-Space | TSPulse | TSPulse w/o Hybrid Pre-training | TSPulse w/o Dual-Space | TSPulse | TSPulse w/o Hybrid Pre-training |
| ETTh1 | 0.251 | 0.238 | **0.220** | 0.173 | **0.163** | 0.539 |
| ETTh2 | 0.099 | 0.095 | **0.091** | 0.074 | **0.070** | 0.231 |
| ETTm1 | 0.105 | 0.094 | **0.088** | 0.063 | **0.057** | 0.485 |
| ETTm2 | 0.056 | 0.054 | **0.053** | 0.039 | **0.037** | 0.163 |
| Weather | 0.058 | 0.056 | **0.055** | 0.041 | **0.039** | 0.155 |
| Electricity | 0.120 | 0.097 | **0.083** | 0.094 | **0.077** | 0.552 |
| Mean | 0.115 | 0.106 | **0.098** | 0.081 | **0.074** | 0.354 |

Table 23: **Imputation Result (MSE, Lower is better)**: Block and Hybrid masking setup averaged across all 4 different mask ratios for different variants of TSPulse. Lowest MSE score across variants is highlighted in **Bold** and second lowest MSE score is Underlined. More Detailed results in Table 24.

| Dataset | Mask_Ratio | Block Masking | | | Hybrid Masking | | |
|---|---|---|---|---|---|---|---|
| | | TSPulse w/o Dual_space | TSPulse | TSPulse w/o Hybrid Pre-training | TSPulse w/o Dual_space | TSPulse | TSPulse w/o Hybrid Pre-training |
| ETTh1 | 0.125 | 0.227 | 0.209 | **0.189** | 0.158 | **0.146** | 0.498 |
| | 0.250 | 0.241 | 0.225 | **0.208** | 0.166 | **0.155** | 0.528 |
| | 0.375 | 0.258 | 0.246 | **0.230** | 0.177 | **0.168** | 0.553 |
| | 0.500 | 0.279 | 0.272 | **0.255** | 0.190 | **0.183** | 0.578 |
| | Mean | 0.251 | 0.238 | **0.220** | 0.173 | **0.163** | 0.539 |
| ETTh2 | 0.125 | 0.086 | 0.081 | **0.077** | 0.069 | **0.065** | 0.216 |
| | 0.250 | 0.093 | 0.088 | **0.084** | 0.072 | **0.068** | 0.226 |
| | 0.375 | 0.102 | 0.098 | **0.094** | 0.075 | **0.072** | 0.235 |
| | 0.500 | 0.115 | 0.113 | **0.107** | 0.080 | **0.077** | 0.247 |
| | Mean | 0.099 | 0.095 | **0.091** | 0.074 | **0.070** | 0.231 |
| ETTm1 | 0.125 | 0.078 | 0.068 | **0.065** | 0.055 | **0.049** | 0.449 |
| | 0.250 | 0.091 | 0.081 | **0.076** | 0.060 | **0.053** | 0.470 |
| | 0.375 | 0.110 | 0.100 | **0.093** | 0.065 | **0.058** | 0.496 |
| | 0.500 | 0.139 | 0.128 | **0.119** | 0.072 | **0.066** | 0.526 |
| | Mean | 0.105 | 0.094 | **0.088** | 0.063 | **0.057** | 0.485 |
| ETTm2 | 0.125 | 0.048 | 0.046 | **0.044** | 0.036 | **0.034** | 0.154 |
| | 0.250 | 0.052 | 0.050 | **0.049** | 0.037 | **0.036** | 0.159 |
| | 0.375 | 0.058 | 0.056 | **0.055** | 0.040 | **0.038** | 0.166 |
| | 0.500 | 0.067 | 0.065 | **0.063** | 0.042 | **0.041** | 0.174 |
| | Mean | 0.056 | 0.054 | **0.053** | 0.039 | **0.037** | 0.163 |
| Weather | 0.125 | 0.051 | **0.048** | **0.048** | 0.038 | **0.036** | 0.146 |
| | 0.250 | 0.055 | 0.053 | **0.051** | 0.039 | **0.038** | 0.151 |
| | 0.375 | 0.060 | 0.058 | **0.057** | 0.042 | **0.040** | 0.157 |
| | 0.500 | 0.067 | 0.066 | **0.063** | 0.044 | **0.042** | 0.165 |
| | Mean | 0.058 | 0.056 | **0.055** | 0.041 | **0.039** | 0.155 |
| Electricity | 0.125 | 0.105 | 0.084 | **0.069** | 0.085 | **0.070** | 0.488 |
| | 0.250 | 0.113 | 0.090 | **0.076** | 0.090 | **0.074** | 0.539 |
| | 0.375 | 0.124 | 0.100 | **0.086** | 0.096 | **0.078** | 0.575 |
| | 0.500 | 0.138 | 0.115 | **0.102** | 0.104 | **0.085** | 0.607 |
| | Mean | 0.120 | 0.097 | **0.083** | 0.094 | **0.077** | 0.552 |

Table 24: Full **Imputation Results (MSE, Lower is better):** Comparison of three variants of TSPulse **(a) TSPulse w/o Dual_Space** : TSPulse model only pre-trained with time domain input **(b) TSPulse** : TSPulse model pre-trained with both time and frequency patches **(c) TSPulse w/o Hybrid_Masking** : TSPulse model pre-trained with block masking

- **Family Match:** Measures coarse-grained retrieval—i.e., finding samples with the same high-level structure. For synthetic data, each of the 56 pattern combinations defines a class. For real data, each dataset name is treated as a class.

- **Fine-Grained Match:** Measures fine-grained retrieval—i.e., In synthetic data, each pattern-frequency combination ($56 \times 10 = 560$) is treated as a class. In real data, each (dataset name, class index) pair defines a unique class, totaling 596.

**Implementation & Evaluation Metrics**    We compute semantic similarity using Euclidean distance between embeddings from the pre-trained models. We use the *Faiss* library (Douze et al., 2024) to index embeddings. Performance is evaluated using standard retrieval metrics applied to the top-3 retrieved candidates:

| Dataset | Mask_Ratio | MOMENT | | UniTS_PMT | |
|---|---|---|---|---|---|
| | | Heterogenous | Only_Zeros | Heterogenous | Only_Zeros |
| ETTh1 | 0.125 | 0.324 | **0.315** | **0.220** | **0.220** |
| | 0.250 | **0.387** | 0.391 | 0.251 | **0.250** |
| | 0.375 | **0.468** | 0.487 | 0.288 | **0.287** |
| | 0.500 | **0.568** | 0.599 | 0.322 | **0.319** |
| | Mean | **0.437** | 0.448 | 0.27 | **0.269** |
| ETTh2 | 0.125 | **0.132** | 0.256 | **0.150** | 0.151 |
| | 0.250 | **0.202** | 0.441 | 0.211 | **0.199** |
| | 0.375 | **0.332** | 0.649 | 0.326 | **0.300** |
| | 0.500 | **0.542** | 0.939 | 0.371 | **0.364** |
| | Mean | **0.302** | 0.571 | 0.264 | **0.254** |
| ETTm1 | 0.125 | **0.156** | 0.173 | **0.123** | **0.123** |
| | 0.250 | **0.218** | 0.251 | **0.156** | **0.156** |
| | 0.375 | **0.299** | 0.344 | **0.187** | **0.187** |
| | 0.500 | **0.407** | 0.462 | 0.226 | **0.222** |
| | Mean | **0.270** | 0.308 | 0.173 | **0.172** |
| ETTm2 | 0.125 | **0.085** | 0.217 | **0.096** | **0.096** |
| | 0.250 | **0.149** | 0.405 | 0.096 | **0.092** |
| | 0.375 | **0.272** | 0.605 | 0.136 | **0.126** |
| | 0.500 | **0.48** | 0.895 | 0.186 | **0.172** |
| | Mean | **0.247** | 0.530 | 0.129 | **0.122** |
| Weather | 0.125 | **0.069** | 0.090 | **0.059** | **0.059** |
| | 0.25 | **0.086** | 0.124 | **0.070** | **0.070** |
| | 0.375 | **0.115** | 0.166 | 0.075 | **0.074** |
| | 0.500 | **0.158** | 0.223 | 0.087 | **0.085** |
| | Mean | **0.107** | 0.151 | 0.073 | **0.072** |
| Electricity | 0.125 | **0.181** | 0.189 | 0.110 | **0.109** |
| | 0.250 | **0.235** | 0.252 | 0.124 | **0.123** |
| | 0.375 | **0.317** | 0.344 | 0.137 | **0.136** |
| | 0.500 | **0.439** | 0.472 | 0.153 | **0.151** |
| | Mean | **0.293** | 0.314 | 0.131 | **0.130** |

Table 25: Full **Imputation Results (MSE, Lower is better)**: MOMENT & UniTS_PMT in **hybrid** setup. Heterogenous : Used the learned mask token to replace the fully missing patches and zeros at other missing positions, Only_Zeros : Using zeros as input to the model at every missing position.

- **PREC@3:** Precision at top-3.
- **MRR@3:** Mean Reciprocal Rank at top-3.
- **AP@3:** Average Precision at top-3.
- **NDCG@3:** Normalized Discounted Cumulative Gain at top-3.

We evaluate the inference time of each model. Specifically, we set the batch size to $1,024$ and accumulate the forward computation time of the model over the dataset. The measured computation time is divided by the size of the dataset to derive the per-sample computation time.

**Results** Figure 7 presents a comparative evaluation of TSPulse's similarity search performance against the zero-shot embeddings produced by two recent pre-trained models: MOMENT and Chronos. To ensure a fair and resource-efficient comparison, we use their smallest available variants, aligning closely with TSPulse in embedding dimensionality and enabling comparable indexing throughput.

As shown in the figure, TSPulse consistently outperforms both baselines across retrieval tasks. For the **Family Match** task, which assesses the model's ability to retrieve semantically similar patterns, TSPulse achieves over 25% higher accuracy than MOMENT. In the more challenging **Fine-Grained Match** task, which requires precise pattern localization, TSPulse demonstrates a 40% performance gain over MOMENT. TSPulse also outperforms Chronos by +100% in both settings.

In addition to accuracy gains, TSPulse also offers substantial efficiency benefits. Its embeddings are $2\times$ smaller than those of MOMENT and Chronos, leading to faster search and reduced memory

overhead. In deployment benchmarks, TSPulse enables 10–100× faster inference on CPU and 9–15× faster inference on GPU, owing to its lightweight architecture and low memory footprint. Importantly, TSPulse achieves these results while being over 40× smaller in model size, making it highly suitable for real-time, resource-constrained environments.

Tables 28 and 29 summarize the scores of PREC@3, AP@3, NDCG@3, and MRR@3. The average scores in Table 28 are plotted in Figure 7.

**Impact of Different Model Configuration**    We conducted the experiments to investigate the impact of different model configurations, as summarized in the table below. Removing Hybrid pre-training (i.e., using only block masking) results in a 23.1% drop in PREC@3 performance. Furthermore, excluding the register embeddings and relying solely on the Time or FFT embeddings leads to even greater reductions of 51.3% and 68.8%, respectively. These results emphasize the importance of both hybrid pre-training and register embeddings in achieving high retrieval accuracy.

| Model Variant | PREC@3 | | MRR@3 | |
|---|---|---|---|---|
| **TSPulse** | **0.645** | | **0.856** | |
| w/o Hybrid PT | 0.496 | 23.1%↓ | 0.669 | 21.8%↓ |
| w/o register (Time emb.) | 0.314 | 51.3%↓ | 0.439 | 48.7%↓ |
| w/o register (FFT emb.) | 0.201 | 68.8%↓ | 0.274 | 68.0%↓ |

Table 26: Impact of Different Model Configuration

**Robustness to Augmentation Complexity**    To further evaluate the robustness of pre-trained models in the similarity search task, we systematically vary the intensity of augmentations used to generate the query samples. Specifically, we introduce a scaling factor $s \in \{0, 10, 20, 30, 40, 50\}$, which controls the strength of three types of distortions: time shifts, magnitude scaling, and additive noise. For example, when $s = 10$, a 10% random temporal shift, 10% scaling perturbation, and 10% additive Gaussian noise are applied to each query. The noise level is capped at 10% across all values of $s$ to maintain a realistic noise regime, as extreme noise rarely dominates in practical time-series settings.

The motivation behind increasing $s$ lies in simulating more challenging retrieval scenarios. In time-series data, even small time shifts or scaling changes can result in significantly different signal appearances, despite underlying similarity in semantics. By gradually increasing $s$, we create a controlled framework to stress-test the distortion invariance of learned embeddings. The setting where $s = 0$ corresponds to a sanity check: since the query is identical to an indexed sample, a high retrieval score is expected from any well-trained model.

Figure 14 presents the retrieval performance of TSPulse, MOMENT, and Chronos across different augmentation levels. As expected, all models perform well at $s = 0$, but performance declines with increasing augmentation strength. Notably, TSPulse exhibits significantly greater resilience, maintaining higher scores across all values of $s$. Compared to MOMENT and Chronos, the performance of TSPulse degrades more slowly, demonstrating superior robustness to distortions in temporal alignment, signal scaling, and additive noise. These results highlight the effectiveness of TSPulse's hybrid masking pre-training and register-token embeddings in capturing stable, semantically meaningful representations under realistic, noisy retrieval conditions.

## A.15    UNIFIED PRE-TRAINED MODEL ACROSS MULTIPLE DOWNSTREAM TASKS

While the primary design of **TSPulse** adopts task-specialized pre-training to maximize downstream accuracy, we additionally evaluate a unified setting in which a single pre-trained checkpoint is used across all downstream tasks without task-specific specialization.

### A.15.1    EXPERIMENTAL SETUP

In this setting, we select the pre-trained variant originally optimized for imputation and retrieval/search tasks and directly evaluate it on: (i) classification, (ii) anomaly detection (AD), (iii) search / retrieval, and (iv) imputation.

| Base pattern name | Base pattern equation |
|---|---|
| sin | $x_t = \sin(b_t)$ |
| modcos | $x_t = \cos(b_t) \cdot \sin(b_t/2)$ |
| square-modcos | $x_t = \mathrm{sign}(\sin(b_t)) \cdot |\cos(2b_t)|$ |
| gaussian-spike | $x_t = \exp(-40(c_t - f/2)^2)$ |
| impulse | $x_t = \begin{cases} 1 & \text{if } \mathrm{mod}(t, 10f) = 0 \\ 0 & \text{otherwise} \end{cases}$ |
| randwalk | $x_t = f + \sum_{t'=0}^{t} \varepsilon_{t'}$ |
| sincos | $x_t = \sin(b_t) \cdot \cos(2 * b_t)$ |
| tanhmix | $x_t = \tanh(\sin(3 * b_t)) + 0.2\varepsilon_t$ |

Table 27: Eight base patterns for synthetic data. $b_t = 2\pi t f /512, t = 0, 1, \ldots, 511$, sign is the sign function, $\varepsilon$ is drawn from the standard Gaussian distribution, i.e., $\varepsilon \sim N(0, 1)$, and $\mathrm{mod}$ is the modulo operation.

| | Family match | | | | | | Fine-grained match | | | | | |
|---|---|---|---|---|---|---|---|---|---|---|---|---|
| | PREC@3 | | | MRR@3 | | | PREC@3 | | | MRR@3 | | |
| | $T_{\text{SPulse}}$ | MOMENT | $C_{\text{hronos}}$ | T | M | C | T | M | C | T | M | C |
| Real | 0.645 | 0.389 | 0.116 | 0.856 | 0.491 | 0.144 | 0.474 | 0.216 | 0.057 | 0.750 | 0.335 | 0.087 |
| Synth. | 0.711 | 0.674 | 0.352 | 0.713 | 0.679 | 0.361 | 0.695 | 0.615 | 0.295 | 0.697 | 0.619 | 0.304 |
| Avg. | 0.678 | 0.532 | 0.234 | 0.784 | 0.585 | 0.252 | 0.584 | 0.415 | 0.176 | 0.723 | 0.477 | 0.195 |

Table 28: **Similarity Search.** PREC@3 and MRR@3 on Synthetic and Real data. Higher is better.

Thus, all four downstream tasks share a single pre-trained backbone and checkpoint. This experiment isolates the impact of task specialization and evaluates the intrinsic generalization ability of the learned representation.

### A.15.2 RESULTS

Table 30 summarizes the comparative performance between: (1) task-specialized TSPulse, (2) a single unified TSPulse model (one checkpoint for all tasks), and (3) the best reported baseline per task. The single unified model remains highly competitive, typically ranking within the top one or two positions across tasks. This demonstrates that TSPulse maintains strong generalization even without task specialization. Enabling task-specific loss reweighting further yields additional strong performance gains.

### A.16 ROBUSTNESS UNDER MAR AND MNAR MISSINGNESS PATTERNS

In this section, we further evaluate whether TSPulse remains robust under empirically grounded missingness patterns, including Missing At Random (MAR) and Missing Not At Random (MNAR) settings Rubin (1976). Before describing these experiments, we first clarify the motivation behind the hybrid masking design.

### A.16.1 MOTIVATION: PRE-TRAINING MASK BIAS

While hybrid masking may appear to be a simple combination of point and block masking, its purpose is to address a critical *pre-training mask bias* present in existing imputation-oriented foundation models. Many prior approaches rely exclusively on fixed-length block masking during pre-training.

As shown in Appendix Table 19, popular pretrained models perform well under fixed-length block masking and substantially outperform linear interpolation in that setting. However, when evaluated under irregular or partially missing spans—where only a portion of a block is missing—the expected improvement (due to shorter gaps) does not occur. Instead, performance often degrades below that of linear interpolation.

Further investigation reveals that these models impute effectively primarily when the inference-time missing-span length matches the mask length used during pre-training. Since real-world missingness

| | Family match | | | | | | Fine-grained match | | | | | |
| | AP@3 | | | NDCG@3 | | | AP@3 | | | NDCG@3 | | |
| | $T_{SPulse}$ | $M_{OMENT}$ | $C_{hronos}$ | T | M | C | T | M | C | T | M | C |
|---|---|---|---|---|---|---|---|---|---|---|---|---|
| Real | 0.619 | 0.366 | 0.105 | 0.683 | 0.403 | 0.118 | 0.447 | 0.195 | 0.050 | 0.525 | 0.234 | 0.061 |
| Synth. | 0.710 | 0.672 | 0.348 | 0.711 | 0.675 | 0.353 | 0.694 | 0.613 | 0.290 | 0.695 | 0.615 | 0.295 |
| Avg. | 0.664 | 0.519 | 0.226 | 0.697 | 0.539 | 0.235 | 0.570 | 0.404 | 0.170 | 0.610 | 0.425 | 0.178 |

Table 29: **Similarity Search.** AP@3 and NDCG@3 on Synthetic and Real data. Higher is better.

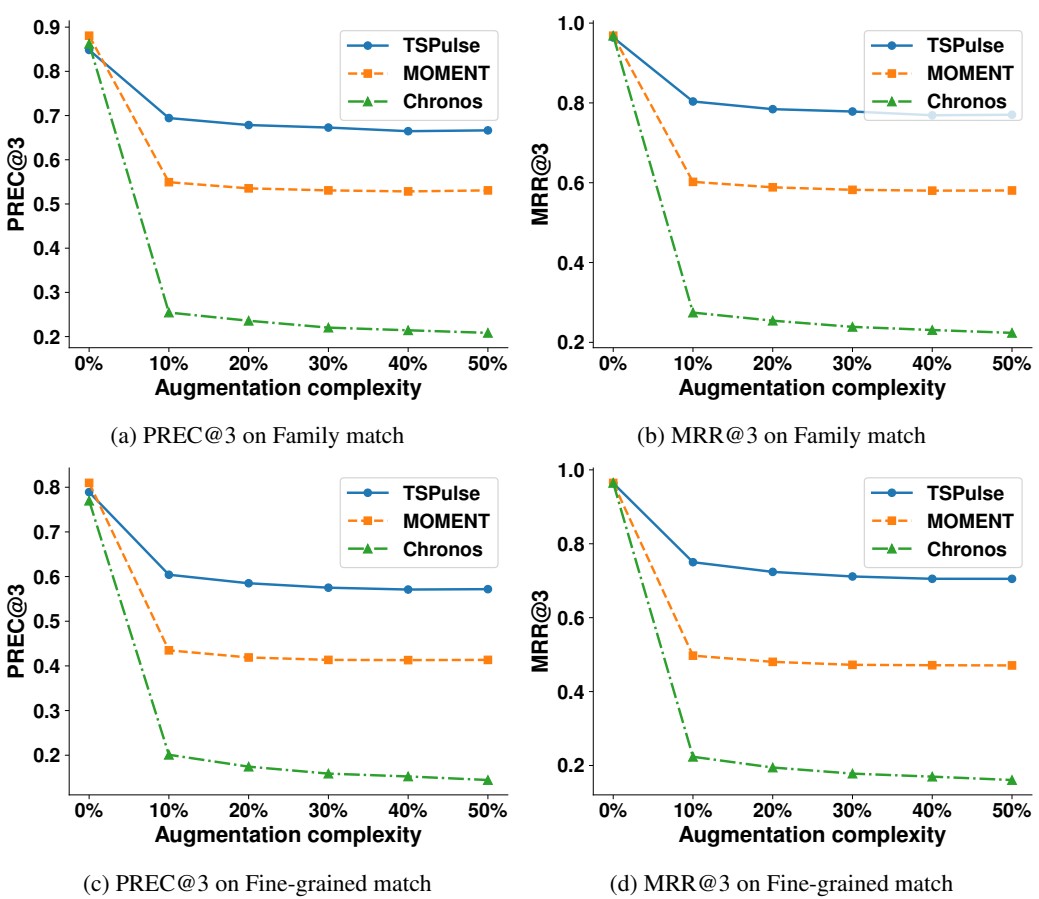

(a) PREC@3 on Family match

(b) MRR@3 on Family match

(c) PREC@3 on Fine-grained match

(d) MRR@3 on Fine-grained match

Figure 14: TSPulse Similarity Search Ablation Study

patterns are highly variable and unpredictable, this creates a systematic mismatch between pre-training assumptions and deployment conditions. We refer to this phenomenon as *pre-training mask bias*.

TSPulse addresses this issue through hybrid masking, which exposes the model to a wide spectrum of variable-length missingness patterns during pre-training rather than a single fixed-length mask. Variable-length masks more closely reflect practical missingness patterns. For clarity, we refer to these as *variable-length missingness patterns*.

### A.16.2 MAR AND MNAR SCENARIO DESIGN

MAR (missing at random): If the probability of a point being missing is the same only within groups defined by the observed data, then the data are missing at random (MAR).

MNAR (missing not at random): If the probability of a point being missing varies for reasons that are unknown (unobserved) to us, then the data are missing not at random (MNAR).

Table 30: Performance comparison between task-specialized and unified TSPulse variant across downstream tasks.

| Task | Metric | Task-Specialized | Unified Model | Best Baseline |
|------|--------|------------------|---------------|---------------|
| Classification | Accuracy ↑ | 0.73 | 0.71 | 0.70 (VQShape) |
| AD (Univariate) | VUS-PR ↑ | 0.48 | 0.42 | 0.42 (Sub-PCA) |
| Search (Family Match) | PREC@3 ↑ | 0.68 | 0.68 | 0.53 (MOMENT) |
| Search (Fine-grained Match) | PREC@3 ↑ | 0.58 | 0.58 | 0.42 (MOMENT) |
| Imputation (Block) | MSE ↓ | 0.106 | 0.106 | 0.176 (UniTS) |
| Imputation (Hybrid) | MSE ↓ | 0.074 | 0.074 | 0.161 (Linear Interpolation) |

Below, we explain several real-world MAR and MNAR situations and how we simulate them. All evaluations are done on ETTh1 from the LTF benchmark, a real multivariate hourly electricity-transformer dataset.

**MAR Scenarios**

- **S1 – Peak-hour sensor overload**
  Real world: Sensors often fail more during peak load hours.
  Simulation: In each 24-point block, the middle 12 points have a missing probability of 0.6, and the remaining points have 0.05.

- **S2 – Cross-channel failure due to high readings**
  Real world: High readings in one channel (e.g., very high temperature) can cause other sensors to fail.
  Simulation: Missingness in a channel increases when another channel's value lies in the top 10%.

- **S3 – Cross-channel failure due to volatility**
  Real world: Sudden fluctuations in one variable (e.g., power/voltage instability) can disrupt other sensors.
  Simulation: Missingness in a channel increases when another channel shows high short-term variance.

**MNAR Scenarios**

- **S1 – Sensor breakdown under extreme values**
  Real world: A sensor becomes unreliable as a measurement (e.g., temperature) crosses high thresholds.
  Simulation: Missingness increases with the magnitude of a channel's values.

- **S2 – Latent stress, combining load levels + rate of change**
  Real world: Failures happen when a system experiences both high load and rapid fluctuations.
  Simulation: A latent variable combining one channel's value and another channel's derivative determines missingness.

- **S3 – System-wide stress causing multi-sensor degradation**
  Real world: Many sensors degrade together when the entire system's operational load or stress increases.
  Simulation: Missingness increases with the moving-average trend of the sum of all non-temperature channels.

- **S4 – Time-dependent degradation**
  Real world: Sensors degrade and drop data more frequently over time.
  Simulation: Missingness probability increases gradually with time.

### A.16.3 DISCUSSION

Across all MAR and MNAR settings, TSPulse consistently outperforms MOMENT. These results support the hypothesis that hybrid masking improves robustness under heterogeneous and diverse missingness conditions. By exposing the model to variable-length missing patterns during pre-training,

Table 31: Robustness evaluation under MAR and MNAR missingness (MSE). Lower is better.

|       | Scenario | TSPulse | MOMENT |       | Scenario | TSPulse | MOMENT |
|-------|----------|---------|--------|-------|----------|---------|--------|
| **MAR** | S1 | 0.147 | 0.575 | **MNAR** | S1 | 0.334 | 0.760 |
|       | S2 | 0.154 | 0.501 |       | S2 | 0.134 | 0.552 |
|       | S3 | 0.138 | 0.402 |       | S3 | 0.141 | 0.498 |
|       |          |         |        |       | S4 | 0.174 | 0.687 |

TSPulse reduces pre-training mask bias and generalizes more effectively to realistic missingness mechanisms.

## A.17 LIMITATIONS AND FUTURE WORK

While TSPulse demonstrates strong performance across a range of tasks, including similarity search, classification, anomaly detection, and imputation, there are several important areas for future development. One key direction is expanding its application to more downstream use cases, such as regression and full-fledged forecasting, to further extend its versatility. Additionally, TSPulse can be adapted for more challenging few-shot learning scenarios, particularly in classification tasks with limited labeled data. This would allow the model to perform effectively even in situations where labeled data is scarce, a common challenge in many real-world applications.

Another exciting direction is the expansion of TSPulse's capabilities by infusing semantic search with anomaly detection. By leveraging its powerful embeddings, future work could focus on identifying similar anomaly patterns from historical time-series data and retrieving related solutions from associated ticketing systems or knowledge bases. This would enhance its application in anomaly detection and make TSPulse a valuable tool for problem resolution across various industries, such as IT operations and healthcare. Additionally, exploring multi-modal fusion of TSPulse embeddings with those from other data types like text and images could unlock cross-domain applications, enabling more sophisticated models for complex tasks.

Finally, while TSPulse is lightweight and efficient, there is potential to enhance its ability to perform continuous learning. In dynamic environments where new data is constantly being generated, it is crucial to update the pre-trained model with new information while ensuring that previously learned knowledge is preserved. Future work will focus on implementing incremental learning techniques, enabling TSPulse to adapt to new data without forgetting the patterns learned from prior training. This approach will make TSPulse even more robust for real-time applications, where the model can continuously improve over time, maintaining both accuracy and stability across evolving datasets.

