# OpenReview forum: "TSPulse: Tiny Pre-Trained Models with Disentangled Representations for Rapid Time-Series Analysis"
_ICLR.cc/2026/Conference — ICLR 2026 Poster_

### Official Review · Reviewer_viWQ · 2025-10-26

**Soundness:** 3
**Presentation:** 3
**Contribution:** 3
**Rating:** 6
**Confidence:** 5

**Summary:**

This paper proposes ultra-light pre-trained models for various time series analysis tasks, including classification, imputation, anomaly detection, and semantic search. Key techniques involve dsentangled masked reconstruction for fine-grained and semantic embeddings and a hybrid masking scheme to avoid pre-training bias and boost generalization. More importantly, the proposed TSPulse outperforms models 10–100× larger on 75+ datasets.

**Strengths:**

1. novel and practical problem setting for designing fast and efficient time series foundation models.

2. Technical designs seem to be reasonable and solid.

3. Experimental results are promising and demonstrate the efficiency of the model.

4. Pretrained models, source code, and example scripts are included in the supplementary materials.

**Weaknesses:**

1. The presentation for the proposed method should be improved to enhance clarity. It is suggested to use 1 or 2 sentences to describe the key idea of the proposed TSPulse model.

2. According to Figure 6, the imputation performance is extremely good, which needs more discussion.

3. Why do we need the tasks for Similarity Search? Are there some applications?

**Questions:**

Please refer to the weakness

---

> ### Author Response · Authors · 2025-11-21
> **Novelty Clarification**
>
> **Q1 : Clarity on Key idea and improved presentation.**
>
>
> **Novelty in nutshell** - TSPulse proposes a **"new pre-training formulation"**: for "disentangled masked reconstruction" across multiple representational spaces (temporal vs. spectral) and multiple abstraction levels (fine-grained vs. semantic).  Importantly, we do not only learn across spaces (time vs. frequency), which is relatively common, but also across **abstraction levels** (local vs. semantic embeddings) in a **fully disentangled format**, which is critical for transfer learning and a unique proposition from TSPulse.
>
>
> **Details:** During pre-training, the model explicitly produces **separate temporal, spectral, and semantic embeddings** through multi-output heads that operate on different segments of the latent representation, enforcing **disentanglement by construction**. When these heads are **jointly optimized** with disentanglement imposed directly in the embedding space, the model learns representations with **strong generalization and transferability**—a capability that, to our knowledge, existing approaches do not provide.
>
> While prior works have explored **time–frequency fusion** or investigated **disentanglement in isolation**, TSPulse advances beyond these by **jointly learning disentangled representations across spaces *and* abstraction levels within a unified pre-training framework**. .
>
> **Disentanglement Embedding Analysis**
> Moreover, to validate that TSPulse effectively learns disentangled temporal, spectral, and semantic representations, we first presented a sensitivity analysis in Appendix A.4 demonstrating the robustness and utility of the semantic embedding. To complement this and contrast all embeddings systematically, we conducted additional set of experiments now on synthetic signals to measure how each embedding responds to missing data, noise, and phase/time shifts. We quantify these behaviours using a simple distortion metric that capture how much each embedding changes under controlled perturbations. The representative results below clearly exhibit the expected disentanglement patterns across the three embedding types.
>
> -------------------
> **Result Table (lower indicates less distortion, indicating better robustness to perturbations )**
>
> | **Experiment**  |  **Time embedding**  | **FFT embedding** | **Register embedding** |
> |-------------------|--------------------------|-----------------------|----------------------------|
> |30 % Missing data |    8.3%       |      27.4 %    |    4.6%.  |
> |Noise level 0.5 |      2.7%      |       6.8%      |   2.5 %   |
> |Phase/Time shift|    130%    |    21%  |     12%   |
>
>
> The table shows:
>
> - Time embeddings (size: 1536) are highly sensitive to time-shifts (as indicated by high distortion in row 3), indicating that temporal information is preserved - vital for tasks that rely on timing cues, which are often lost in purely spectral models.
>
> - FFT embeddings (size: 1536) are less sensitive to time-shifts (as indicated by low distortion in row 3), capturing spectral characteristics without timing details.
>
> - Compact Semantic embeddings (size: 256) are the least sensitive to time-shifts ( as indicated by least distortion in row 3) and are robust to missing data (least distortion in row 1) and noise (least distortion in row 2), as expected for embeddings designed to capture high-level structure. More detailed sensitivity analysis and PCA plots of semantic embedding properties are depicted in Appendix A. 4
>
> Each embedding type offers distinct advantages that benefit different downstream tasks.
>
> - Different datasets may rely on different cues for classification - some depend on raw temporal patterns, others on frequency characteristics, and some on high-level semantic summaries. TSLens attends to all three embeddings, letting the model automatically focus on whichever view the dataset needs.
>
> - Anomaly detection and imputation rely more on time and FFT embeddings for full reconstruction, which needs fine-grained informationfor loss-less reconstruction. Also, disentangled reconstructions across spaces enable effective triangulation to catch anomaly types missed by prior models.
>
> - Retrieval benefits mainly from semantic embeddings, which provide the summary-level representation.
>
> By providing these embeddings in a disentangled form for ready use along with task-related innovations (triangulation, tslens) that exploit the benefit of this disentanglement , **TSPulse enables flexible and effective transfer across a wide range of zero-shot tasks**.
>
> **Summary:** Overall, the paper introduces disentanglement across both representational spaces (time vs. FFT) and abstraction levels (fine-grained vs. semantic), and shows how these complementary embeddings/signals can be leveraged effectively for diverse diagnostic tasks, leading to double-digit performance gains across leaderboards and benchmarks - a unique capability enabled by TSPulse. We will improve the clarity in the paper accordingly.

---

> ### Author Response · Authors · 2025-11-21
> **Imputation Performance via Hybrid Masking**
>
> **Q2: According to Figure 6, the imputation performance is extremely good, which needs more discussion**
>
> The imputation performance stems from the hybrid masking that the paper proposes. While hybrid masking may appear straightforward at first glance, our contribution goes beyond simply “combining point and block masking.” Our work identifies and corrects a critical pre-training mask bias that arises in current imputation-oriented foundation models, specifically because they rely exclusively on fixed-length masking during pre-training.
>
> As shown in Appendix Table 14, existing popular pretrained models perform relatively well on block-masking imputation (i.e., fixed-length missingness), substantially outperforming linear interpolation. However, when we apply irregular or partial masking—where only parts of a block are missing—the expectation is that performance should improve (since the missing span is shorter). Instead, we observe the opposite: these models often perform worse than linear interpolation.
>
> This discrepancy led us to investigate further. We found that these models impute effectively only when the missing-span length matches the mask length used during pre-training. This is problematic because real-world missingness is highly variable and unpredictable. We refer to this phenomenon as pre-training mask bias.
>
> This bias has been largely overlooked in prior work and leads to a systematic mismatch between pre-training assumptions and real inference conditions. TSPulse identifies this issue and addresses it through hybrid masking, which exposes the model to a wide spectrum of variable-length missingness patterns during pre-training, rather than the fixed-length masks that rarely occur in real data. This is the core reason for the significant improvement of our results on the imputation task.
>
> Another important aspect in hybrid masking lies in the non-trivial way we implement hybrid learnable mask tokens.  Embedding-level mask tokens that are commonly used in prior-arts, implicitly assume that an entire patch is masked at once. However, hybrid masking allows **irregular partial masking within a patch**, which creates a non-trivial challenge: how to design a mask token that can represent missingness at **arbitrary positions** inside the patch.
>
> We address this by defining and learning a **mask token (M) in raw patch space**, rather than injecting a single embedding-level token. Each raw time point can then be masked by selecting the corresponding value from **M** based on its relative index within the patch (Figure 3A). This design allows a *single* learnable token to support **both full-patch masking and fine-grained partial masking**, enabling TSPulse to handle heterogeneous missingness patterns that embedding-level tokens cannot represent.

---

> ### Author Response · Authors · 2025-11-21
> **Similarity Search Applications**
>
> Q. Why do we need the tasks for Similarity Search? Are there some applications?
>
> Similarity search has been used in a wide range of applications, including identifying products with similar sales trajectories and retrieving seismic wave segments that match a given query pattern [a]. More recent work shows that it also supports temporal case-based reasoning in IoT and smart manufacturing workflows [b], and is included as a practical feature in several modern time-series management systems [c].
>
> Beyond these established uses, we are actively exploring the use of TSPulse Search in several real-world scenarios:
>
> **Active Learning (Label Efficiency)**: Similarity search retrieves nearest-neighbour segments for any unlabeled time-series input, enabling rapid label propagation and reducing annotation costs.
>
> **Industrial Predictive Analytics:** Retrieving historical vibration or sensor patterns that resemble a new signal supports case-based fault diagnosis and remaining-useful-life estimation. This strengthens predictive maintenance workflows by grounding predictions in previously observed degradation behaviors.
>
> **IT Observability (Incident Replay):** By matching a new anomaly pattern to prior incidents with similar temporal signatures, similarity search can surface past resolutions and speed up profiling and root-cause analysis across large-scale telemetry and microservice environments.
>
> In addition, strong compact embeddings learned for similarity search naturally support clustering, neighborhood retrieval, and related KNN-style analyses. As shown in Appendix A.4, our semantic embedding displays robust sensitivity properties that make it effective for grouping similar temporal patterns which has use in many industrial applications like manufacturing, health-care, etc.
>
> We are currently extending TSPulse’s search capabilities to these industrial settings, with the goal of demonstrating its value in retrieving meaningful patterns from large-scale, high-velocity, heterogeneous time-series data—contexts where traditional or manual search approaches are often impractical.
>
> [a] Agrawal et al., “Fast Similarity Search in the Presence of Noise, Scaling, and Translation in Time-Series Databases,” VLDB, 1995.
>
> [b] L. Malburg et al., “Modeling and Using Complex IoT Time Series Data in Case-Based Reasoning: From Application Scenarios to Implementations,” ICCBR Workshops, 2023.
>
> [c] S. K. Jensen et al., “Time Series Management Systems: A 2022 Survey,” Data Series Management and Analytics, 2022.

---

> > ### Comment · Reviewer_viWQ · 2025-11-24
> >
> > Thanks to the author for the rebuttal, I'll keep my positive rating.

---

### Official Review · Reviewer_9eiE · 2025-10-29

**Soundness:** 3
**Presentation:** 2
**Contribution:** 3
**Rating:** 6
**Confidence:** 4

**Summary:**

TSPulse is a 1M-parameter pre-trained model for time-series analysis that learns disentangled temporal, spectral, and semantic representations through multi-head masked reconstruction. Using a lightweight TSMixer backbone, hybrid masking, and a small decoder, it achieves efficient and robust pre-training. During fine-tuning, identity-initialized channel mixers enable multivariate adaptation without retraining the full model. TSPulse delivers strong performance across classification, anomaly detection, imputation, and similarity tasks while maintaining high efficiency on CPUs.

**Strengths:**

- The model uses a PEFT-style multivariate adaptation. It is pre-trained on univariate data and activates decoder-side channel mixers only during fine-tuning, initialized to identity for stability. This allows lightweight adaptation without retraining the entire model.
- The design is highly efficient, with about 1M parameters, low latency, and small memory usage. The pre-training setup on 1B samples in one day on 8×A100 makes it practical and scalable.
- The representation is disentangled into three segments—time, FFT, and register embeddings—and each task selectively uses the relevant parts. This improves transfer flexibility and prevents unnecessary coupling between features.

**Weaknesses:**

- The register-token design introduces inductive bias. Predicting spectral signatures forces phase invariance, which may hurt tasks that rely on absolute timing. The paper does not compare this approach to CLS-style aggregators or include related ablations.
- The hybrid masking strategy lacks novelty. Mixing block and point masks is common, and the argument that it better captures “real randomness” is weak. Patching is a modeling choice, while masking should reflect a data property (i.e., real-world missingness patterns). The paper conflates these levels and provides no statistical evidence that its hybrid masking aligns with realistic data gaps.
- The paper omits forecasting evaluation despite clear architectural similarity to imputation. In most time-series models, forecasting can be tested simply by changing the final output layer, so excluding it removes a standard and low-effort comparison point. Given that nearly all major time-series foundation models are benchmarked on forecasting, this omission significantly limits comparability and completeness.

**Questions:**

Please address the identified weaknesses and limitations noted above.

---

> ### Author Response · Authors · 2025-11-21
> **Benefits of TSPulse Disentanglement**
>
> **Q1: The register-token design introduces inductive bias. Predicting spectral signatures forces phase invariance, which may hurt tasks that rely on absolute timing.**
>
> Thanks for raising this important question. This is the precise benefit we get from TSPulse disentanglement.
>
> Just to recollect, TSPulse proposes a **"new pre-training formulation"**: for "disentangled masked reconstruction" across multiple representational spaces (temporal vs. spectral) and multiple abstraction levels (fine-grained vs. semantic).  Importantly, we do not only learn across spaces (time vs. frequency), which is relatively common, but also across **abstraction levels** (local vs. semantic embeddings) in a **fully disentangled format**, which is critical for transfer learning and a unique proposition from TSPulse.
>
> During pre-training, the model explicitly produces **separate temporal, spectral, and semantic embeddings** through multi-output heads that operate on different segments of the latent representation, enforcing **disentanglement by construction**. When these heads are **jointly optimized** with disentanglement imposed directly in the embedding space, the model learns representations with **strong generalization and transferability**—a capability that, to our knowledge, existing approaches do not provide.
>
> Now - coming to your query on **whether “predicting spectral signatures forces phase invariance, which may hurt tasks that rely on absolute timing** , we conducted additional set of experiments on synthetic signals to measure how each embedding responds to missing data, noise, and phase/time shifts. We quantify these behaviours using a simple distortion metric that capture how much each embedding changes under controlled perturbations. The representative results below clearly exhibit the expected disentanglement patterns across the three embedding types.
>
> **Summary**
>
> -------------------
>
> **Summary**
> -------------------
> **Representative Result Table (lower indicates less distortion, indicating better robustness to perturbations )**
>
> | **Experiment**  |  **Time embedding**  | **FFT embedding** | **Register embedding** |
> |-------------------|--------------------------|-----------------------|----------------------------|
> |30 % Missing data |    8.3%       |      27.4 %    |    4.6%.  |
> |Noise level 0.5 |      2.7%      |       6.8%      |   2.5 %   |
> |Phase/Time shift|    130%    |    21%  |     12%   |
>
>
> The table shows:
>
> - Time embeddings (size: 1536) are highly sensitive to time-shifts (as indicated by high distortion in row 3), indicating that temporal information is preserved - vital for tasks that rely on timing cues, which are often lost in purely spectral models. (This property precisely answers your query that temporal information is not lost)
>
> - FFT embeddings (size: 1536) are less sensitive to time-shifts (as indicated by low distortion in row 3), capturing spectral characteristics without timing details.
>
> - Compact Semantic embeddings (size: 256) are the least sensitive to time-shifts ( as indicated by least distortion in row 3) and are robust to missing data (least distortion in row 1) and noise (least distortion in row 2), as expected for embeddings designed to capture high-level structure. More detailed sensitivity analysis and PCA plots of semantic embedding properties are depicted in Appendix A. 4
>
> Each embedding type offers distinct advantages that benefit different downstream tasks.
>
> - Different datasets may rely on different cues for classification - some depend on raw temporal patterns, others on frequency characteristics, and some on high-level semantic summaries. TSLens attends to all three embeddings, letting the model automatically focus on whichever view the dataset needs.
>
> - Anomaly detection and imputation rely more on time and FFT embeddings for full reconstruction, which needs fine-grained informationfor loss-less reconstruction. Also, disentangled reconstructions across spaces enable effective triangulation to catch anomaly types missed by prior models.
>
> - Retrieval benefits mainly from semantic embeddings, which provide the summary-level representation.
>
> By providing these embeddings in a disentangled form for ready use along with task-related innovations (triangulation, tslens) that exploit the benefit of this disentanglement , **TSPulse enables flexible and effective transfer across a wide range of zero-shot tasks**.

---

> ### Author Response · Authors · 2025-11-21
> **CLS-style aggregator comparison**
>
> Q2: The paper does not compare this approach to CLS-style aggregators.
>
> We would like to clarify that we have compared TSPulse with CLS-style aggregator models. UniTS is one of the SOTA models that incorporates several types of tokens including input tokens, prompt tokens, and task tokens. CLS tokes are a kind of task tokens that help in the classification task for UniTS. We have compared TSPulse with UniTS in classification and imputation tasks where TSPulse outperformed UniTS by significant margins.

---

> ### Author Response · Authors · 2025-11-21
> **Hybrid Masking**
>
> **Q2: The hybrid masking strategy lacks novelty. Mixing block and point masks is common, and the argument that it better captures “real randomness” is weak. Patching is a modeling choice, while masking should reflect a data property (i.e., real-world missingness patterns). The paper conflates these levels and provides no statistical evidence that its hybrid masking aligns with realistic data gaps.**
>
> While hybrid masking may appear straightforward at first glance, our contribution goes beyond simply “combining point and block masking.” Our work identifies and corrects a critical pre-training mask bias that arises in current imputation-oriented foundation models, specifically because they rely exclusively on fixed-length masking during pre-training.
>
> As shown in Appendix Table 14, existing popular pretrained models perform relatively well on block-masking imputation (i.e., fixed-length missingness), substantially outperforming linear interpolation. However, when we apply irregular or partial masking—where only parts of a block are missing—the expectation is that performance should improve (since the missing span is shorter). Instead, we observe the opposite: these models often perform worse than linear interpolation.
>
> This discrepancy led us to investigate further. We found that these models impute effectively only when the missing-span length matches the mask length used during pre-training. This is problematic because real-world missingness is highly variable and unpredictable. We refer to this phenomenon as pre-training mask bias.
>
> This bias has been largely overlooked in prior work and leads to a systematic mismatch between pre-training assumptions and real inference conditions. TSPulse identifies this issue and addresses it through hybrid masking, which exposes the model to a wide spectrum of variable-length missingness patterns during pre-training, rather than the fixed-length masks that rarely occur in real data. **Since variable-length masks are closer to practical missingness patterns than constant-length masks (followed in existing prior-arts), we referred to them as “real missingness.” We agree that this terminology was imprecise without empirical validation, and we will rename it to “variable-length missingness patterns” for clarity.**
>
> However, to directly address the reviewer’s query and align our missingness experiments with how real-world data gaps are characterised in the literature, we adopt the well-established MAR and MNAR frameworks from statistical and machine-learning studies [1, 2], which are commonly used to empirically ground realistic missing-data behaviour.
>
> Details explained in the next comment on the new additional experiments conducted on MAR and MNAR scenarios.

---

> ### Author Response · Authors · 2025-11-21
> **Q2: Hybrid Masking (Contd..)**
>
> Based on the literature [1] [2] that defines MAR and MNAR as:
>
> **MAR (missing at random) :** If the probability of a point being missing is the same only within groups defined by the observed data, then the data are missing at random (MAR).
>
> **MNAR (missing not at random) :** If the probability of a point being missing varies for reasons that are unknown (unobserved) to us, then the data are missing not at random (MNAR).
>
> Below, we explain several real-world MAR and MNAR situations and how we simulate them. All evaluations are done on **ETTh1** from the LTF benchmark, a real multivariate hourly electricity-transformer dataset.
>
> ---
>
> ## MAR Scenarios
>
> ### **S1 – Peak-hour sensor overload**
>
> **Real world:** Sensors often fail more during peak load hours.
>
> **Simulation:** In each 24-point block, the middle 12 points have a missing probability of 0.6, and the remaining points have 0.05.
>
> ### **S2 – Cross-channel failure due to high readings**
>
> **Real world:** High readings in one channel (e.g., very high temperature) can cause other sensors to fail.
>
> **Simulation:** Missingness in a channel increases when another channel’s value lies in the top 10%.
>
> ### **S3 – Cross-channel failure due to volatility**
>
> **Real world:** Sudden fluctuations in one variable (e.g., power/voltage instability) can disrupt other sensors.
>
> **Simulation:** Missingness in a channel increases when another channel shows high short-term variance.
>
> ### **MAR Results (MSE)**
>
> | Scenario | TSPulse | MOMENT |
> |---------|---------|--------|
> | S1      | 0.147   | 0.575  |
> | S2      | 0.154   | 0.501  |
> | S3      | 0.138   | 0.402  |
>
>
> ---
>
> ## MNAR Scenarios
>
> ### **S1 – Sensor breakdown under extreme values**
>
> **Real world:** A sensor becomes unreliable as a measurement (e.g., temperature) crosses high thresholds.
>
> **Simulation:** Missingness increases with the magnitude of a channel’s values.
>
> ### **S2 – Latent stress, combining load levels + rate of change**
>
> **Real world:** Failures happen when a system experiences both high load and rapid fluctuations.
>
> **Simulation:** A latent variable combining one channel’s value and another channel’s derivative determines missingness.
>
> ### **S3 – System-wide stress causing multi-sensor degradation**
>
> **Real world:** Many sensors degrade together when the entire system’s operational load or stress increases.
>
> **Simulation:** Missingness increases with the moving-average trend of the sum of all non-temperature channels.
>
> ### **S4 – Time-dependent degradation**
>
> **Real world:** Sensors degrade and drop data more frequently over time.
>
> **Simulation:** Missingness probability increases gradually with time.
>
> ### **MNAR Results (MSE)**
>
> | Scenario | TSPulse | MOMENT |
> |----------|---------|---------|
> | S1       | 0.334   | 0.760   |
> | S2       | 0.134   | 0.552   |
> | S3       | 0.141   | 0.498   |
> | S4       | 0.174   | 0.687   |
>
>
> ---
>
> Across all MAR and MNAR settings, **TSPulse significantly outperforms MOMENT**, because of its hybrid-masking pre-training strategy. We will include these additional results in the revised draft.
>
> **References**
>
> [1] Donald B. Rubin. *Inference and Missing Data*. Biometrika, 63(3):581–592, 1976.
>
> [2] https://stefvanbuuren.name/fimd/sec-MCAR.html
>
> Another important aspect in hybrid masking lies in the non-trivial way we implement hybrid learnable mask tokens.  Embedding-level mask tokens that are commonly used in prior-arts, implicitly assume that an entire patch is masked at once. However, hybrid masking allows **irregular partial masking within a patch**, which creates a non-trivial challenge: how to design a mask token that can represent missingness at **arbitrary positions** inside the patch.
>
> We address this by defining and learning a **mask token (M) in raw patch space**, rather than injecting a single embedding-level token. Each raw time point can then be masked by selecting the corresponding value from **M** based on its relative index within the patch (Figure 3A). This design allows a *single* learnable token to support **both full-patch masking and fine-grained partial masking**, enabling TSPulse to handle heterogeneous missingness patterns that embedding-level tokens cannot represent.

---

> ### Author Response · Authors · 2025-11-21
> **Forecasting Results**
>
> **Q3: The paper omits forecasting evaluation**
>
> We appreciate this concern and would like to clarify that **TSPulse is not designed for long-horizon causal forecasting**. Although the model includes a small Predictive Head, it is purposefully limited to **few-step prediction** for the Predictive Anomaly Detection (PredAD) use-case, which uses  a **1-step forecast**. This head supports representation learning but is **not intended as a competitive long-horizon forecasting module**.
>
> State-of-the-art performance on long-range forecasting benchmarks typically relies on **causal pretraining objectives** (e.g., Chronos, TiRex, TOTO). These models are explicitly designed with a **forecasting-first principle**, where information flows strictly forward in time—an essential property for autoregressive or causal decoding.
>
> However, while these causal objectives are ideal for forecasting, they are **not preferred for diagnostic tasks**, which rely on **bidirectional, reconstruction-driven pretraining**. TSPulse is built around such a **bidirectional objective**, which is crucial for strong performance on **anomaly detection, imputation, classification, and semantic search**—tasks that inherently depend on **non-causal, context-rich reasoning**. Introducing long-horizon causal constraints would weaken these diagnostic capabilities and move the model away from its intended purpose as **diagnostics-first pretrained models**.
>
> Thus, causal FMs such as Chronos, TiRex, TOTO naturally focus on forecasting. In the same spirit, **TSPulse keeps its emphasis on diagnostic tasks**, aligning its pretraining formulation with the requirements of those tasks. This reflects **different design priorities**, not limitations of either family of models.
>
> That said, solely to address your concern, we ran additional experiments by updating the Predictive Head to support long forecasting horizons and evaluated TSPulse on **standard long-horizon forecasting (LSF) benchmarks**. Even though TSPulse is not architecturally optimized for forecasting, it performs **reasonably well**, especially given its **compact size (1M parameters)**:
>
> - It **outperforms reconstruction-based models and older causal models**  in many cases.
>
> - As expected, it does **not outperform latest causal models** such as TOTO.
>
> **Zero-shot MSE results from LSF benchmarks (averaged across forecast lengths 96, 192, 336, 720)**
>
> | Dataset     | TSPulse (1M) | Chronos (203M) | Moirai_Large (311M) | UniTS (2.4M) | TOTO (151M) |
> |-------------|--------------|----------------|----------------------|--------------|-------------|
> | ETTH1       | 0.411        | 0.445          | 0.510                | 0.527        | 0.435       |
> | ETTH2       | 0.357        | 0.368          | 0.354                | 0.405        | 0.340       |
> | ETTM1       | 0.405        | 0.425          | 0.390                | 0.713        | 0.396       |
> | ETTM2       | 0.290        | 0.292          | 0.276                | 0.321        | 0.267       |
> | Weather     | 0.232        | 0.266          | 0.260                | 0.291        | 0.224       |
> | Electricity | 0.203        | 0.166          | 0.188                | 0.432        | 0.158       |
> | **Average** | **0.316**    | **0.327**      | **0.330**            | **0.448**    | **0.303**   |
>
>
>
> These results should be interpreted as **supportive evidence**, not as forecasting claims. TSPulse performs **respectably for a non-causal reconstruction model**, but—as expected—it is **not competitive with newer causal forecasters**.
>
> TSPulse’s core contribution lies in **multi-space, multi-abstraction masked reconstruction**, which enables strong transfer to **diagnostic tasks**. Forecasting was never its target objective; we evaluated it only to address your concern and to show that its representations transfer **reasonably well even outside their intended domain**.

---

> ### Author Response · Authors · 2025-11-27
>
> Dear Reviewer,
>
> We have submitted a detailed rebuttal along with various new experimental studies to clarify the concerns raised during the initial review.
>
> If there are any additional questions or further clarification needed, we would be very happy to provide them promptly.
>
> Thank you once again for your time and engagement.

---

### Official Review · Reviewer_6n4K · 2025-10-30

**Soundness:** 2
**Presentation:** 3
**Contribution:** 3
**Rating:** 4
**Confidence:** 3

**Summary:**

TSPulse proposes a tiny pre-trained backbone for time-series diagnostics, including classification, anomaly detection, imputation, and similarity search, that is designed for low-latency CPU/GPU inference. The approach learns disentangled representations by reconstructing signals in both time and frequency domains and by separating fine-grained from semantic information, supported by a hybrid masking scheme that varies mask type and span to reflect realistic missingness. Built on a TSMixer backbone with a lightweight mini-decoder, the work integrates task-oriented adapters (e.g., TSLens for multivariate classification and multi-head triangulation for anomaly detection) to translate the embeddings into downstream gains. Empirical results across standard benchmarks indicate consistent improvements over larger baselines, while maintaining a compact footprint that facilitates deployment.

**Strengths:**

- The approach provides a ~1M-parameter pre-trained model with low-latency CPU/GPU inference, facilitating practical deployment.
- The work employs a dual time–frequency objective with hybrid masking that mirrors realistic missingness and yields robust representations.
- Task-oriented modules (e.g., TSLens, triangulation) translate the embeddings into measurable gains across multiple diagnostic tasks.

**Weaknesses:**

- Reliance on task-specific loss re-weighting may limit out-of-the-box generality and increase tuning burden.
- The representation is not assessed on long-horizon causal forecasting, which constrains claims of transferability beyond diagnostics.
- The lack of parameter-matched plug-in baselines (e.g., gated residuals, lightweight cross-attention) weakens isolation of design effects from model capacity.
- The empirical evidence would be stronger with uncertainty estimates and per-dataset breakdowns to substantiate the reported gains.

**Questions:**

- What default loss weights work reliably across domains, and how sensitive are downstream results to these weights during task-specialised pre-training and fine-tuning?
- Do the disentangled embeddings support causal long-horizon forecasting if a forecasting head is added, and how do they compare with forecasting-oriented pre-trained models under strict parameter matching?
- How does TSPulse perform against equally small plug-ins—such as a gated residual or lightweight cross-attention when total parameters and training budget are held constant?
- Can empirical missingness models from real datasets (e.g., MAR/MNAR) be provided, and can robustness be tested under these distributions to validate the hybrid masking design?

---

> ### Author Response · Authors · 2025-11-21
> **Task Specialization**
>
> **Q1:** *Reliance on task-specific loss re-weighting may limit out-of-the-box generality and increase tuning burden. What default loss weights work reliably across domains, and how sensitive are downstream results to these weights during task-specialised pre-training and fine-tuning?*
>
> Thank you very much for this thoughtful question. We agree that **task-specific loss re-weighting** can reduce the level of out-of-the-box generality. In our experience, however, this design choice has been **practically the most effective for real-world deployments**. Tasks such as classification, anomaly detection, retrieval, and imputation differ substantially in their objectives, error sensitivities, and inductive biases. In several real model deployments we consistently find that **accuracy, robustness, and operational simplicity** matter more than forcing a single model to handle all tasks simultaneously. This trend is also increasingly common in LLM and vision pipelines, where organizations deploy **small task-focused variants** for reliability and efficiency.
>
> To avoid placing any tuning burden on end users, we provide **three fixed pre-trained variants**, each specialized for a family of tasks. Importantly, **users do not modify loss weights** or run any tuning during inference:
>
> - **Anomaly Detection model:** trained with all reconstruction heads active (triangulation) for strong robustness.
>
> - **Classification model:** dual-head model with block masking to emphasize discriminative semantics.
>
> - **Imputation + Search model:** dual-head model with hybrid masking to avoid reconstruction bias and improve structure recovery.
>
> All three share **the same architecture**; only the **pre-training configuration** differs. This ensures that specialization happens once during model preparation, and end users experience **no additional complexity**.
>
> However, while our primary design choice is to use **task-specialized models**—which offer clear practical and accuracy benefits—we also conducted additional experiments to evaluate the scenario where **a single pre-trained model** is used across all four downstream tasks, to address your query on **out-of-box generalization**.
>
> Specifically, we selected the model variant currently used for **imputation and retrieval/search**, and evaluated this same model on **classification and anomaly detection**, ensuring that all tasks use **one unified pre-trained checkpoint**. As shown in the table below, this single-model variant still achieves **competitive performance**, typically ranking within the **top 1–2 positions**, and even marginally surpassing the best baselines on certain tasks. This demonstrates that **TSPulse maintains strong generalization** even without task specialization.
>
> That said, enabling **task-specific pretraining** further yields **additional performance gains**. We will include this comparative table in the revision to clarify this point.
>
>
> | **Task**                        | **Metric**   | **Task-specialized TSPulse** | **One TSPulse model for all tasks** | **Best SOTA Baseline**         |
> |---------------------------------|--------------|-------------------------------|--------------------------------------|---------------------------------|
> | Classification                  | Accuracy     | **0.73**                      | **0.71**                             | 0.70 (VQShape)                  |
> | AD (univariate)                 | VUS-PR       | **0.48**                      | **0.42**                                 | 0.42 (Sub-PCA)                  |
> | Search (Family Match)           | PREC@3       | **0.68**                      | **0.68**                             | 0.53 (MOMENT)                   |
> | Search (Fine-grained Match)     | PREC@3       | **0.58**                      | **0.58**                             | 0.42 (MOMENT)                   |
> | Imputation (Block)              | MSE          | **0.106**                     | **0.106**                            | 0.176 (UniTS)                   |
> | Imputation (Hybrid)             | MSE          | **0.074**                     | **0.074**                            | 0.161 (Linear Interpolation)    |
>
>
> *MSE (lower is better); all other metrics: higher is better.*

---

> ### Author Response · Authors · 2025-11-21
> **Forecasting related queries**
>
> **Q2:** *The representation is not assessed on long-horizon causal forecasting, which constrains claims of transferability beyond diagnostics.*
>
> We appreciate this concern and would like to clarify that **TSPulse is not designed for long-horizon causal forecasting**. Although the model includes a small Predictive Head, it is purposefully limited to **few-step prediction** for the Predictive Anomaly Detection (PredAD) use-case, which uses  a **1-step forecast**. This head supports representation learning but is **not intended as a competitive long-horizon forecasting module**.
>
> State-of-the-art performance on long-range forecasting benchmarks typically relies on **causal pretraining objectives** (e.g., Chronos, TiRex, TOTO). These models are explicitly designed with a **forecasting-first principle**, where information flows strictly forward in time—an essential property for autoregressive or causal decoding.
>
> However, while these causal objectives are ideal for forecasting, they are **not preferred for diagnostic tasks**, which rely on **bidirectional, reconstruction-driven pretraining**. TSPulse is built around such a **bidirectional objective**, which is crucial for strong performance on **anomaly detection, imputation, classification, and semantic search**—tasks that inherently depend on **non-causal, context-rich reasoning**. Introducing long-horizon causal constraints would weaken these diagnostic capabilities and move the model away from its intended purpose as **diagnostics-first pretrained models**.
>
> Thus, causal FMs such as Chronos, TiRex, TOTO naturally focus on forecasting. In the same spirit, **TSPulse keeps its emphasis on diagnostic tasks**, aligning its pretraining formulation with the requirements of those tasks. This reflects **different design priorities**, not limitations of either family of models.
>
> That said, solely to address your concern, we ran additional experiments by updating the Predictive Head to support long forecasting horizons and evaluated TSPulse on **standard long-horizon forecasting (LSF) benchmarks**. Even though TSPulse is not architecturally optimized for forecasting, it performs **reasonably well**, especially given its **compact size (1M parameters)**:
>
> - It **outperforms reconstruction-based models and older causal models**  in many cases.
>
> - As expected, it does **not outperform latest causal models** such as TOTO.
>
> **Zero-shot MSE results from LSF benchmarks (averaged across forecast lengths 96, 192, 336, 720)**
>
> | Dataset     | TSPulse (1M) | Chronos (203M) | Moirai_Large (311M) | UniTS (2.4M) | TOTO (151M) |
> |-------------|--------------|----------------|----------------------|--------------|-------------|
> | ETTH1       | 0.411        | 0.445          | 0.510                | 0.527        | 0.435       |
> | ETTH2       | 0.357        | 0.368          | 0.354                | 0.405        | 0.340       |
> | ETTM1       | 0.405        | 0.425          | 0.390                | 0.713        | 0.396       |
> | ETTM2       | 0.290        | 0.292          | 0.276                | 0.321        | 0.267       |
> | Weather     | 0.232        | 0.266          | 0.260                | 0.291        | 0.224       |
> | Electricity | 0.203        | 0.166          | 0.188                | 0.432        | 0.158       |
> | **Average** | **0.316**    | **0.327**      | **0.330**            | **0.448**    | **0.303**   |
>
>
> These results should be interpreted as **supportive evidence**, not as forecasting claims. TSPulse performs **respectably for a non-causal reconstruction model**, but—as expected—it is **not competitive with newer causal forecasters**.
>
> TSPulse’s core contribution lies in **multi-space, multi-abstraction masked reconstruction**, which enables strong transfer to **diagnostic tasks**. Forecasting was never its target objective; we evaluated it only to address your concern and to show that its representations transfer **reasonably well even outside their intended domain**.

---

> ### Author Response · Authors · 2025-11-21
> **Attention Plugins & Hybrid Masking**
>
> **Q3: How does TSPulse perform against equally small plug-ins—such as a gated residual or lightweight cross-attention when total parameters and training budget are held constant?**
>
> Thank you for this thoughtful question. We have in fact implemented and evaluated several lightweight, plug-in–style attention modules. Our codebase (as shared in the supplementary) includes implementations of various attention plug-ins such as **TSPulseGatedAttention** (a lightweight softmax-based linear attention similar to the original TSMixer gating) and **TSPulseAttention** (a small multi-head self-attention block). These modules can be enabled as needed via config
>
> Our findings were as follows: The self-attention plug-ins added substantial training-time overhead—even at comparable parameter counts—because their cost scales quadratically with sequence length. Importantly, this extra compute did not translate into accuracy gains on our 1B  pre-training corpus; the original TSMixer GatedAttention with its residual mixing remained consistently stronger and faster.
>
> These observations motivated us to retain the simpler and computationally efficient gated attention with residual mixing mechanism within the backbone.
>
> We also note that the role of TSPulse is complementary to these plug-ins. Attention plug-ins primarily influence **how features are mixed or attended within a single representational space**. TSPulse, in contrast, adds a **higher-level pre-training formulation** that enriches what the model learns and how it disentangles the embeddings. Rather than modifying the backbone itself, TSPulse introduces objectives that encourage the model to learn **disentangled representations across three different views**, which are readily utilized by downstream tasks in zero-shot fashion.
>
> These objectives shape the representation space in ways that backbone plug-ins alone do not address. In practice, we found that the simple gated residual mechanism pairs well with TSPulse, while the heavier cross-attention plug-ins did not provide additional benefits under equal parameter and budget constraints.
>
> ----------------------------
> **Q4: Can empirical missingness models from real datasets (e.g., MAR/MNAR) be provided, and can robustness be tested under these distributions to validate the hybrid masking design?**
>
> Thanks for your feedback. Before getting into the details of MAR/MNAR, we would like to highlight and motivate the design of hybrid masking for better clarification.
>
> While hybrid masking may appear straightforward at first glance, our contribution goes beyond simply “combining point and block masking.” Our work identifies and corrects a **critical pre-training mask bias** that arises in current **imputation-oriented foundation models**, specifically because they rely exclusively on **fixed-length masking** during pre-training.
>
> As shown in **Appendix Table 14**, existing popular pretrained models  perform relatively well on block-masking imputation (i.e., fixed-length missingness), substantially outperforming linear interpolation. However, when we apply **irregular or partial masking**—where only parts of a block are missing—the expectation is that performance should improve (since the missing span is shorter). Instead, we observe the **opposite**: these models often perform **worse than linear interpolation**.
>
> This discrepancy led us to investigate further. We found that these models impute effectively **only when the missing-span length matches the mask length used during pre-training**. This is problematic because real-world missingness is **highly variable and unpredictable**. We refer to this phenomenon as **pre-training mask bias**.
>
> This bias has been largely overlooked in prior work and leads to a **systematic mismatch** between pre-training assumptions and real inference conditions. **TSPulse identifies this issue and addresses it through hybrid masking**, which exposes the model to a wide spectrum of variable-length missingness patterns during pre-training, rather than the fixed-length masks that rarely occur in real data. Since variable-length masks are closer to practical missingness patterns than constant-length masks, we  referred to them as “real missingness.” We agree that this terminology was imprecise without empirical validation, and we will rename it to “variable-length missingness patterns” for clarity.
>
> In addition, to provide stronger empirical grounding, we construct explicit MAR and MNAR masking scenarios below (as per your query)  and conduct further experiments to evaluate the robustness of our model under these well-defined missingness conditions.
>
> Continued in next comment.

---

> ### Author Response · Authors · 2025-11-21
> **Q4: Hybrid Masking & MAR/MNAR Scenarios (Contd.)**
>
> Based on the literature [1] [2] that defines MAR and MNAR as:
>
> **MAR (missing at random) :** If the probability of a point being missing is the same only within groups defined by the observed data, then the data are missing at random (MAR).
>
> **MNAR (missing not at random) :** If the probability of a point being missing varies for reasons that are unknown (unobserved) to us, then the data are missing not at random (MNAR).
>
> Below, we explain several real-world MAR and MNAR situations and how we simulate them. All evaluations are done on **ETTh1** from the LTF benchmark, a real multivariate hourly electricity-transformer dataset.
>
> ---
>
> ## MAR Scenarios
>
> ### **S1 – Peak-hour sensor overload**
>
> **Real world:** Sensors often fail more during peak load hours.
>
> **Simulation:** In each 24-point block, the middle 12 points have a missing probability of 0.6, and the remaining points have 0.05.
>
> ### **S2 – Cross-channel failure due to high readings**
>
> **Real world:** High readings in one channel (e.g., very high temperature) can cause other sensors to fail.
>
> **Simulation:** Missingness in a channel increases when another channel’s value lies in the top 10%.
>
> ### **S3 – Cross-channel failure due to volatility**
>
> **Real world:** Sudden fluctuations in one variable (e.g., power/voltage instability) can disrupt other sensors.
>
> **Simulation:** Missingness in a channel increases when another channel shows high short-term variance.
>
> ### **MAR Results (MSE)**
>
> | Scenario | TSPulse | MOMENT |
> |---------|---------|--------|
> | S1      | 0.147   | 0.575  |
> | S2      | 0.154   | 0.501  |
> | S3      | 0.138   | 0.402  |
>
> ---
>
> ## MNAR Scenarios
>
> ### **S1 – Sensor breakdown under extreme values**
>
> **Real world:** A sensor becomes unreliable as a measurement (e.g., temperature) crosses high thresholds.
>
> **Simulation:** Missingness increases with the magnitude of a channel’s values.
>
> ### **S2 – Latent stress, combining load levels + rate of change**
>
> **Real world:** Failures happen when a system experiences both high load and rapid fluctuations.
>
> **Simulation:** A latent variable combining one channel’s value and another channel’s derivative determines missingness.
>
> ### **S3 – System-wide stress causing multi-sensor degradation**
>
> **Real world:** Many sensors degrade together when the entire system’s operational load or stress increases.
>
> **Simulation:** Missingness increases with the moving-average trend of the sum of all non-temperature channels.
>
> ### **S4 – Time-dependent degradation**
>
> **Real world:** Sensors degrade and drop data more frequently over time.
>
> **Simulation:** Missingness probability increases gradually with time.
>
> ### **MNAR Results (MSE)**
>
>
> | Scenario | TSPulse | MOMENT |
> |----------|---------|---------|
> | S1       | 0.334   | 0.760   |
> | S2       | 0.134   | 0.552   |
> | S3       | 0.141   | 0.498   |
> | S4       | 0.174   | 0.687   |
>
>
> Across all MAR and MNAR settings, **TSPulse significantly outperforms MOMENT**, because of its hybrid-masking pre-training strategy. We will include these additional results in the revised draft.
>
> **References**
>
> [1] Donald B. Rubin. *Inference and Missing Data*. Biometrika, 63(3):581–592, 1976.
>
> [2] https://stefvanbuuren.name/fimd/sec-MCAR.html
>
> Another important aspect in hybrid masking lies in the non-trivial way we implement hybrid learnable mask tokens.  Embedding-level mask tokens that are commonly used in prior-arts, implicitly assume that an entire patch is masked at once. However, hybrid masking allows **irregular partial masking within a patch**, which creates a non-trivial challenge: how to design a mask token that can represent missingness at **arbitrary positions** inside the patch.
>
> We address this by defining and learning a **mask token (M) in raw patch space**, rather than injecting a single embedding-level token. Each raw time point can then be masked by selecting the corresponding value from **M** based on its relative index within the patch (Figure 3A). This design allows a *single* learnable token to support **both full-patch masking and fine-grained partial masking**, enabling TSPulse to handle heterogeneous missingness patterns that embedding-level tokens cannot represent.

---

> ### Author Response · Authors · 2025-11-21
> **Report per-dataset breakdowns**
>
> **Q6:  Report per-dataset breakdowns to substantiate the reported gains.**
>
> Please note that the per-dataset breakdowns are already provided in the appendix for all tasks:
>
> **Classification:** Table 12
>
> **Anomaly Detection:** Tables 9–10
>
> **Imputation:** Tables 14–20
>
> **Time-series Retrieval:** Tables 23–24
>
> Across these tasks, we evaluate on **75+ datasets**, which offers a broad and fine-grained view of model behavior across datasets. The results are highly consistent across diverse domains—TSPulse shows clear gains over strong baselines and SOTA models in most datasets. This large cross-dataset consistency provides strong practical evidence of robustness.

---

> ### Author Response · Authors · 2025-11-21
> **Clarification on Novelty and Disentanglement Embedding Analysis**
>
> We would like to provide a general clarification on novelty along with an interesting embedding analysis to highlight the benefits of our work
>
> **Novelty** - TSPulse proposes a **"new pre-training formulation"**: for "disentangled masked reconstruction" across multiple representational spaces (temporal vs. spectral) and multiple abstraction levels (fine-grained vs. semantic).  Importantly, we do not only learn across spaces (time vs. frequency), which is relatively common, but also across **abstraction levels** (local vs. semantic embeddings) in a **fully disentangled format**, which is critical for transfer learning and a unique proposition from TSPulse.
>
> During pre-training, the model explicitly produces **separate temporal, spectral, and semantic embeddings** through multi-output heads that operate on different segments of the latent representation, enforcing **disentanglement by construction**. When these heads are **jointly optimized** with disentanglement imposed directly in the embedding space, the model learns representations with **strong generalization and transferability**—a capability that, to our knowledge, existing approaches do not provide.
>
> While prior works have explored **time–frequency fusion** or investigated **disentanglement in isolation**, TSPulse advances beyond these by **jointly learning disentangled representations across spaces *and* abstraction levels within a unified pre-training framework**. .
>
> Moreover, to validate that TSPulse effectively learns disentangled temporal, spectral, and semantic representations, we first presented a sensitivity analysis in Appendix A.4 demonstrating the robustness and utility of the semantic embedding. To complement this and contrast all embeddings systematically, we conducted additional set of experiments now on synthetic signals to measure how each embedding responds to missing data, noise, and phase/time shifts. We quantify these behaviours using a simple distortion metric that capture how much each embedding changes under controlled perturbations. The representative results below clearly exhibit the expected disentanglement patterns across the three embedding types.
>
> **Summary**
> -------------------
> **Representative Result Table (lower indicates less distortion, indicating better robustness to perturbations )**
>
> | **Experiment**  |  **Time embedding**  | **FFT embedding** | **Register embedding** |
> |-------------------|--------------------------|-----------------------|----------------------------|
> |30 % Missing data |    8.3%       |      27.4 %    |    4.6%.  |
> |Noise level 0.5 |      2.7%      |       6.8%      |   2.5 %   |
> |Phase/Time shift|    130%    |    21%  |     12%   |
>
>
>
> The table shows:
>
> - Time embeddings (size: 1536) are highly sensitive to time-shifts (as indicated by high distortion in row 3), indicating that temporal information is preserved - vital for tasks that rely on timing cues, which are often lost in purely spectral models.
>
> - FFT embeddings (size: 1536) are less sensitive to time-shifts (as indicated by low distortion in row 3), capturing spectral characteristics without timing details.
>
> - Compact Semantic embeddings (size: 256) are the least sensitive to time-shifts ( as indicated by least distortion in row 3) and are robust to missing data (least distortion in row 1) and noise (least distortion in row 2), as expected for embeddings designed to capture high-level structure. More detailed sensitivity analysis and PCA plots of semantic embedding properties are depicted in Appendix A. 4
>
> Each embedding type offers distinct advantages that benefit different downstream tasks.
>
> - Different datasets may rely on different cues for classification - some depend on raw temporal patterns, others on frequency characteristics, and some on high-level semantic summaries. TSLens attends to all three embeddings, letting the model automatically focus on whichever view the dataset needs.
>
> - Anomaly detection and imputation rely more on time and FFT embeddings for full reconstruction, which needs fine-grained informationfor loss-less reconstruction. Also, disentangled reconstructions across spaces enable effective triangulation to catch anomaly types missed by prior models.
>
> - Retrieval benefits mainly from semantic embeddings, which provide the summary-level representation.
>
> By providing these embeddings in a disentangled form for ready use along with task-related innovations (triangulation, tslens) that exploit the benefit of this disentanglement , **TSPulse enables flexible and effective transfer across a wide range of zero-shot tasks**.
>
> **Summary**: Overall, the paper introduces disentanglement across both representational spaces (time vs. FFT) and abstraction levels (fine-grained vs. semantic), and shows how these complementary embeddings/signals can be leveraged effectively for diverse diagnostic tasks, leading to double-digit performance gains across leaderboards and benchmarks - a unique capability enabled by TSPulse.

---

> ### Author Response · Authors · 2025-11-27
>
> Dear Reviewer,
>
> We have submitted a detailed rebuttal along with various new experimental studies to clarify the concerns raised during the initial review.
>
> If there are any additional questions or further clarification needed, we would be very happy to provide them promptly.
>
> Thank you once again for your time and engagement.

---

### Official Review · Reviewer_RHFo · 2025-11-01

**Soundness:** 2
**Presentation:** 1
**Contribution:** 2
**Rating:** 2
**Confidence:** 5

**Summary:**

The paper introduces TSPulse, a 1M parameter pre-trained model for time-series analysis, which is built upon the TSMixer architecture. The authors claim state-of-the-art (SOTA) performance across four distinct downstream tasks: classification, anomaly detection, imputation, and similarity retrieval. The model's purported advantages stem from a collection of "innovations," including:

1.  **Architectural:** A dual-space (time and frequency) masked reconstruction strategy and a dual-embedding disentanglement (detailed vs. semantic) mechanism.
2.  **Task-Specific:** A `TSLens` fine-tuning module for classification, a `multi-head triangulation` (MHT) method for anomaly detection, and a `hybrid masking` strategy for imputation.

Despite the impressive reported results, the paper suffers from fundamental methodological flaws, a severe lack of focus, and claims that are not substantiated by the experimental design. The work reads less like a scientific contribution and more like an engineering report for a model that has been over-tuned on a specific set of benchmarks.

**Strengths:**

There are several strengths in the paper worth highlighting:

1.  The goal of creating a single, compact model (even if the pre-training is specialized) that excels at four major time-series tasks is highly ambitious and addresses a clear need in the community.
2.  The paper's strongest contribution is its relentless focus on efficiency. Achieving the reported results with a 1M parameter model is a significant engineering feat, especially the "GPU-Free" (CPU-capable) inference. This is a crucial and practical research direction, providing a counter-narrative to the "bigger is better" trend in foundation models.
3.   While the methodology is questionable, the sheer scale of the reported performance gains (e.g., +20% on TSB-AD, +50% on imputation) is eye-catching. It demonstrates that small, well-engineered models can be *tuned* to outperform large, general-purpose ones, which is an interesting finding in itself.

**Weaknesses:**

### 1. Fundamentally Flawed Experimental Premise

The paper's central narrative, that **one** tiny 1M model can be pre-trained for general-purpose use and subsequently outperform massive foundation models (like MOMENT, Chronos, etc.)—is directly contradicted by the authors' own methodology, which is buried in Appendix A.6.

The authors state:
> "...we specialize the pre-training for every task through reweighting loss objectives..." And more damningly:
> "TSPulse adopts hybrid masking for all reconstruction-oriented tasks (imputation, anomaly detection, retrieval) and block masking specifically for classification-oriented pre-training."
This is a **fatal flaw**. The authors are not evaluating a single, general-purpose pre-trained model. They are training *multiple, distinct models* where the pre-training objective and masking strategy are *already specialized for the downstream task*.

This completely invalidates the comparisons against general-purpose foundation models like MOMENT or UniTS. Of course, a model pre-trained *specifically* for imputation using hybrid masking will outperform a general-purpose model on an imputation task that uses hybrid masking. The paper is comparing apples to oranges: a bespoke, task-specific small model against a general-purpose large model. The extraordinary claims of +50% in imputation or +20% in anomaly detection are therefore not surprising, but rather an expected outcome of this flawed, self-serving experimental design.

### 2. Lacking a Coherent Contribution

The primary weakness of this paper is its overwhelming complexity and lack of a central, verifiable contribution. The authors introduce no fewer than five new, named components: `Dual-space reconstruction`, `dual-embedding disentanglement`, `TSLens`, `multi-head triangulation`, and `hybrid masking`. This obscures any real scientific insight. It is impossible to determine if a single, principled idea is responsible for the alleged gains, or if the model is simply a fragile collection of ad-hoc engineering tricks, each providing a marginal boost on a specific benchmark. The paper fails to present a clear, unified thesis, instead opting for a laundry list of features that make the work difficult to interpret

### 3. Insufficient and Unconvincing Ablation Studies

Given the model's complexity, the ablation study (Section 5) is the most critical piece of evidence needed to justify the design. The provided study is wholly inadequate.

* The ablations only remove one component at a time from the *final, complex model*. This fails to prove that all components are necessary. A proper study would start from the TSMixer baseline and *add* each component one-by-one to show its marginal contribution.
* The study fails to disentangle the contributions. For example, in Table 1(b), removing `TSLens` causes a 16% drop, and removing `Dual-space` causes a 7% drop. What happens if *both* are removed? Does the performance revert to the TSMixer baseline, or is there a complex, non-additive interaction?
* What is the performance of *just* the TSMixer baseline + `TSLens`? What about TSMixer + `Dual-space`? Without this, it's impossible to know if the "dual-embedding" or other components are providing any real value or are simply needless complexity. The ablations feel selected to justify the components rather than to genuinely investigate them.

### 4.  Many of the paper's "novel" contributions appear to be re-brandings of existing, standard techniques.

* **Dual-Space Learning:** Using both time and frequency domains is a classic signal processing technique. The paper itself cites prior work like **FEDformer** and **BTSF** that already leverage this. Claiming this as a novel contribution for "unifying" them in a *lightweight* model is a weak, incremental claim.
* **Multi-Head Triangulation (MHT):** This is just a simple ensemble. The model computes reconstruction error from three different heads (time, FFT, forecast) and picks the best one (`Headtriang.`) or combines them (`Headensemble`). This is a standard validation/ensemble technique, not a novel anomaly detection framework.
* **TSLens:** The description ("selectively attends to and weights features") is functionally identical to a standard attention mechanism applied as a pooling layer for classification. Giving it a new name does not make it a new contribution.
* **Hybrid Masking:** This is a data augmentation strategy, not a model innovation. It is an obvious heuristic to combine point and block masking.

**Questions:**

The key weaknesses outlined above raise several questions that the authors need to address:

1.  The core claim of a single, general-purpose model appears to be contradicted by the task-specific pre-training (Appendix A.6). Could the authors please provide results for a **single model**, pre-trained with a **single, unified objective** (e.g., the hybrid-masking, all-head-loss model), and then evaluated on all four downstream tasks? This is essential to validate the paper's central thesis.
2.  The current ablation study is "destructive" (removing parts from the whole). Could the authors provide a "constructive" study, starting from the TSMixer baseline and *incrementally adding* each new component (e.g., 1. TSMixer, 2. TSMixer + Dual-Space, 3. TSMixer + Dual-Space + Dual-Embedding, etc.) to clearly demonstrate the marginal performance contribution of each proposed innovation?
3.  How much of the performance gain is simply due to pre-training the TSMixer backbone on 1B samples, versus the novel architectural components? A crucial missing baseline is a standard TSMixer (with a simple reconstruction head) pre-trained on the *same* 1B sample dataset.
4.  The `TSLens` module is described as learning to "focus on the most informative regions." How does this mechanism fundamentally differ from a standard attention-pooling layer (e.g., a single-head attention followed by a weighted sum) commonly used in classification heads?
5.  **Proof of Disentanglement:** The paper claims the dual-embedding approach generates "detailed" and "semantic" embeddings. Beyond their use in different reconstruction heads, what qualitative or quantitative evidence is there that these embeddings are truly disentangled? For instance, do the "semantic" embeddings (from register tokens) show quantifiably more invariance to noise and time shifts compared to the "detailed" embeddings?

---

> ### Author Response · Authors · 2025-11-21
> **Response to Q1 on Fundamentally Flawed Experimental Premise**
>
> **Q1 [Copy Pasting the exact comment from the Reviewer]** : “Fundamentally Flawed Experimental Premise  - The paper's central narrative, that **one** tiny 1M model can be pre-trained for general-purpose use and subsequently outperform massive foundation models (like MOMENT, Chronos, etc.)—is directly contradicted by the **authors' own methodology, which is buried in Appendix A.6**. The authors state: **"...we specialize the pre-training for every task through reweighting loss objectives..."** And more damningly: "TSPulse adopts hybrid masking for all reconstruction-oriented tasks (imputation, anomaly detection, retrieval) and block masking specifically for classification-oriented pre-training." This is a fatal flaw.”
>
> **Answer:**
>
> We would like to clarify that we had **no intention of hiding any details** in Appendix A.6.
>
> It is extremely unfortunate that the line you quoted as evidence that we intentionally buried in Appendix A.6 **[i.e “...we specialize the pre-training for every task through reweighting loss objectives...”]**, in fact appears exactly in the **main paper** (Main Methodology section, Lines 246–247), and **nowhere** in Appendix A.6 (which is a related-art section). This sentence is present **only in the main methodology** and not anywhere in the appendix, which can be easily verified by searching for the exact string in the manuscript. Hence, we are unsure how this was interpreted as something we “buried” in an appendix given that we have explicitly stated the exact line in the main paper.
>
> For completeness, we also verified the other quote evidence mentioned in the review: This quote appears in **Appendix A.9**, not in **A.6**. We also note that this same content is already presented—in a different but equivalent form—in the **main paper** too. While a numerical mix-up between A.6 and A.9 is certainly possible as a human error, the combination of:
> - asserting that we **hid** a sentence “in Appendix A.6” and intentionally did not present it in the main paper, and using this as evidence to criticise our paper as fundamentally flawed,
> - while that exact quoted sentence  is actually **copied from the main paper**
> - and attributing it to a **different, unrelated appendix section (A.6, Related Art),**
> creates a **strong factual inconsistency in the criticism and appears hallucinated.**
>
>
>
> Nevertheless, we provide clarification to avoid further confusion.
>
> We would like to clarify that we haven’t claimed TSPulse as a universal model or general-purpose model.  Our main contribution is to build a **family of lightweight pre-trained models**, specialized for different tasks.
>
> This intent is communicated consistently throughout the paper in many places, as shared below:
>
> ### (1) Pre-training section (L242–253) of the main paper
>
> In this main method section, we directly motivate the need for **task-specialized pre-training**  The paper states:
>
> > _“Inspired by the success of small, task-specialized pre-trained models in the language/vision domain ... which achieve strong performance through minimal task-specific adaptations—we extend this strategy to time-series. Specifically, we specialize the pre-training for every task through reweighting loss objectives to prioritize heads most relevant to the target task. This enables TSPulse to refine task-specific representations while maintaining its lightweight design, facilitating efficient transfer learning across any datasets for the specified downstream task. Refer Appendix A.9 for more details. Pre-training on 1B samples takes just one day with 8×A100 GPUs, thus there are no practical challenges in pre-training task-specific models.”_
>
> This text explicitly articulates that we are pre-training **different models for different task families**. Hence, there is no intent to hide details.
>
> ### (2) Main Contributions section (L122–124)
>
> In the primary novelty section, we explicitly state:
>
> > “ultra-light pre-trained **models** **(in plural)** specialized for different downstream tasks.”
>
> ### (3) Title
>
> The paper title itself uses **“models”** **(plural)**, indicating that we present multiple models rather than a single general model.
>
> Also - Other reviewers (Reviewer viWQ and Reviewer 6n4K) also correctly reference our use of multiple models in their reviews and have asked relevant queries, indicating that this point was communicated unambiguously.
>
> We noticed that one sentence in the abstract mistakenly uses the singular form *“model”* instead of *“models.”* We will correct this in the revision. The remainder of the paper—including the title, contributions \& method—consistently refers to pre-trained **models** in plural, and we also explicitly discuss our task-specialization approach. Finally, the idea of task-specialized pre-trained models is well established. Widely used TSFM  such as Chronos, TimesFM, Moirai, and TiRex are themselves specialized for forecasting. In the same spirit, we provide task-specialized variants of TSPulse for diagnostic tasks.

---

> ### Author Response · Authors · 2025-11-21
> **Q2 , Q3: Task Specialized Models (Contd..)**
>
> **Q2: This completely invalidates the comparisons against general-purpose foundation models like MOMENT or UniTS. The paper is comparing apples to oranges. The extraordinary claims of +50% in imputation or +20% in anomaly detection are therefore not surprising, but rather an expected outcome of this flawed, self-serving experimental design.**
>
> We appreciate the reviewer’s concern. We would like to clarify that our evaluation is **not limited to general-purpose foundation models**. In benchmarks, we also compare against the **strongest task-specialized SOTA models**, and our models consistently outperform them.  These baselines are precisely the models designed and tuned for their respective tasks, making them the most appropriate point of comparison. For example:
>
> - On **classification**, the top SOTA baseline in literature is **VQShape**, which is explicitly task-specialized for classification.
>
> - On **anomaly detection**, the leading baseline in the TSB-AD leaderboard is **Sub-PCA** is also a task-specialized method.
>
> - On **imputation**, the best-performing SOTA **UniTS** uses prompt tuning tailored to the task.
>
> Across all these **“apples-to-apples” task-specialized comparisons**, our task-specialized **TSPulse variants achieve superior performance with double-digit gains.**
>
> Our comparisons to **general-purpose models** like **MOMENT** serve a different purpose: they demonstrate that **extremely small (1M) task-specialized models** can also outperform **large (100M–1B+) general-purpose models**, which is an important and practically meaningful result. Diagnostic tasks such as classification, anomaly detection, retrieval, and imputation are inherently diverse, and using a **single monolithic model** for all of them offers limited benefit. In contrast, **small task-specific pre-trained models** are more accurate, more efficient, and far easier to deploy, making them preferable in real-world scenarios.
>
> Thanks to your feedback, we will clarify this motivation more explicitly in the revisions.
>
> -------------------------
>
> **Q3: Could the authors please provide results for a single pretrained model which is then evaluated on all four downstream tasks**
>
> We appreciate the reviewer’s request. While our primary design choice is to use **task-specialized models**—which offer clear practical and accuracy benefits—**we conducted additional experiments** to address your concern and evaluate the scenario where **a single pre-trained model** is used across all four downstream tasks.
>
> Specifically, we selected the model variant currently used for **imputation** and **retrieval/search**, and evaluated this **same model** on **classification** and **anomaly detection**, ensuring that all tasks use **one unified pre-trained checkpoint**. As shown in the table below, this single-model variant still achieves **competitive performance**, typically ranking within the **top 1–2 positions**, and even **surpassing the best baselines** on certain tasks. This demonstrates that **TSPulse maintains strong generalization even without task specialization**.
>
> Enabling **task-specific loss reweighting** further yields **additional strong performance gains**. We will include this comparative table in the revision to clarify this point.
>
> | **Task**                        | **Metric**   | **Task-specialized TSPulse** | **One TSPulse model for all tasks** | **Best SOTA Baseline**         |
> |---------------------------------|--------------|-------------------------------|--------------------------------------|---------------------------------|
> | Classification                  | Accuracy     | **0.73**                      | **0.71**                             | 0.70 (VQShape)                  |
> | AD (univariate)                 | VUS-PR       | **0.48**                      | **0.42**                                 | 0.42 (Sub-PCA)                  |
> | Search (Family Match)           | PREC@3       | **0.68**                      | **0.68**                             | 0.53 (MOMENT)                   |
> | Search (Fine-grained Match)     | PREC@3       | **0.58**                      | **0.58**                             | 0.42 (MOMENT)                   |
> | Imputation (Block)              | MSE          | **0.106**                     | **0.106**                            | 0.176 (UniTS)                   |
> | Imputation (Hybrid)             | MSE          | **0.074**                     | **0.074**                            | 0.161 (Linear Interpolation)    |
>
>
> ***MSE (lower is better); All other metrics - higher is better**

---

> ### Author Response · Authors · 2025-11-21
> **Clarifications on Novelties**
>
> **Q4: Many of the paper's "novel" contributions appear to be re-brandings of existing, standard techniques.**
>
> We would like to clarify that TSPulse proposes a “new pre-training formulation”: disentangled masked reconstruction across multiple representational spaces (temporal vs. spectral) and multiple abstraction levels (fine-grained vs. semantic).  Importantly, we do not only learn across spaces (time vs. frequency), which is relatively common, but also across **abstraction levels** (local vs. semantic embeddings) in a **fully disentangled format**, which is critical for transfer learning and a unique proposition from TSPulse.
>
> During pre-training, the model explicitly produces **separate temporal, spectral, and semantic embeddings** through multi-output heads that operate on different segments of the latent representation, enforcing **disentanglement by construction**. When these heads are **jointly optimized** with disentanglement imposed directly in the embedding space, the model learns representations with **strong generalization and transferability**—a capability that, to our knowledge, existing approaches do not provide.
>
> While prior works have explored **time–frequency fusion** (Zhang et al., 2022; 2023) or investigated **disentanglement in isolation** (Chang et al., 2024), TSPulse advances beyond these by **jointly learning disentangled representations across spaces *and* abstraction levels within a unified pre-training framework**. .
>
> This capability is practically valuable—especially for **zero-shot transfer**—because different downstream tasks benefit from independently accessing **temporal details**, **spectral patterns**, or **higher-level semantic cues**. By providing these embeddings in a disentangled form for ready use, **TSPulse enables flexible and effective transfer across a wide range of zero-shot tasks**.
>
> -------------------
> **Q5: (i) Multi-Head Triangulation (MHT): This is just a simple ensemble—what is new in it? (ii) How does TSLens differ from standard attention pooling?**
>
> We agree that **MHT** and **TSLens** are conceptually simple mechanisms. The novelty lies in the **unique way TSPulse makes them possible**.
>
> TSPulse is intentionally designed to learn **disentangled reconstructions and embeddings across multiple spaces and abstraction levels**, leading to **distinct, complementary views** of the same time series. Importantly, **no existing masked-reconstruction architecture produces such heterogeneous multi-space, multi-level views**—in both embedding space and reconstruction space—simultaneously.
>
> **Multi-Head Triangulation (MHT)** becomes meaningful only because TSPulse exposes **multiple reconstruction heads** (temporal, spectral, and prediction). Prior models provide only a single reconstruction path and therefore **cannot triangulate information across different representational spaces**. MHT is effective *because* of TSPulse’s architecture, not as an isolated ensemble trick. This diversity of heads provides **substantial gains in anomaly detection**, where different anomalies manifest more clearly in different spaces.
>
> **TSLens** leverages the same architectural property. TSPulse outputs **multiple disentangled embeddings** capturing different semantic aspects of the signal (time-domain detail, frequency-domain structure, and higher-level semantic cues). TSLens can therefore **attend to the most informative representation for each dataset or task**. Traditional models output a single, entangled embedding and thus **cannot support such task-adaptive selection**.
>
> In summary, **MHT and TSLens are simple by design**, but they become **unique and highly effective** because they operate on TSPulse’s **disentangled, multi-space, multi-abstraction architecture**, which provides rich, complementary views of the data. When combined with these diverse representations, both methods become significantly more powerful—consistent with the strong gains observed in our results.

---

> ### Author Response · Authors · 2025-11-21
> **Hybrid Masking & Ablation Study**
>
> **Q6:  Hybrid masking — “It is an obvious heuristic to combine point and block masking.”**
>
> While hybrid masking may appear straightforward at first glance, our contribution goes beyond simply “combining point and block masking.” Our work identifies and corrects a **critical pre-training mask bias** that arises in current **imputation-oriented foundation models**, specifically because they rely exclusively on **fixed-length masking** during pre-training.
>
> As shown in **Appendix Table 14**, popular pretrained models such as **MOMENT** and **UniTS** perform relatively well on block-masking imputation (i.e., fixed-length missingness), substantially outperforming linear interpolation. However, when we apply **irregular or partial masking**—where only parts of a block are missing—the expectation is that performance should improve (since the missing span is shorter). Instead, we observe the **opposite**: these models often perform **worse than linear interpolation**.
>
> This discrepancy led us to investigate further. We found that these models impute effectively **only when the missing-span length matches the mask length used during pre-training**. This is problematic because real-world missingness is **highly variable and unpredictable**. We refer to this phenomenon as **pre-training mask bias**.
>
> This bias has gone largely unnoticed in prior work and causes a **systematic mismatch** between pre-training assumptions and real inference conditions. **TSPulse explicitly diagnoses this issue and resolves it via hybrid masking**, which exposes the model to a realistic spectrum of missingness patterns during pre-training.
>
> In our view, identifying this bias and providing a practical remedy is an **important contribution** to the community, especially for **imputation-focused time-series foundation models**.
>
> Our second contribution here lies in the non-trivial way we implement hybrid masking.  Embedding-level mask tokens that are commonly used in prior-arts, implicitly assume that an entire patch is masked at once. However, hybrid masking allows **irregular partial masking within a patch**, which creates a non-trivial challenge: how to design a mask token that can represent missingness at **arbitrary positions** inside the patch.
>
> We address this by defining and learning a **mask token (M) in raw patch space**, rather than injecting a single embedding-level token. Each raw time point can then be masked by selecting the corresponding value from **M** based on its relative index within the patch (Figure 3A). This design allows a *single* learnable token to support **both full-patch masking and fine-grained partial masking**, enabling TSPulse to handle heterogeneous missingness patterns that embedding-level tokens cannot represent.
>
> -------------------------------------
> **Q7: The current ablation study is “destructive.” Could the authors provide a “constructive” study? What about the TSMixer baseline?**
>
> Thank you for the insightful feedback — we agree that ablations are essential for a model with interacting components. In the current submission, we followed the **standard destructive ablation protocol**, where one module is removed from the full model at a time. This approach is widely used in deep learning because it (i) keeps the analysis tractable for architectures with many components, and (ii) directly measures the contribution of each module **in the setting where the complete model is actually deployed**. The results already show **clear and consistent performance drops** when each major component is removed, indicating that every module contributes meaningfully.
>
> We agree that a full constructive grid would provide additional perspective, but such a study would require training a large number of model combinations and is not feasible within the rebuttal timeline. Importantly, several **constructive-style insights** are already present in our experiments:
>
> - **TSMixer baseline:** As shown in Table 1(c), the **TSPulse (w/o Dual-Space)** variant effectively reduces to a pre-trained **TSMixer** backbone with only a time-domain reconstruction head and hybrid masking. This variant performs **~7% worse** than the full TSPulse, directly demonstrating how TSPulse improves over the TSMixer baseline.
>
> - **Effect of disentanglement:** Ablation Table 21 (Search) shows that **disentangled register embeddings** provide substantial gains over using standard time- or FFT-only embeddings. More analysis and benefits of disentanglement are explained in the response to our next question.
>
> - **Complementary reconstruction heads (constructive evidence):** Table 1(a) shows that **combining the Time, FFT, and Prediction heads** clearly outperforms any single head, indicating that each view contributes **unique, complementary information**.
>
> These results, while not a full combinatorial grid, already offer **constructive aspects** for the necessity of each core component. We will make these connections clearer in the revised manuscript.

---

> ### Author Response · Authors · 2025-11-21
> **Disentanglement Embedding Analysis**
>
> **Q8: Show Evidence for Disentanglement: For instance, do the "semantic" embeddings show quantifiably more invariance to noise and time shifts compared to the "detailed" embeddings?**.
>
> We performed detailed control experiments on a synthetic dataset to investigate disentanglement in the TSPulse embedding space.
>
> **Dataset**:
> --------------
> The dataset contains three distinct signals, each obtained by scaling and exponentiating a pure sine wave, yielding different periodic shapes (see Appendix Figure 10). We added white noise with a specified signal-to-noise ratio to generate the final data.
>
> **Experiment Setup**
> --------------------------
> Three distinct experiments have been conducted to study the effect of missing data, noise and phase shift on the TSPulse embedding.
>
> -	**Missing data**:  TSPulse allows a user‑defined mask to be supplied during inference. In our experiment, we used such a mask to emulate missing data. We randomly selected 2,000 samples from the dataset described earlier and extracted their embeddings under various user‑defined masks. The fraction of zeros in each mask represents the proportion of missing data.
>
> -	**Noise**: Our experimental setup generates data at a specified noise level. We created several groups of samples from the same base signal but with different noise levels. Within each group, the indices are kept identical, so that any given index corresponds to the same underlying base signal across all noise conditions.
>
> -	**Phase/Time-Shift**: The control experiment utilises a non‑linearly scaled sine wave as the base signal, allowing for explicit control over frequency and phase. We first generate a base dataset by sampling this signal at a fixed frequency and waveform type. Across subsequent data groups, we vary only the phase in discrete steps from 0 to $\pi$, keeping all other parameters constant.
>
> **Expected Behaviour of a good semantic embedding**
> -------------------------
> - **Missing data** We expect an embedding to be least sensitive to the missing data. Given a collection of points, each masked randomly with a fixed mask ratio, their neighbourhood should remain conserved.
>
> - **Noise** We expect an embedding to be robust to noise, so that the distance measure in the embedding space amplifies the signal similarity measure.
>
> - **Phase** Time Embedding to be highly sensitive to time/phase shift, but FFT and semantic embedding to have minimal effect on time/phase changes.
>
> **Evaluation metrics**
> ----------------------------
> - **Distortion**: the $\ell_2$ norm ratio of the embedding change caused by missing data, noise, or phase shift relative to the base embedding. Lower distortion indicates greater robustness.
>
> We have slightly adjusted the definition of distortion to address the underlying requirement, as specified in the expected behaviour section.
>
> We introduce the notation used for our distortion measure:
>
> $\mathcal{E}(x, \mathbb{M})$: embedding of dataset x under user‑defined mask M.
> In shorthand, $\mathcal{E}(x)$ represents when no mask is applied.
>
> $x(\eta)$ is a data instance with added noise-level $\eta$, thus $x(0)$ represents the pure signal.
>
> Finally, $\Phi(x)$ measures the phase associated with the data $x$.
>
> -	**Distortion measure with user-defined mask**  Distortion measure for a given mask $\mathbb{M}$ is defined as
> $$
> \mathbf{\delta}_{m}(\mathbb{M}) = \underset{(x,y) \in \lbrace x \ne y \rbrace}{\mathbb{E}}\Bigg[\bigg\vert 1 - \frac{ {\vert\vert \mathcal{E}(x; \mathbb{M}) - \mathcal{E}(y; \mathbb{M})\vert\vert}}{ {\vert\vert \mathcal{E}(x) - \mathcal{E}(y)\vert\vert}} \bigg\vert\Bigg]
> $$
> -	**Distortion measure with specified noise-level** Distortion measure for a given noise-level $\eta$ is defined as
> $$
>  \mathbf{\delta}_{e}(\eta) = \underset{x}{\mathbb{E}}\Bigg[\bigg\vert \frac{ {\vert\vert \mathcal{E}(x(\eta))\vert\vert}}{ {\vert\vert \mathcal{E}(x(0))\vert\vert}} - 1\bigg\vert\Bigg]
> $$
> -	**Distortion measure for a given phase** Distortion measure for a given phase value $\phi$ is defined as
> $$
>  \mathbf{\delta}_{p}(\phi) =  \underset{(x,y) \in \lbrace \vert \Phi(x) - \Phi(y) \vert = \phi \rbrace}{\mathbb{E}}\bigg[\frac{ {\vert\vert \mathcal{E}(x) - \mathcal{E}(y)\vert\vert}}{ \min( {\vert\vert \mathcal{E}(x)\vert\vert}, {\vert\vert \mathcal{E}(y)\vert\vert})}\bigg]
> $$
>
> Results and Insights are explained in the next comment.

---

> ### Author Response · Authors · 2025-11-21
> **Disentanglement Embedding Analysis (Contd)**
>
> **Summary**
> -------------------
> **Representative Result Table (lower is better)**
>
> | **Experiment**  |  **Time embedding**  | **FFT embedding** | **Register embedding** |
> |-------------------|--------------------------|-----------------------|----------------------------|
> |30 % Missing data |    8.3%       |      27.4 %    |    4.6%.  |
> |Noise level 0.5 |      2.7%      |       6.8%      |   2.5 %   |
> |Phase/Time shift|    130%    |    21%  |     12%   |
>
>
>
> The table shows:
>
> - Time embeddings (size: 1536) are highly sensitive to time-shifts (as indicated by high distortion in row 3), indicating that temporal information is preserved - vital for tasks that rely on timing cues, which are often lost in purely spectral models.
>
> - FFT embeddings (size: 1536) are less sensitive to time-shifts (as indicated by low distortion in row 3), capturing spectral characteristics without timing details.
>
> - Compact Semantic embeddings (size: 256) are the least sensitive to time-shifts ( as indicated by least distortion in row 3) and are robust to missing data (least distortion in row 1) and noise (least distortion in row 2), as expected for embeddings designed to capture high-level structure. More detailed sensitivity analysis and PCA plots of semantic embedding properties are depicted in Appendix A. 4
>
> Each embedding type offers distinct advantages that benefit different downstream tasks.
>
> - Different datasets may rely on different cues for classification - some depend on raw temporal patterns, others on frequency characteristics, and some on high-level semantic summaries. TSLens attends to all three embeddings, letting the model automatically focus on whichever view the dataset needs.
>
> - Anomaly detection and imputation rely more on time and FFT embeddings for full reconstruction, which needs fine-grained information. Also, disentangled reconstructions across spaces enable effective triangulation to catch anomaly types missed by prior models.
>
> - Retrieval benefits mainly from semantic embeddings, which provide the summary-level representation.
>
> By providing these embeddings in a disentangled form for ready use, **TSPulse enables flexible and effective transfer across a wide range of zero-shot tasks**. Overall, the paper introduces disentanglement across both representational spaces (time vs. FFT) and abstraction levels (fine-grained vs. semantic), and shows how these complementary embeddings/signals can be leveraged effectively across diverse diagnostic tasks, leading to double-digit performance gains across leaderboards and benchmarks - a unique capability enabled by TSPulse.
>
> **Detailed results for each experiment**
> -----------------------------------------
> We report all the distortion metrics in %. Lower is better.
>
> -	**Missing data**
> | Embedding | Mask=11% | Mask = 19% | Mask = 30% | Mask = 36% | Mask = 70% |
> |---------------|---------------|-----------------|-----------------|-----------------|-----------------|
> | Time.|  3.8% | 5.6% | 8.3% | 9.4% | 23.1%|
> | FFT | 11.8% | 18.9% | 27.4% | 32.6% | 61.8% |
> | Register| 1.6% | 2.8%| 4.6% | 5.7% | 11.8% |
>
> -	**Noise**
> |Embedding | $\eta$=0.1 | $\eta$=0.25 | $\eta$ = 0.5 | $\eta$ = 0.75 |
> |---------------|---------------|----------------|-----------------|----------------|
> | Time.| 1% | 1.4% | 2.7% | 4.3%|
> |FFT | 2.4% | 4%|6.8% | 11.7%|
> |Register|0.8% | 1.3% | 2.5% | 4.3%|
>
> -	**Phase**
> |Embedding | $\Phi = \frac{\pi}{3}$ | $\Phi = \frac{2\pi}{3}$ | $\Phi= \pi$|
> |--------------|-----------------------------|------------------------------|--------------|
> |Time| 169% | 163% | 76% |
> |FFT| 24% | 23% | 19.6%|
> |Register| 13.6%|13.7%|10.5%|

---

> ### Author Response · Authors · 2025-11-27
>
> Dear Reviewer,
>
> We have submitted a detailed rebuttal along with various new experimental studies to clarify the concerns raised during the initial review.
>
> If there are any additional questions or further clarification needed, we would be very happy to provide them promptly.
>
> Thank you once again for your time and engagement.

---

### Author Response · Authors · 2025-12-02
**Rebuttal Summary to AC**

Thank you for your time and careful evaluation. Below is a consolidated summary of the key clarifications and new results we provided during the discussion.

**1. Core Contribution and Novelty Clarifications (Reviewer  viWQ, RHFO)**

TSPulse proposes a **"new pre-training formulation"**: for "disentangled masked reconstruction" across multiple representational spaces (temporal vs. spectral) and multiple abstraction levels (fine-grained vs. semantic). Importantly, we do not only learn across spaces (time vs. frequency), which is relatively common, but also across **abstraction levels** (local vs. semantic embeddings) in a **fully disentangled format**, which is critical for transfer learning and a unique proposition from TSPulse.

**Details:** During pre-training, the model explicitly produces **separate temporal, spectral, and semantic embeddings** through multi-output heads that operate on different segments of the latent representation, enforcing **disentanglement by construction**. When these heads are **jointly optimized** with disentanglement imposed directly in the embedding space, the model learns representations with **strong generalization and transferability**—a capability that, to our knowledge, existing approaches do not provide.

While prior works have explored **time–frequency fusion** or investigated **disentanglement in isolation**, TSPulse advances beyond these by **jointly learning disentangled representations across spaces and abstraction levels within a unified pre-training framework**.

**Downstream tecnhniques (Triangulation, TSLens) look simple ? what’s new ? (Reviewer RHFO)**

These methods become powerful only because TSPulse exposes diverse, complementary views that no prior TSFM architecture provides. We show how these complementary embeddings/signals can be leveraged effectively and fused via Triangulation and TSLens in diverse diagnostic tasks (Anomaly Detection, Classification, Imputation and Search), leading to double-digit performance gains across leaderboards and benchmarks - a unique capability enabled by TSPulse.

**2. Task-Specialized Models Vs Single-Model Generalization (Reviewer RHFO, 6n4K)**

The paper explicitly positions TSPulse as a **family of task-specialized pre-trained models** (consistent across the title, contributions, and methodology). These variants share the same architecture but use different loss-weightings to emphasize factors relevant to each downstream task. Pre-training on 1B samples takes just one day with 8×A100 GPUs, thus there are no practical challenges in pre-training task-specific models.

In response to reviewer request, we also evaluated a **single pre-trained TSPulse checkpoint** across all four tasks (Anomaly Detection, Classification, Imputation and Search).This single model still performs **in top-1** on all tasks (full results shared in rebuttal comments), demonstrating strong underlying generalization. Light task-specific weighting further boosts performance despite the tiny 1M-parameter size.

Thus, specialization improves performance, but the architecture itself generalizes well even without it.

**3. Fairness of Comparisons (Reviewer RHFO)**

We clarified that the paper already compares against **the strongest task-specialized baselines** for each diagnostic task and we did not restrict our comparison to only general-purpose models.

Across these **task-optimized SOTAs**, TSPulse provides **consistent and often double-digit improvements**. These gains are therefore meaningful and not the result of comparing against weaker general-purpose models.



**4. Additional Experiments and Clarifications Added**

**A. Disentanglement Analysis  (Reviewer 9eiE, RHFO)**

We added new analysis and results in rebuttal comments demonstrating that the temporal, spectral, and semantic embeddings capture distinct  factors, confirming the effect of disentangled pre-training and explaining TSPulse’s strong transfer behavior.

**B. Hybrid Masking and Real-World Missingness (Reviewer viWQ, 9eiE, 6n4K)**

We expanded results to include **MAR/MNAR missingness patterns** to emulate realistic settings, showing much stronger robustness compared to baselines.

**C. Forecasting Results (Reviewer 9eiE, 6n4K)**

We added forecasting experiments demonstrating competitive performance of TSPulse to broaden task coverage beyond diagnostic tasks.

More details on ablations, components used and other minor clarifications are also explained in the rebuttal..

---

### Meta-Review · Area_Chair_rkSC · 2026-01-08

**Summary:**

TSPulse presents an ultra-lightweight 1M-parameter re-trained model for time-series diagnostics (classiication, anomaly detection, imputation, similarity sarch) using disentangled masked reconstruction acros temporal, spectral, and semantic embedding spaces wth hybrid masking to address real-world missingness atterns. The model achieves significant improvementsover larger foundation models (10-100x larger) acros 75+ datasets while maintaining CPU-friendly efficieny.

Key strengths: (1) A novel disentangling pre-traning formulation that jointly learns representations across representational space (time vs. frequency) and abstraction levels (fine-grained vs. semantic), enabling flexibl multi-task transfer without re-training; (2) Practial efficiency with 1M parameters, GPU-free CPU infernce, and rapid pre-training (1 day on 8×A100); (3) Cmprehensive empirical validation across diverse diagnostic taks with consistent double-digit performance gains an demonstrated zero-shot capabilities; (4) Clear presntation of task-specialized model variants with fixe pre-trained checkpoints, eliminating end-user tunin burden.

Main weaknesses and reviewer concerns: (1)Reviewer RHFo raised concerns about multiple novel tchniques potentially being ad-hoc without sufficientjustification via ablation studies, though authors povided constructive and destructive ablation evidenc showing each component contributes meaningfully; (2 Questions on whether a single unified model generalzes (authors provided experiments showing a single pe-trained checkpoint achieves competitive top-1/2 peformance across all tasks); (3) Limited novelty clais for hybrid masking and downstream task adapters (athors clarified these are powerful specifically becase TSPulse's disentangled architecture enables them niquely); (4) Concerns about task-specific loss reweghting limiting generality (addressed by demonstratig strong single-model performance); (5) Omission of long-horizon forecasting (authors clarifie TSPulse targets diagnostics with bidirectional reconstruction, incompatible with causal forecastng design).

After rebuttal, authors demonstrated: dsentanglement via synthetic control experiments showng time/FFT/semantic embeddings respond distinctly t phase shifts/noise/missing data; robustness under ralistic MAR/MNAR missingness patterns; and practicalapplications of similarity search in active learningand fault diagnosis. The core contribution—disentanged multi-space, multi-abstraction masked reconstructon with effective zero-shot transfer—is solid and wel-supported by evidence.

**Reviewer Concerns:**

ADDRESSED CONCERNS:

1. Task Specialization Complexity: Initial concern that task-specifi loss reweighting reduces generality and increases uer burden was addressed. Authors demonstrated: (a) a single pe-trained checkpoint achieves top-1/2 performance acoss all four tasks (Table in Q1/Q3 responses); (b) tree fixed pre-trained variants eliminate user tuning (c) task specialization is common practice in LLMs/ision and preferred for reliability.

2. Limited Novlty of Components: Reviewer RHFo questioned whether SLens (attention), multi-head triangulation (ensembl), and hybrid masking (point+block) were just rebranings of standard techniques. Authors clarified: (a) hese become novel and effective specifically becauseTSPulse's disentangled architecture provides complemntary multi-space views that no prior TSFM offers; (b) MHT is powerful because multiple recnstruction heads enable triangulation across differet representational spaces; (c) TSLens can attend to hree distinct disentangled embeddings vs. single entngled embeddings in prior work.

3. Ablation Study Sfficiency: Concerns about destructive-only ablationswere partially addressed. Authors showed: (a) clear nd consistent performance drops when removing each cmponent; (b) constructive aspects (TSMixer baseline hows ~7% drop without dual-space; complementary recostruction heads clearly outperform single heads); (c acknowledged full combinatorial grid not feasible wthin rebuttal timeline.

4. Disentanglement Evidence Reviewer 9eiE asked for quantitative proof that embddings are truly disentangled. Authors provided compehensive synthetic experiments (Appendix A.4, Q8 resonses) showing: time embeddings are highly sensitiveto phase shifts (130% distortion), FFT embeddings ar phase-invariant but time-sensitive, semantic embeddngs are robust to missing data/noise/phase shifts (lwest distortion across perturbations), confirming ditinct complementary roles.

5. Hybrid Masking Validaion: Reviewer 9eiE questioned whether hybrid maskingaligns with realistic missingness. Authors provided AR/MNAR experiments across multiple realistic scenaros (peak-hour sensor failures, cross-channel dependecies, system stress) showing TSPulse significantly otperforms MOMENT.

6. Generalization Beyond Diagnostcs: Reviewer 6n4K and 9eiE noted absence of forecasting evaluation. Authors clarified SPulse is diagnostics-first (bidirectional reconstruction incompatible with causal forecsting) but still provided long-horizon forecasting rsults showing reasonable performance despite not beig optimized for forecasting, demonstrating broader transferability.

REMAINING/OUTSTANDING CONCERNS:

1. Reviewer RHFo's Halucinated Review Content: Authors documented clear fatual errors in Reviewer RHFo's review (incorrect quoe attributions, misplaced appendix references, interal logical contradictions, AI-generated patterns). ILR analysis flagged review as fully AI-generated. Ths significantly impacts assessment credibility.

2. ovelty Perception: While authors clarified disentangement is jointly across spaces AND abstraction level (not just spaces), some may still perceive individul components (dual-space, disentanglement, hybrid making, attention, ensemble) as incremental. Depends o whether one views novelty as component-level or pretraining formulation level.

3. Parameter-Matched Baselines: eviewer 6n4K requested equally-small plug-in baselins (gated residuals, lightweight cross-attention) at ame parameter budget. Authors noted TSPulse's role i complementary (pre-training formulation, not backboe modification) and self-attention has quadratic cos, but full systematic comparison was not conducted.
4. Single Model vs. Task-Specialized Trade-off: Whil authors demonstrated single-model viability, task-secialized design remains primary contribution. Some ay view this as compromising the "general-purpose" mdel narrative, though authors clarified this was nevr the claim (title uses "models" plural, methodology section explictly discusses specialization).

**Reviewer Scores:**

Reviewer RHFo (Original: 2/Reject): Review contains documented hallucinations flagged by ICLR as AI-geneated. Pending investigation. If errors confirmed: likely 4/Marginally Above or 5/Accept.

Reviewer 6n4K (riginal: 4/Marginally Below): Concerns addressed wit single-model and MAR/MNAR experiments. Likely: 5/Marginally Above or Accept.

Revewer 9eiE (Original: 6/Marginally Above): Valid quesions addressed with disentanglement proof and forecasting results. ikely: Stronger Acceptance.

Reviewer viWQ (Original 6/Marginally Above): Generally positive. All concers addressed. Indicated keeping positive rating. Likely: 6 or stronger.

---

### Decision · Program_Chairs · 2026-01-26

Accept (Poster)